# Phase separation of FSP1 promotes ferroptosis

Toshitaka Nakamura[1], Clara Hipp[2,3], André Santos Dias Mourão[2], Jan Borggräfe[2,3], Maceler Aldrovandi[1], Bernhard Henkelmann[1], Jonas Wanninger[1], Eikan Mishima[1,4], Elena Lytton[1], David Emler[1], Bettina Proneth[1], Michael Sattler[2,3] & Marcus Conrad[1 ✉]

Ferroptosis is evolving as a highly promising approach to combat difficult-to-treat tumour entities including therapy-refractory and dedifferentiating cancers[1–3]. Recently, ferroptosis suppressor protein-1 (FSP1), along with extramitochondrial ubiquinone or exogenous vitamin K and NAD(P)H/H[+] as an electron donor, has been identified as the second ferroptosis-suppressing system, which efficiently prevents lipid peroxidation independently of the cyst(e)ine–glutathione (GSH)–glutathione peroxidase 4 (GPX4) axis[4–6]. To develop FSP1 inhibitors as next-generation therapeutic ferroptosis inducers, here we performed a small molecule library screen and identified the compound class of 3-phenylquinazolinones (represented by icFSP1) as potent FSP1 inhibitors. We show that icFSP1, unlike iFSP1, the first described on-target FSP1 inhibitor[5], does not competitively inhibit FSP1 enzyme activity, but instead triggers subcellular relocalization of FSP1 from the membrane and FSP1 condensation before ferroptosis induction, in synergism with GPX4 inhibition. icFSP1-induced FSP1 condensates show droplet-like properties consistent with phase separation, an emerging and widespread mechanism to modulate biological activity[7]. N-terminal myristoylation, distinct amino acid residues and intrinsically disordered, low-complexity regions in FSP1 were identified to be essential for FSP1-dependent phase separation in cells and in vitro. We further demonstrate that icFSP1 impairs tumour growth and induces FSP1 condensates in tumours in vivo. Hence, our results suggest that icFSP1 exhibits a unique mechanism of action and synergizes with ferroptosis-inducing agents to potentiate the ferroptotic cell death response, thus providing a rationale for targeting FSP1-dependent phase separation as an efficient anti-cancer therapy.

Ferroptosis, a metabolic form of non-apoptotic cell death characterized by iron-dependent lipid peroxidation, has been defined only recently[1,8]. Ferroptosis has attracted tremendous interest because of its high relevance to human diseases such as neurodegenerative disorders, tissue damage during cold exposure, ischaemia–reperfusion injury and cancer[9–12]. In particular, triggering ferroptosis in the context of malignancies has emerged as a highly promising approach that shows synergistic effects with cancer immunotherapy and even kills therapy-resistant and metastatic cancers[2,3,13–15]. We recently showed that FSP1 represents a powerful backup system for the guardian of ferroptosis, known as GPX4, rendering tumours resistant to inhibition of this node[5,6]. However, because the first described FSP1-specific inhibitor iFSP1 (ref. 5) does not qualify to be further developed as an anti-cancer drug, owing to its limited potential for medicinal chemistry development in terms of an unfavourable structure and substitution pattern[16], next-generation, efficacious in vivo FSP1 inhibitors for tumour treatment are urgently required.

## icFSP1 triggers ferroptosis

To identify possible in vivo-applicable FSP1-specific inhibitors as potential future drugs to combat difficult-to-treat cancers, we carefully revalidated hit compounds from ~10,000 drug-like small molecule compounds (as described previously[5]) in terms of their potential for medicinal chemistry development, using cheminformatics tools[17] for the prediction of physicochemical properties and drug-likeness. Hit validation studies identified the class of 3-phenylquinazolinones (represented by the lead compound icFSP1) as a class of potent pharmacological inhibitors of FSP1 (Fig. 1a). Preliminary structure–activity relationship studies have yet to identify compounds with substantial improvement over icFSP1 (Extended Data Fig. 1a). Treatment with icFSP1 caused marked lipid peroxidation and associated ferroptotic cell death in 4-hydroxytamoxifen (TAM)-inducible *Gpx4*-knockout mouse Pfa1 cells[18] stably overexpressing human FSP1 (hFSP1) and in the human fibrosarcoma HT-1080 cell line (Fig. 1b–g and Extended

[1]Institute of Metabolism and Cell Death, Molecular Targets and Therapeutics Center, Helmholtz Munich, Neuherberg, Germany. [2]Bavarian NMR Center, Department of Bioscience, School of Natural Sciences, Technical University of Munich, Garching, Germany. [3]Institute of Structural Biology, Molecular Targets and Therapeutics Center, Helmholtz Munich, Neuherberg, Germany. [4]Division of Nephrology, Rheumatology and Endocrinology, Tohoku University Graduate School of Medicine, Sendai, Japan. ✉e-mail: marcus.conrad@helmholtz-munich.de

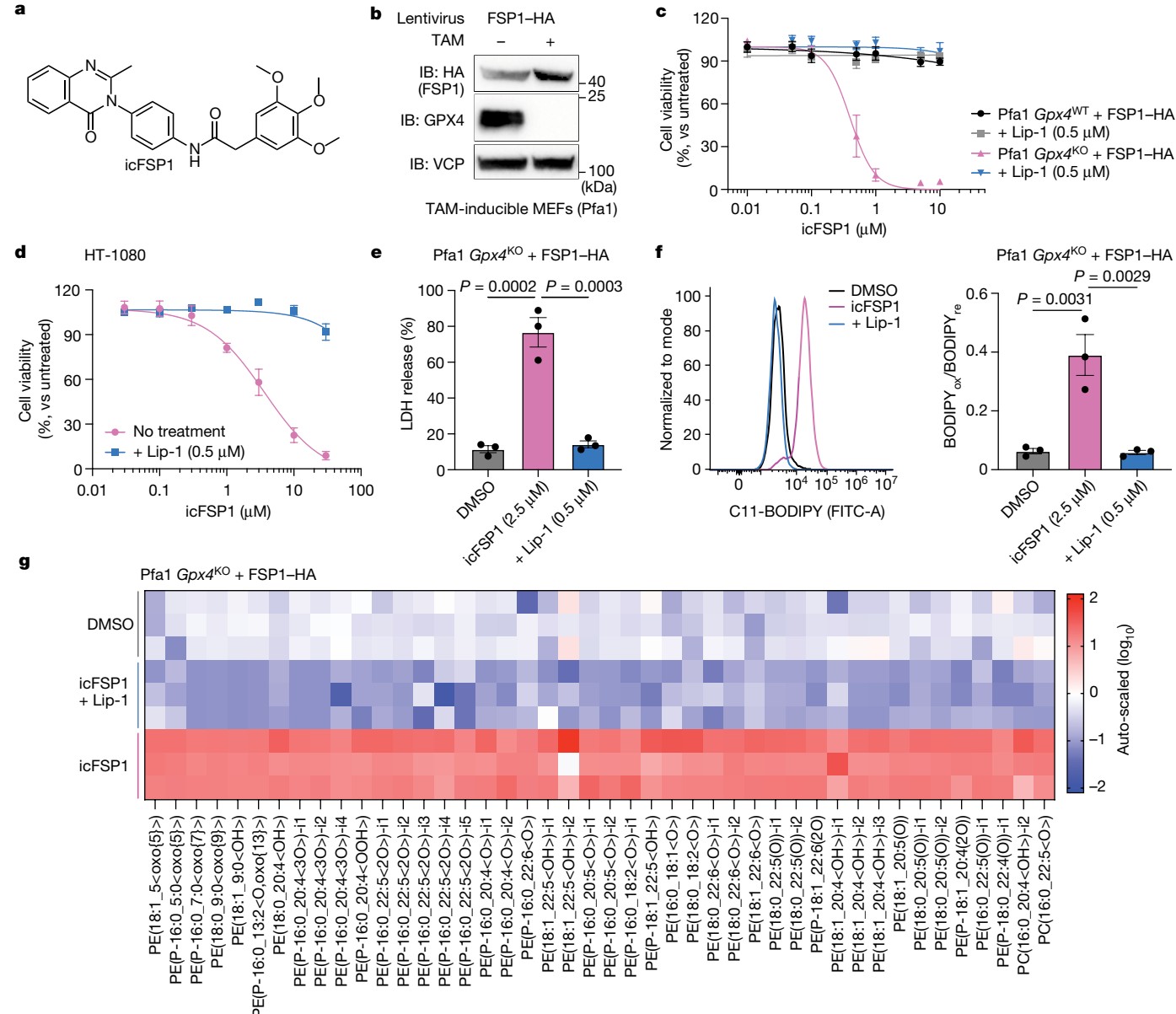

**Fig. 1 | icFSP1 induces ferroptosis in synergy with GPX4 inhibition.**
**a**, Chemical structure of icFSP1. **b**, Representative immunoblot (IB) analysis of GPX4, HA (FSP1) and VCP expression in TAM-induced *Gpx4*-knockout mouse embryonic fibroblasts (MEFs; Pfa1 cells) stably overexpressing HA-tagged hFSP1 from one of two independent experiments. **c**, Cell viability of wild-type or knockout *Gpx4* (*Gpx4*$^{WT}$ or *Gpx4*$^{KO}$, respectively) Pfa1 cells stably expressing HA-tagged hFSP1 treated with icFSP1 alone or in combination with the ferroptosis inhibitor liproxstatin-1 (Lip-1; 0.5 μM) for 48 h. **d**, Cell viability of HT-1080 cells treated with icFSP1 and 0.5 μM Lip-1 for 72 h. **e**, Lactate dehydrogenase (LDH) release determined after treating *Gpx4*-knockout Pfa1 cells overexpressing HA-tagged hFSP1 with DMSO, 2.5 μM icFSP1, or 2.5 μM icFSP1 + 0.5 μM Lip-1 for 24 h. **f**, Lipid peroxidation evaluated by C11-BODIPY

581/591 staining after treating *Gpx4*-knockout cells stably overexpressing HA-tagged hFSP1 with DMSO, 2.5 μM icFSP1, or 2.5 μM icFSP1 + 0.5 μM Lip-1 for 3 h. Representative plots of one of three independent experiments (left) and quantified median values of three independent experiments (right) are shown. BODIPY$_{ox}$/BODIPY$_{re}$, ratio of oxidized to reduced BODIPY. Data represent the mean ± s.e.m. of three (**c**,**e**,**f**) or four (**d**) independent experiments. *P* values were calculated by one-way ANOVA followed by Tukey's multiple-comparison test (**e**,**f**). **g**, Lipid peroxidation profiles measured by liquid chromatography and tandem mass spectrometry (LC–MS/MS) after treatment of *Gpx4*-knockout Pfa1 cells stably overexpressing HA-tagged hFSP1 with DMSO, 5 μM icFSP1, or 5 μM icFSP1 + 0.5 μM Lip-1 for 5 h. The heat map shows three technical replicates from one of two independent experiments.

Data Fig. 1b). icFSP1-induced cell death was rescued by ferroptosis inhibitors, but not by inhibitors targeting other forms of cell death, thus confirming its specificity for ferroptosis. Killing of (cancer) cells by targeting FSP1 usually requires co-treatment with other types of canonical ferroptosis inducers[5,19], such as the system Xc⁻ inhibitor erastin, the glutamate cysteine ligase (GCL) inhibitor L-buthionine sulfoximine (BSO), and the GPX4 inhibitors (1*S*,3*R*)-RSL3 (RSL3), ML210 and FIN56, as well as the iron oxidation compound FINO2

(refs. 1,5,20–22), but not with compounds inducing other forms of cell death (Extended Data Fig. 1c–e). Thus, it was surprising that treatment of HT-1080 cells, the primary cell model in ferroptosis research, with icFSP1 alone or doxycycline-inducible *FSP1* knockout for 72 h was sufficient to trigger ferroptosis (Fig. 1d and Extended Data Fig. 1f–h), in contrast to a panel of different human cancer cell lines (Extended Data Fig. 1c). To address whether icFSP1 may have off-target effects, HT-1080 and HEK293T cells and primary peripheral blood mononuclear cells

(PBMCs) were exposed to the FSP1 inhibitors (icFSP1 and iFSP1) for 72 h and 24 h, respectively. In these experiments, icFSP1 did not show off-target effects even at higher concentrations as compared with iFSP1 (Extended Data Fig. 1h–j). Apart from this, *FSP1*-knockout cells treated with increasing concentrations of icFSP1 did not show any additional synergistic effects when cells were co-incubated with a GPX4 inhibitor (Extended Data Fig. 1k,l), indicating that icFSP1 should be considered to be a selective FSP1 inhibitor.

To investigate whether icFSP1 also induces death in non-human cells, we included two different mouse cell lines and one rat fibroblast cell line in our study (that is, 4T1, B16F10 and Rat1, respectively), showing that co-treatment with RSL3 or *Gpx4* knockout failed to synergistically kill these cells (Extended Data Fig. 2a–g). In line with this, lipid peroxidation and cell viability were not affected by icFSP1 in *Gpx4*-knockout Pfa1 cells stably overexpressing mouse FSP1 (mFSP1; Extended Data Fig. 2h,i), suggesting that icFSP1 specifically inhibits the human isoform. Additionally, we tested other orthologues of FSP1 in Pfa1 cells, overexpressing FSP1 from *Gallus gallus* (chicken) and *Xenopus laevis* (frog). Although FSP1 expression (except for that from *X. laevis*) fully prevented RSL3-induced ferroptosis, icFSP1 reduced cell viability only in cells overexpressing hFSP1 (Extended Data Fig. 2j–l). These data therefore reinforce the notion that icFSP1 is an hFSP1-specific inhibitor.

## icFSP1 induces FSP1 condensation

To investigate the mechanism of action of these next-generation FSP1 inhibitors, the direct inhibitory activity of FSP1 was measured in vitro by using an established assay with recombinant hFSP1 enzyme (Fig. 2a). Whereas the enzymatic activity of hFSP1 was inhibited by iFSP1 in a cell-free system, as reported[5,6], icFSP1 did not inhibit hFSP1 activity in the low-micromolar range, although it clearly affected cell viability and lipid peroxidation in Pfa1 cells overexpressing hFSP1 (Fig. 2b and Extended Data Fig. 3a–k). In fact, the estimated half-maximal inhibitory concentration ($IC_{50}$) of icFSP1 in the in vitro assay (enzyme) was more than 100-fold higher than the half-maximal effective concentration ($EC_{50}$) observed in Pfa1 cells (Fig. 2b and Extended Data Fig. 3i). These results strongly argue that icFSP1, as compared with iFSP1, uses a different mechanism of action to inhibit hFSP1 activity in cells.

To shed light on the mechanism of action, we first analysed whether icFSP1 may decrease the expression levels of hFSP1. Immunoblotting of hFSP1-expressing H460 or HT-1080 cells after icFSP1 treatment for 48 h and 72 h showed that expression levels of hFSP1 were not affected by icFSP1 treatment (Extended Data Fig. 3l–n). Next, we considered that icFSP1 might change the subcellular localization of hFSP1 by detaching it from lipid membranes, thereby preventing its anti-ferroptotic function of scavenging phospholipid radicals through ubiquinone and/or vitamin E or K[4–6]. To this end, hFSP1 fused to enhanced green fluorescent protein and streptavidin (hFSP1–EGFP–Strep) was stably overexpressed in Pfa1 cells and its localization was monitored in response to icFSP1 treatment. Unlike iFSP1, icFSP1 markedly changed the subcellular localization of hFSP1–EGFP–Strep, as illustrated by the appearance of distinct foci and cellular condensates (Fig. 2c, Extended Data Fig. 3o and Supplementary Video 1). These condensates accumulated in cells in a time-dependent manner (Fig. 2d) and only occurred in cells expressing hFSP1 and not in those expressing mFSP1 (Extended Data Fig. 3p), corroborating that icFSP1 is specific for the human orthologue. To test whether the change in subcellular localization of hFSP1 causes induction of ferroptosis, *Gpx4*-knockout Pfa1 cells stably overexpressing hFSP1 fused to blue fluorescent protein (BFP) were established. hFSP1–BFP signal was monitored by live-cell imaging of cells co-stained with Liperfluo (a lipid hydroperoxide sensor) and propidium iodide, which can only stain nuclei when the plasma membrane becomes leaky. Straight after treatment of hFSP1–BFP-expressing cells with icFSP1, condensates were induced followed by Liperfluo oxidation, whereas lipid peroxide signals gradually increased in cells until cells became

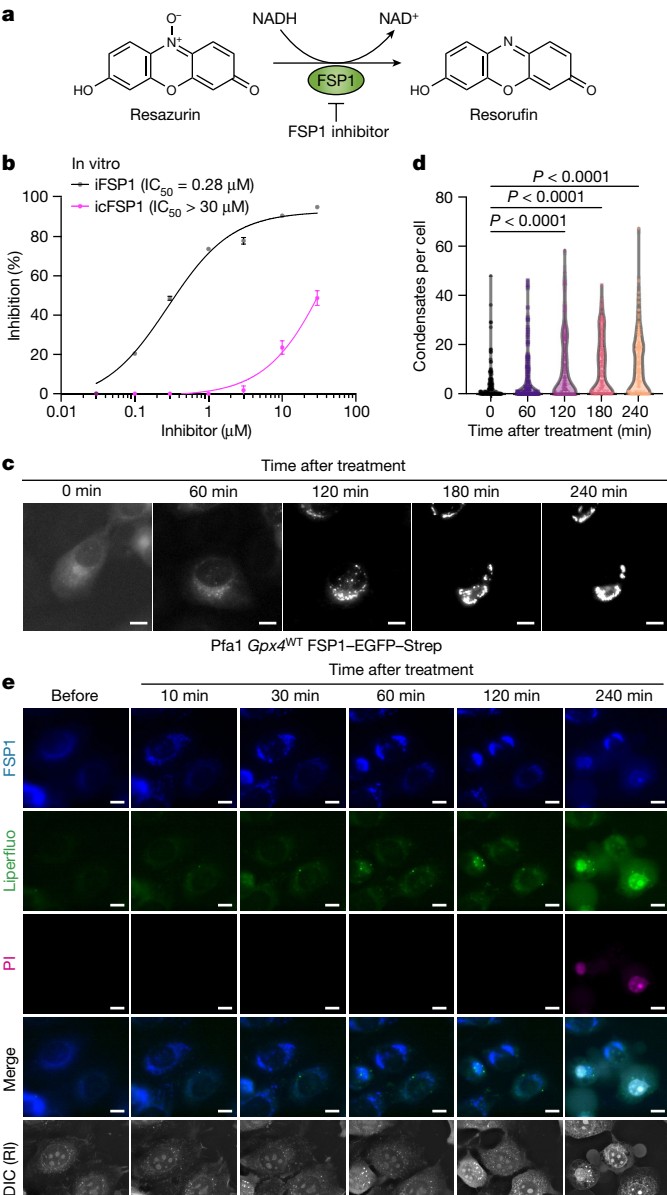

**Fig. 2 | icFSP1 indirectly inhibits FSP1 by inducing condensate formation. a**, Schematic representation of the FSP1 enzyme activity assay using resazurin as the substrate. **b**, Representative dose–response curves for the effect of iFSP1 and icFSP1 on hFSP1 activity using recombinant purified hFSP1 protein. Data represent the mean ± s.d. of 3 wells of a 96-well plate from one of three independent experiments. **c**, Representative time-lapse fluorescence images acquired immediately after treatment of wild-type *Gpx4* Pfa1 cells stably overexpressing hFSP1–EGFP–Strep with 2.5 μM icFSP1. Scale bars, 10 μm. Representative results from one of three independent experiments. See also Supplementary Video 1. **d**, Number of condensates per cell quantified from time-lapse images at different time points (0, 60, 120, 180 and 240 min) after treatment obtained from one of two independent experiments. Dots represent each cell and *n* corresponds to cell number (*n* = 129, 124, 130, 130 and 134 (left to right)). *P* values were calculated by one-way ANOVA followed by Dunnett's multiple-comparison test. **e**, Representative time-lapse fluorescence images before and after treatment of *Gpx4*-knockout Pfa1 cells stably overexpressing hFSP1–mTagBFP with 10 μM icFSP1 in FluoroBrite DMEM containing propidium iodide (PI; 0.2 μg ml⁻¹). Cells were prestained with 5 μM Liperfluo for 1 h. Scale bars, 10 μm. Representative results from three independent experiments. Differential interference contrast (DIC) is displayed with refractive index (RI). See also Supplementary Video 2.

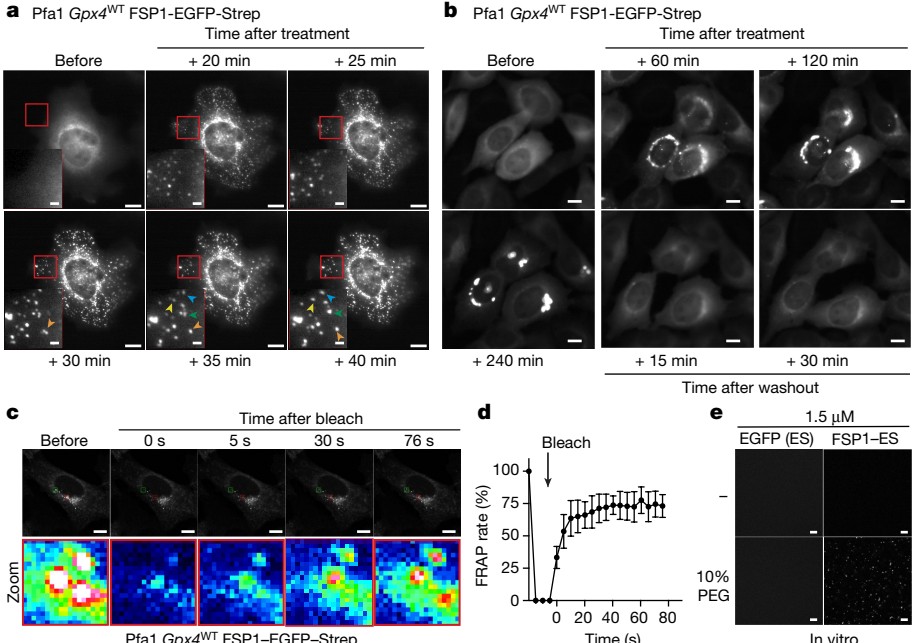

**a** Pfa1 *Gpx4*[WT] FSP1-EGFP-Strep

Time after treatment

Before   + 20 min   + 25 min

+ 30 min   + 35 min   + 40 min

**b** Pfa1 *Gpx4*[WT] FSP1-EGFP-Strep

Time after treatment

Before   + 60 min   + 120 min

+ 240 min   + 15 min   + 30 min

Time after washout

**c**

Time after bleach

Before   0 s   5 s   30 s   76 s

Zoom

Pfa1 *Gpx4*[WT] FSP1–EGFP–Strep

**d**

Bleach

FRAP rate (%)

Time (s)

**e**

1.5 µM

EGFP (ES)   FSP1–ES

–

10% PEG

In vitro

**Fig. 3 | FSP1 condensates are liquid droplets. a**, Representative time-lapse fluorescence images before and after treatment of wild-type *Gpx4* Pfa1 cells stably overexpressing hFSP1–EGFP–Strep with 2.5 µM icFSP1. Representative results are shown from one of three independent experiments. Scale bars, 10 µm (2 µm for zoomed-in images). Arrowheads indicate fusion events of individual condensates. See also Supplementary Video 3. **b**, Reversibility of hFSP1 condensates. Representative time-lapse fluorescence images before and after treatment of wild-type *Gpx4* Pfa1 cells stably overexpressing hFSP1–EGFP–Strep with 2.5 µM icFSP1. After treatment of cells with icFSP1 for 240 min, the medium was replaced with fresh medium without icFSP1 and recordings were restarted. Scale bars, 10 µm. Representative results from one of three independent experiments. See also Supplementary Video 4. **c**, FRAP assays after treatment of hFSP1–EGFP–Strep-overexpressing wlid-type *Gpx4* Pfa1

cells with 2.5 µM icFSP1 for 120 min. Top, greyscale images corresponding to representative FRAP images immediately before and after photobleaching. Bottom, lookup table (LUT) images showing enlarged views of the areas in red rectangles in the top FRAP images. Scale bars, 10 µm. Representative results from one of three independent experiments. See also Supplementary Video 5. **d**, Quantified FRAP rate of each condensate. Data represent the mean ± s.d. of five condensates from **c**. Representative results from one of three independent experiments. **e**, FSP1 condensation in vitro. Representative fluorescence images of 1.5 µM EGFP–Strep and hFSP1–EGFP–Strep purified from transfected HEK29T cells were obtained immediately after mixing with or without 10% PEG. Scale bars, 10 µm. Representative results from one of three independent experiments are shown.

positive for propidium iodide as a measure for cell membrane rupture (Fig. 2e and Supplementary Video 2). These results indicate that changes in the subcellular localization of FSP1 precede lipid peroxidation and ferroptosis.

## icFSP1 induces phase separation of FSP1

To interrogate whether hFSP1 condensates may localize to specific subcellular compartments, hFSP1–EGFP–Strep-expressing cells were co-stained with a number of organelle-specific markers. Reportedly, FSP1 localizes to different subcellular structures, including the endoplasmic reticulum (ER), Golgi apparatus, lipid droplets and perinuclear structures[4,5] (Extended Data Fig. 4a). Yet, treatment of cells with icFSP1 did not induce hFSP1 condensates that clearly colocalized with any of these subcellular structures. Moreover, hFSP1 condensates did not colocalize with other cell organelles, such as endosomes, lysosomes, mitochondria, ubiquitin-dependent aggresomes or stress granules (G3BP1) (Extended Data Fig. 4b,c). Following treatment with icFSP1, these condensates were also detectable in H460 cells (expressing only endogenous hFSP1) and in cells with even lower expression levels than those of endogenous hFSP1 using doxycycline-dependent, scalable expression of hFSP1 (Extended Data Fig. 4d–g). We further noted that hFSP1 condensates dynamically and freely moved around and fused in cells in response to icFSP1 treatment (Fig. 3a, Extended Data Fig. 4h and Supplementary Video 3), which appeared to be reversible after washing out the inhibitor (Fig. 3b and Supplementary Video 4). To investigate the state of condensates in more detail, we established fluorescence

recovery after photobleaching (FRAP) analysis. These studies showed that only early-state condensates exhibited FRAP (Fig. 3c,d and Supplementary Video 5), in contrast to late-state condensates (Extended Data Fig. 5a,b and Supplementary Video 6).

On the basis of these liquid droplet-like properties of hFSP1, we considered that hFSP1 condensates might involve phase separation. Phase separation is a physicochemical process characterized by the reversible formation of biomolecular condensates[7]. These condensates are involved in the regulation of cellular signalling following stress and have been linked to human diseases, including cancer and neurodegeneration[7,23–26]. In particular, phase separation is involved in the partitioning of target proteins in cancer tissue[27] and promotes the formation of aggregates of neurodegenerative disease-related proteins[28]. Thus, we asked whether hFSP1 has the propensity to generate condensates in a cell-free system. To this end, hFSP1–EGFP–Strep was immunoprecipitated from transfected HEK293T cells using the Strep-tag to isolate natively myristoylated hFSP1 from cells. For condensate formation assays, polyethylene glycol (PEG) was used as a molecular crowding agent[29]. Purified hFSP1 was reconstituted with 10% PEG, whereupon hFSP1 immediately formed viscoelastic material[30], in contrast to immunoprecipitated EGFP–Strep controls (Fig. 3e and Extended Data Fig. 5c,d). To investigate whether icFSP1 alone can initiate hFSP1 condensates, purified hFSP1 was reconstituted with PEG and/or icFSP1. However, hFSP1 could form condensates in the presence of PEG regardless of icFSP1. These differences in condensate formation are presumably due to the fact that cell-free and cellular conditions differ greatly and condensates can form viscoelastic material

as shown in cells. To reinforce our finding of hFSP1 condensation in a cell-free system through phase separation, we used recombinant hFSP1 without myristoylation (non-myr-FSP1) purified from *Escherichia coli*. Again, hFSP1 could form condensates in an FSP1 and PEG concentration-dependent manner (Extended Data Fig. 5e–h). Finally, we performed sedimentation assays to investigate whether hFSP1 condensates induced by PEG lead to stable viscoelastic material. Almost all hFSP1 could be recovered from supernatant fractions in pellets in the presence of PEG (Extended Data Fig. 5i,j), suggesting that hFSP1 has the propensity to form condensates induced by phase separation.

## Structural basis of droplet formation

Phase separation can lead to formation of membrane-less compartments in cells, where multivalent interactions, typically involving intrinsically disordered regions (IDRs) and low-complexity regions (LCRs), are known to be critical[7]. Phase separation predictors[31] revealed that hFSP1 contains two putative IDRs and one LCR in its sequence (Extended Data Fig. 6a), which may be required for condensate formation through phase separation. To analyse in detail the role of these predicted domains, the following series of hFSP1 deletion mutants was generated: ΔIDR1, ΔLCR, ΔN (ΔIDR1 and ΔLCR) and ΔIDR2 (Fig. 4a). Two additional mutants were generated in which membrane localization is known to be affected[4,5]: (1) the G2A mutant, which is a myristoylation-defective mutant with strongly affected localization, abrogating the ferroptosis-suppressive function of FSP1 (ref. 5), and (2) the Lyn11–G2A mutant, which contains the membrane-targeting Lyn11 sequence fused N-terminally to FSP1[G2A] and can suppress ferroptosis[4]. After icFSP1 treatment, only wild-type hFSP1 and Lyn11–hFSP1[G2A] changed their subcellular localization to form hFSP1 condensates, and all other mutants did not undergo hFSP1 condensate formation following icFSP1 treatment (Fig. 4b). In line with this, pretreatment of cells with the myristoylation inhibitor IMP-1088 also abrogated hFSP1 condensation in cells (Extended Data Fig. 6b). To investigate whether the hFSP1 deletion mutants might have the propensity to form condensates in a cell-free system, purified hFSP1 mutants from transfected HEK293T cells were reconstituted with 10% PEG. Only wild-type hFSP1 and the G2A and Lyn11–G2A mutants formed condensates in the presence of PEG, whereas the IDR and LCR deletion mutants did not form condensates (Extended Data Fig. 6c). Next, we produced recombinant hFSP1 with myristoylation (myr-FSP1) purified from *E. coli* (Extended Data Fig. 6d) to test whether myristoylation may afford in vitro phase separation induced by PEG, lower salt concentrations and icFSP1 (Extended Data Fig. 6e,f). PEG and lower salt concentrations seemed to facilitate phase separation of myristoylated hFSP1 in vitro, whereas icFSP1 alone did not induce phase separation, suggesting that the cellular context (that is, other binding partners, the membrane environment, post-translational modifications, etc.) is important for phase separation in vitro. Furthermore, phase separation of myristoylated hFSP1 could be induced by icFSP1, whereas non-myristoylated hFSP1[G2A] did not form condensates, even when hFSP1 condensates of the wild-type enzyme were present in Pfa1 cells (Extended Data Fig. 6g). These data imply that the presence of a myristoylation tag facilitates condensate formation, as seen for other myristoylated proteins such as the enhancer of zeste 2 (EZH2) polycomb repressive complex 2 subunit[32]. Moreover, myristoylation may function as a sticker enhancing polyphasic linkage[33,34] for phase separation that is modulated by icFSP1, as a ligand.

To investigate the potential ferroptosis-suppressive functions of these mutants, cell viability experiments were performed using RSL3 and TAM-inducible *Gpx4* knockout (Extended Data Fig. 6h,i). These analyses confirmed that only wild-type hFSP1 and Lyn11–hFSP1[G2A] could efficiently suppress ferroptosis. Hence, these data indicate that the IDRs and LCR may be required for the ferroptosis-suppressive role of FSP1, with icFSP1-induced ferroptosis triggered by hFSP1 condensation through phase separation.

## S187, L217 and Q319 afford condensation

Because icFSP1 was specific for the human enzyme and did not inhibit its mouse counterpart (Extended Data Fig. 2), we addressed the underlying molecular mechanisms that may account for this species-specific inhibitory activity (Extended Data Fig. 7a). The phase separation predictor[31] revealed that mFSP1 also harbours two predicted IDRs and an LCR with some sequence differences at its N terminus. Thus, we decided to generate a chimeric enzyme (that is, hmFSP1) consisting of the first 27 residues of hFSP1 (comprising IDR1 and the LCR) fused to residues 28–373 of mFSP1. However, hmFsp1 did not form condensates (Extended Data Fig. 7b,c), implying that other amino acids contribute to sensitizing cells to icFSP1-mediated condensate formation. On the basis of amino acid differences between the human and mouse orthologues, we generated a series of point mutations in human *FSP1* and stably expressed the corresponding mutants in Pfa1 cells (Extended Data Fig. 7d). These studies allowed us to identify the S187, L217 and Q319 residues of hFSP1 as being critical for icFSP1-dependent condensate formation and ferroptosis induction, as substitution at these sites rendered the mutant proteins resistant (Fig. 4c–e and Extended Data Fig. 7d,e). Given that (1) binding of icFSP1 to FSP1 was not affected by these substitutions (Extended Data Fig. 7f) and (2) reverse substitutions at these three positions to the human residues allowed mFSP1 to form icFSP1-dependent condensates and induce ferroptosis (Extended Data Fig. 7g–i), S187, L217 and Q319 may have critical effects on FSP1–FSP1 interactions to trigger phase separation in cells in the presence of icFSP1 (although the precise binding site of icFSP1 to wild-type hFSP1 remains to be structurally resolved).

## icFSP1 impairs tumour growth in vivo

To evaluate whether icFSP1 might be applicable for in vivo use, metabolic stability and pharmacokinetics analyses were performed, showing that icFSP1 has clearly improved microsomal stability in mice and maximum concentration in mouse plasma as compared with iFSP1 (Extended Data Fig. 8a,b). Furthermore, to evaluate the efficacy of icFSP1 in a tumour-bearing mouse model, *Gpx4* and *Fsp1* double knockout B16F10 cells overexpressing hFSP1 were subcutaneously injected into female C57BL/6J mice. After tumours reached approximately 25–50 mm³ in size, mice were randomized and treated intraperitoneally with vehicle or icFSP1 twice daily. icFSP1 treatment significantly inhibited tumour growth and decreased tumour weight, without affecting body weight (Fig. 4f and Extended Data Fig. 8c). Notably, treatment of tumour-bearing mice with icFSP1 markedly increased the abundance of hFSP1 condensates and immunoreactivity to 4-hydroxynonenal (4-HNE), a lipid peroxidation breakdown product[6] (Fig. 4g and Extended Data Fig. 8d). These data indicate that icFSP1 may trigger phase separation of hFSP1 and thereby impair tumour growth in vivo. To substantiate these findings, we established a *Gpx4* and *Fsp1* double-knockout B16F10 mutant cell line overexpressing hFSP1[Q319K] to study condensate formation in vivo. Like the results obtained with Pfa1 and H460 cells, melanoma cell lines dependent on hFSP1[Q319K] were resistant to icFSP1 in cultured cells and in vivo (Extended Data Fig. 8e–j); in line with this, FSP1 condensates were not observed after icFSP1 treatment in vivo (Fig. 4h and Extended Data Fig. 8h). To investigate whether icFSP1 may also work in the human context, we used a human melanoma cell line (A375) and a human lung cancer cell line (H460), which are known to express substantial levels of FSP1 (ref. 5) and which can survive after withdrawal of radical trapping agents even when *GPX4* is genetically deleted (that is, with *GPX4* knockout)[3,4]. In fact, *GPX4*-knockout cells were highly sensitive to icFSP1 treatment, and the tumour growth of *GPX4*-knockout cells was substantially inhibited in the corresponding xenograft tumour model (Extended Data Fig. 9a–i). In conclusion, our results indicate that the hFSP1-specific inhibitor icFSP1 may trigger phase separation of FSP1 and synergize with canonical ferroptosis

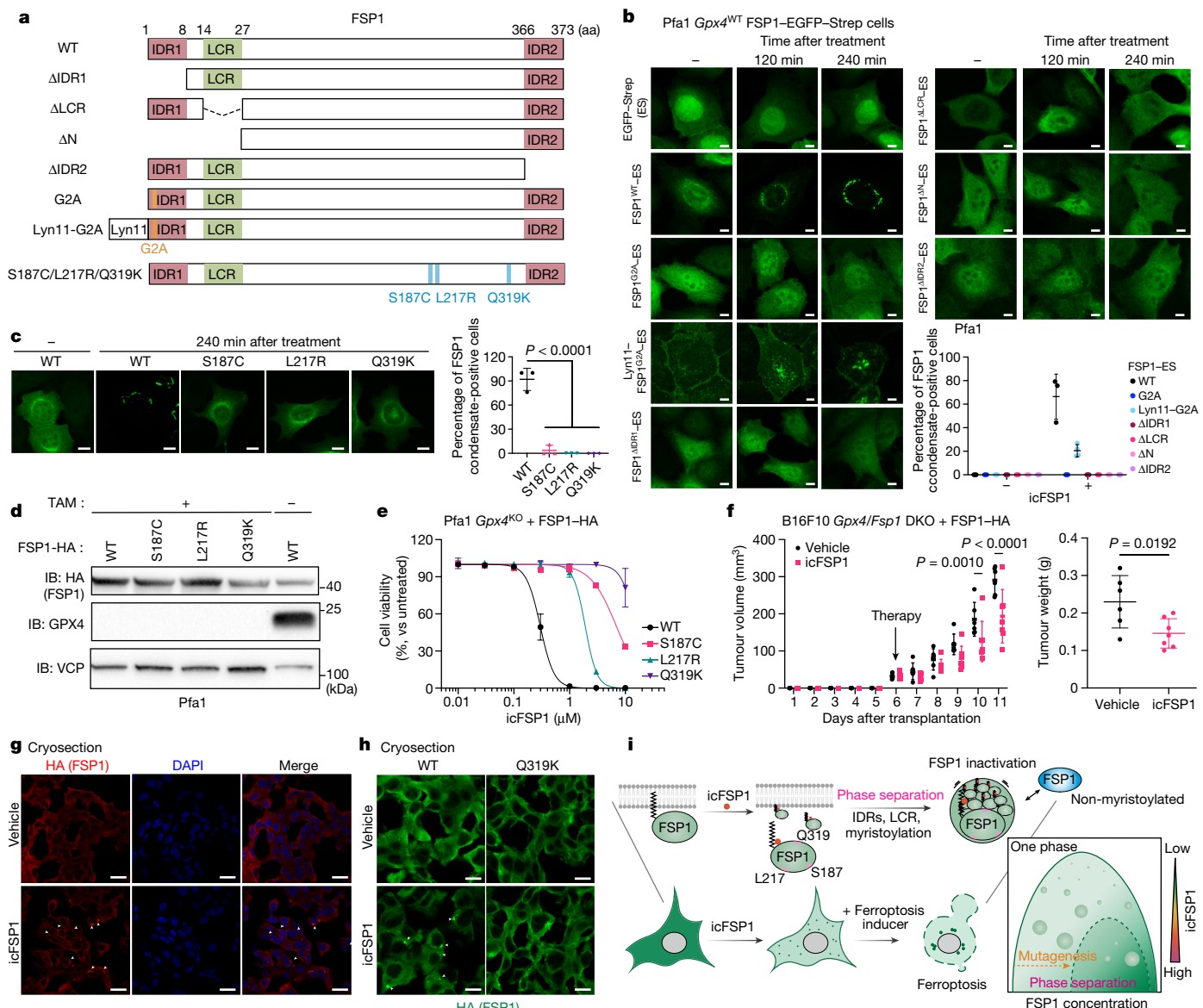

**Fig. 4 | Distinct structural features of FSP1 are required for phase separation.**
**a**, Schematic diagram of the FSP1 mutants. **b**, Representative images of Pfa1 cells overexpressing hFSP1–EGFP–Strep mutants treated with 2.5 μM icFSP1. Scale bars, 10 μm. **c**, Representative images of Pfa1 cells overexpressing wild-type hFSP1–EGFP–Strep or the S187C, L217R or Q319K variant treated with 2.5 μM icFSP1. Scale bars, 10 μm. Data are shown as the mean ± s.d. of n = 3 or 4 different fields from one of three independent experiments (**b**,**c**). Statistical analysis was performed by one-way ANOVA followed by Dunnett's multiple-comparison test (**c**). **d**, Representative immunoblot analysis of Pfa1 cells overexpressing hFSP1–HA from one of two independent experiments. **e**, Cell viability measured after treatment of *Gpx4*-knockout Pfa1 cells overexpressing wild-type hFSP1 or the S187C, L217R or Q319K variant with icFSP1 for 24 h. Data represent the mean ± s.d. of n = 3 wells from one of four independent experiments. **f**, icFSP1 inhibits tumour growth in vivo. hFSP1–HA-overexpressing

*Gpx4* and *Fsp1* double knockout (*Gpx4*/*Fsp1* DKO) B16F10 cells were subcutaneously implanted into C57BL/6J mice. At day 6, mice were randomized and treatment was started with icFSP1 (50 mg kg⁻¹ intraperitoneally twice a day, n = 7) or vehicle (n = 6). Data represent the mean ± s.d. from one of two independent experiments. Statistical analysis was performed by two-way ANOVA followed by Bonferroni's multiple-comparison test (left) or two-sided unpaired t test (right). **g**, Tumour samples from the end of the in vivo studies stained with anti-HA (hFSP1). **h**, Wild-type or Q319K hFSP1 tumour samples visualized with HA immunostaining. Representative zoomed-in images from Extended Data Fig. 8h are shown from one of three different tumour samples from one of two independent experiments (**g**,**h**). Arrowheads indicate FSP1 condensates (**g**,**h**). Scale bars, 20 μm (**g**) and 10 μm (**h**). **i**, Graphical abstract depicting icFSP1-induced FSP1 condensate formation, lipid peroxidation and ferroptosis. Image created using BioRender.com.

inhibitors to induce ferroptosis, as a viable way for efficient eradication of certain cancer entities.

## Discussion

Here we report on a yet-unrecognized class of in vivo-efficacious hFSP1 inhibitors, that is, 3-phenylquinazolinones, that exhibit a unique mechanism of action in which they trigger ferroptosis through dissociation of

FSP1 from the membrane and formation of FSP1 condensates involving phase separation (Fig. 4i). Our experiments show that icFSP1-mediated phase separation requires several molecular features in hFSP1, such as N-terminal myristoylation, specific amino acid residues (S187, L217 and Q319), and the IDRs and LCR, resulting in lipid peroxidation and ferroptosis under GPX4-inhibited conditions. In the absence of an experimentally determined three-dimensional structure of FSP1, we mapped the amino acid residues that contribute to icFSP1-induced

phase separation of FSP1, on the basis of our mutational analysis, onto the structure predicted by AlphaFold2 (refs. 35,36). Although the globular fold prediction had a high-confidence IDR1 region, it was annotated as an uncertain region, in contrast to the LCR, which was predicted to exhibit an α-helical conformation with an intermediate score. The role of the LCR needs to be experimentally studied, to examine how this region may contribute to phase separation and potentially interact with the globular domain of FSP1. These interactions may well be modulated by icFSP1 and should help in understanding the underlying structural mechanisms. Moreover, considering that icFSP1 initiates formation of FSP1 condensates, including of plasma membrane-localized mutants, immediately after treatment and that icFSP1 by itself cannot trigger condensation of myristoylated FSP1 in vitro, icFSP1 may induce condensates by modulating FSP1–membrane interactions and thereby reducing the membrane-binding affinity of FSP1, similar to the KRAS inhibitor Cmpd2 (ref. 37) and other modulators (for example, $Ca^{2+}$ for recoverin[38]) of myristoyl–ligand switches. In support of our hypothesis, the three-dimensional structural model of FSP1 (refs. 35,36) showed that residues S187, L217 and Q319 are all located on the surface of FSP1, with Q319 in particular close to the expected membrane-binding surface. Considering the known relevance of charged and polar residues for protein–protein interactions during phase separation[39], changing S187 to cysteine, L217 to arginine and Q319 to lysine should increase the positive surface charge or reduce polarity and may thereby impair phase separation in cells (Extended Data Fig. 7i).

The concept in which ligands modulate the driving forces for phase separation is known as polyphasic linkage[33,34]. Conceptually, icFSP1 would modulate the phase transition of FSP1, probably through interactions that directly or indirectly involve residues such as S187, L217 and Q319, and disruption of interactions by mutagenesis changes the phase boundary. In particular, myristoylation appears to be indispensable for this process in which icFSP1, as a ligand, preferentially binds to the myristoylated form of FSP1 (Extended Data Fig. 6g).

Although inhibition of FSP1 alone is generally not sufficient to drive cancer cell death through ferroptosis[5,19], a subset of cancer cell lines can potentially be sensitive to FSP1 inhibition alone under certain conditions, as was observed for HT-1080 cells. In this respect, database analysis might be helpful to predict the sensitivity of cancer cell lines to FSP1 inhibition (Extended Data Fig. 10). In light of the fact that *Fsp1*-knockout mice are fully viable[6] and that icFSP1 does not show any observed off-target activity and does not affect body weight even at high concentrations (Extended Data Figs. 1 and 8), FSP1 should be regarded as an attractive target for tumour treatment. Thus, future studies should be geared to developing pharmacological approaches that simultaneously target both the cyst(e)ine–GSH–GPX4 node and the FSP1 system, allowing for efficient tumour cell eradication by triggering ferroptosis as a new anti-cancer paradigm.

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

## Methods

### Chemicals

Lip-1 (Selleckchem, cat. no. S7699), doxycycline (Dox; Sigma, cat. no. D9891), RSL3 (Cayman, cat. no. 19288), BSO (Sigma, cat. no. B2515), iFSP1 (ChemDiv, cat. no. 8009-2626), icFSP1 (ChemDiv, cat. no. L892-0224 or custom synthesis by Intonation Research Laboratories), erastin (Merck, cat. no. 329600), ML210 (Cayman, cat. no. Cay23282-1), FIN56 (Cayman, cat. no. Cay25180-5), FINO2 (Cayman, cat. no. Cay25096-1), deferoxamine mesylate salt (DFO; Sigma, cat. no. 138-14-7), ferrostatin-1 (Fer-1; Sigma, cat. no. SML0583), zVAD-FMK (zVAD; Enzo Life Sciences, cat. no. ALX-260-02), necrostatin-1s (Nec-1s; Enzo Life Sciences, cat. no. BV-2263-5), MCC950 (Sigma, cat. no. 5381200001), olaparib (Selleck-chem, cat. no. S1060), staurosporine (STS; Cayman, cat. no. 81590), recombinant human tumour necrosis factor (TNF; R&D Systems, cat. no. NBP2-35076), Smac mimic (BV-6; Selleckchem, cat. no. S7597), nigericin (Thermo Fisher, cat. no. N1495), IMP-1088 (Cayman, cat. no. Cay25366-1) and lipopolysaccharide (LPS; Sigma, cat. no. L2880) were used in this study.

### Mice

Five- to six-week-old female C57BL/6J and athymic nude mice were obtained from Charles River, and 6- to 7-week-old mice were used for the experiments. All mice were kept under standard conditions with water and food provided ad libitum and in a controlled environment ($22 \pm 2$ °C, $55\% \pm 5\%$ humidity, 12-h light/12-h dark cycle) in the Helmholtz Munich animal facility under SPF-IVC standard conditions. All experiments were performed in compliance with the German Animal Welfare Law and were approved by the institutional committee on animal experimentation and the government of Upper Bavaria (ROB-55.2-2532.Vet_02-17-167).

### Cell lines

TAM-inducible *Gpx4*-knockout mouse immortalized fibroblasts (referred to as Pfa1 cells) were reported previously[18]. Genomic *Gpx4* deletion can be achieved by TAM-inducible activation of Cre recombinase using the CreER[T2]/*loxP* system. HT-1080, 786-O, A375, NCI-H460 (H460), MDA-MB-436, HT-29, B16F10, and 4T1 cells were purchased from ATCC. THP-1 cells were obtained from DSMZ). Human PBMC cells were purchased from Tebu-bio (cat. no. 088SER-PBMC-F). SUDHL5, SUDHL6, DOHH2 and OCI-Ly19 cells were a gift from S. Hailfinger. Rat1 cells were a gift from Medizinische Hochschule Hannover. Pfa1, HT-1080, 786-O, A375, HT-29, Rat1 and B16F10 cells were cultured in high-glucose DMEM ($4.5$ g $l^{-1}$ glucose) with 10% FBS, 2 mM L-glutamine and 1% penicillin-streptomycin. H460, MDA-MB-436, THP-1, PBMC, SUDHL5, SUDHL6, DOHH2 and 4T1 cells were cultured in RPMI GlutaMax with 10% FBS and 1% penicillin-streptomycin. OCI-Ly19 cells were cultured in IMDM with 10% FBS and 1% penicillin-streptomycin. To generate cell lines with stable overexpression, appropriate antibiotics ($1$ µg $ml^{-1}$ puromycin, $10$ µg $ml^{-1}$ blasticidin and $0.5$–$1.0$ mg $ml^{-1}$ G418) were used. *GPX4*-knockout human A375 and H460 cells were cultured in the presence of $1$ µM Lip-1 for maintenance. All cells were cultured at 37 °C with 5% $CO_2$ and verified to be negative for mycoplasma.

### Cell viability assays

Cells were seeded on 96-well plates and cultured overnight. All cell number conditions for each cell line are described in Supplementary Table 1. The next day, the medium was changed to medium containing the following compounds: RSL3, ML210, erastin, FIN56, FINO2, BSO, iFSP1, icFSP1, Lip-1, DFO, Fer-1, zVAD, Nec-1s, MCC950, olaparib, STS, TNF, Smac mimic or nigericin at the indicated concentrations. For TAM and Dox treatment, cells were seeded with compounds at the same time. Cell viability was determined 1 h (for nigericin), 24–48 h (for RSL3, ML210, erastin, FIN56, FINO2, iFSP1, icFSP1, STS, TNF, Smac mimic and zVAD) or 72 h (for BSO, icFSP1, TAM and Dox) after the start of treatment using AquaBluer (MultiTarget Pharmaceuticals, cat. no. 6015) as an indicator of viable cells according to the manufacturer's protocol. For apoptosis induction, HT-1080 cells were incubated with different concentrations of STS for 24 h. For necroptosis induction, HT-29 cells were incubated with different concentrations of TNF with Smac mimic (400 nM) and zVAD (30 µM) for 24 h. For pyroptosis induction, THP-1 cells stimulated with LPS ($1$ µg $ml^{-1}$, 2 h) were incubated with nigericin for 1 h. For ferroptosis induction, cells were incubated with ferroptosis inducers for 24–72 h.

As readout, fluorescence was measured at an excitation/emission wavelength of 540/590 nm using a SpectraMax M5 microplate reader with SoftMax Pro v.7 (Molecular Devices) after 4 h of incubation with AquaBluer in normal cell culture conditions. The relative cell viability (%) was calculated as follows: (fluorescence of samples − background)/(fluorescence of appropriate control samples − background) × 100.

### LDH release assays

Cells were seeded on 96-well plates and cultured overnight. The next day, the medium was changed to medium containing compounds and cells were incubated for another 24 h. Cell death rates were measured using the Cytotoxicity Detection Kit (LDH) (Roche, cat. no. 11644793001) in principle following the manufacturer's protocol. In brief, cell culture supernatant was collected as a medium sample. Cells were then lysed with PBS containing 0.1% Triton X-100 as a lysate sample. The medium and lysate samples were individually mixed with reagents on microplates, and the absorbance was measured at 492 nm using a SpectraMax M5 microplate reader after incubation for 15–30 min at room temperature. The cell death ratio was calculated by LDH release (%) as follows: (absorbance of medium samples − background)/((absorbance of lysate samples − background) + (absorbance of medium samples − background)) × 100.

### Screening of FSP1 inhibitors

Wild-type and *Gpx4*-knockout Pfa1 cells stably overexpressing hFSP1 were seeded on separate 384-well plates (500 cells per well) and screened with a library of small molecule inhibitor compounds as reported previously[5]. Cell viability of the different cell lines was assessed 48 h after the start of treatment using AquaBluer. Compounds showing selective lethality in *Gpx4*-knockout Pfa1 cells stably overexpressing hFSP1 were then validated in cell viability assays and in vitro FSP1 enzymatic assays.

### Lipid peroxidation assays

For lipid peroxidation assays, 100,000 cells per well were seeded on a 12-well plate 1 day before the experiments. The next day, cells were treated with 2.5 µM icFSP1 for 3 h and then incubated with 1.5 µM C11-BODIPY 581/591 (Invitrogen, cat. no. D3861) for 30 min in a 5% $CO_2$ atmosphere at 37 °C. Subsequently, cells were washed once with PBS, trypsinized and then resuspended in 500 µl PBS. Cells were passed through a 40-µm cell strainer and analysed with a flow cytometer (Cyto-FLEX, Beckman Coulter) with a 488-nm laser for excitation. Data were collected from the FITC detector (for the oxidized form of BODIPY) with a 525/40-nm bandpass filter and from the PE detector (for the reduced form of BODIPY) with a 585/42-nm bandpass filter using CytExpert v.2.4 (Beckman Coulter). At least 10,000 events were analysed per sample. Data were analysed using FlowJo software (FlowJo). The ratio of fluorescence of C11-BODIPY 581/591 (lipid peroxidation) (FITC/PE ratio (oxidized/reduced ratio)) was calculated as follows[12]: (median FITC-A fluorescence − median FITC-A fluorescence of unstained samples)/(median PE-A fluorescence − median PE-A fluorescence of unstained samples). An example gating strategy is shown in Supplementary Fig. 1.

### Oxilipidomics analysis

Two million cells were seeded on 15-cm dishes 1 day before the experiments. The next day, cells were treated with 5 µM icFSP1 to induce lipid

peroxidation. Five hours later, cells were collected, sampled to liquid nitrogen and stored at −80 °C. Lipids from cells were extracted using the methyl-*tert*-butyl ether (MTBE) method[40]. In brief, cell pellets collected in PBS containing dibutylhydroxytoluene (BHT; 100 μM) and diethylenetriamine pentaacetate (DTPA; 100 μM) were washed and centrifuged. SPLASH LIPIDOMIX (Avanti Polar Lipids) was added (2.5 μl), and samples were incubated on ice for 15 min. After addition of ice-cold methanol (375 μl) and MTBE (1,250 μl), samples were vortexed and incubated for 1 h at 4 °C (orbital shaker, 32 rpm). Phase separation was induced by adding water (375 μl), and samples were vortexed, incubated for 10 min at 4 °C (orbital shaker, 32 rpm) and centrifuged to separate the organic and aqueous phases (10 min, 4 °C, 1,500$g$). The organic phase was collected, dried in a vacuum evaporator and redissolved in 100 μl isopropanol. Lipid extracts were transferred to glass vials for LC–MS analysis.

Reversed-phase LC was carried out on a Shimadzu ExionLC equipped with an Accucore C30 column (150 × 2.1 mm, 2.6 μm, 150 Å; Thermo Fisher Scientific). Lipids were separated by gradient elution with solvent A (1:1 (v/v) acetonitrile/water) and solvent B (85:15:5 (v/v/v) isopropanol/acetonitrile/water), both containing 5 mM $NH_4HCO_2$ and 0.1% (v/v) formic acid. Separation was performed at 50 °C with a flow rate of 0.3 ml min$^{-1}$ using the following gradient: 0–20 min, increase from 10% to 86% B (curve 4); 20–22 min, increase from 86% to 95% B (curve 5); 22–26 min, 95% isocratic; 26–26.1 min, decrease from 95% to 10% B (curve 5), which was followed by re-equilibration for 5 min at 10% B[6]. MS analysis was performed on a Sciex 7500 system equipped with an electrospray ionization (ESI) source and operated in negative-ion mode. Products were analysed in MRM mode monitoring transitions from the parent ion to the daughter ion, as described in Supplementary Table 1, with the following parameters: TEM, 500 °C; GS1, 40; GS2, 70; CUR, 40; CAD, 9; IS, −3,000 V.

The area under the curve for the parent to daughter ion transition was integrated and normalized by appropriate lipid species, PC(15:0/18:1(d7)) or PE(15:0/18:1(d7)), from the SPLASH LIPIDOMIX Mass Spec Standard (Avanti Polar Lipids). Normalized peak areas were further log transformed and auto-scaled in MetaboAnalyst online platform v.5.0 (https://www.metaboanalyst.ca)[41]. Zero values were replaced by 0.2 times the minimum value detected for a given oxidized lipid in the samples. Oxidized lipids showing a significant difference (ANOVA, adjusted $P$ value (false discovery rate (FDR)) cut-off of 0.05) between samples were used for the heat maps. The heat maps were created in GraphPad Prism 9. The colour scheme corresponds to auto-scaled log-transformed fold change relative to the mean log value for the samples.

### Cell lysis and immunoblotting

Cells were lysed in LCW lysis buffer (0.5% Triton X-100, 0.5% sodium deoxycholate salt, 150 mM NaCl, 20 mM Tris-HCl, 10 mM EDTA and 30 mM sodium pyrophosphate tetrabasic decahydrate) supplemented with protease and phosphatase inhibitor cocktail (cOmplete and phoSTOP; Roche, cat. nos. 04693116001 and 4906837001) and centrifuged at 20,000$g$ for 1 h at 4 °C. The supernatant was sampled by adding 6× SDS sample buffer (375 mM Tris-HCl pH 6.8, 9% SDS, 50% glycerol, 9% β-mercaptoethanol and 0.03% bromophenol blue). After heating at 98 °C (55 °C for xCT) for 3 min, the samples were resolved on 12% SDS–PAGE gels (Bio-Rad, cat. no. 4568043 or 4568046) and subsequently electroblotted onto PVDF membrane (Bio-Rad, cat. no. 170-4156). The membranes were blocked with 5% skim milk (Roth, cat. no. T145.2) in TBS-T (20 mM Tris-HCl, 150 mM NaCl and 0.1% Tween-20) and then probed with primary antibodies, diluted in first antibody dilution buffer (TBS-T with 5% BSA and 0.1% NaN$_3$ (Sigma, cat. no. S2002)), against GPX4 (1:1,000; Abcam, cat. no. ab125066), valosin-containing protein (VCP; 1:1,0000; Abcam, cat. no. ab11433 or ab109240), Flag tag (1:5,000; Cell Signaling Technology, cat. no. 2368), HA tag (1:1,000; clone 3F10, homemade), hFSP1 (1:1,000;

Santa Cruz, cat. no. sc-377120, AMID), mFSP1 (1:500; clone AIFM21A1, rat IgG2a), hFSP1 (1:10; clone 6D8, rat IgG2a), mFSP1 (1:5; clone AIFM2 14D7, rat IgG2b), human SLC7A11 (1:10; rat IgG2a monoclonal antibody against an N-terminal peptide of human xCT, clone 3A12-1-1, developed in house), mouse SLC7A11 (1:1,000; Cell Signaling Technology, cat. no. 98051), ACSL4 (1:1,000; clone A-5, Santa Cruz, cat. no. sc-271800) or β-actin-HRP (1:50,000; Sigma, cat. no. A3854) diluted in 5% skim milk in TBS-T overnight. After membranes were washed and probed with appropriate secondary antibodies diluted in 5% skim milk in TBS-T, antibody–antigen complexes were detected with the ChemiDoc Imaging System with Image Lab v.6.0 (Bio-Rad). Representative images are shown after adjustment to the appropriate brightness and angle using ImageJ/Fiji software (v.1.52 and v.1.53).

### Expression and sgRNA plasmid construction

All plasmids for this study were constructed using standard molecular biology techniques and verified by sequencing as follows. A human *FSP1* cDNA (NM_001198696.2, 1008:C>T) was cloned from previously reported vectors[5]. Codon-optimized sequences for *Mus musculus* (mouse) FSP1 (NP_001034283.1), *Rattus norvegicus* (rat) FSP1 (NP_001132955.1), *Gallus gallus* (chicken) FSP1 (XP_421597.1) and *Xenopus laevis* (frog) FSP1 (NP_001091397.1) were cloned into the p442 vector. Codon-optimized sequences for human *FSP1* (NP_001185625.1) and mouse *Fsp1* (NP_001034283.1) were synthesized by TWIST Bioscience and subcloned into 141-IRES-puro vector. To generate deletion mutants or perform subcloning, desired DNA sequences were first amplified using KOD One PCR master mix (Sigma, cat. no. KMM-201NV) or PrimeSTAR Max DNA polymerase master mix (Takara Bio, cat. no. R045A) and resulting PCR products were purified by Wizard SV Gel&PCR Clean-up System (Promega, cat. no. A9285) according to the manufacturer's protocol. Ligation reactions of PCR products or single guide RNA (sgRNA) duplexes with digested vectors were performed using T4 ligase (NEB, cat. no. M0202L) or In-Fusion cloning enzymes (Takara Bio, cat. no. 639649 or 638948) according to the manufacturer's protocol. Subsequently, reaction mixtures were transformed into stable competent cells (NEB, cat. no. C3040H). Plasmids were isolated using the QIAprep Spin Miniprep kit (Qiagen, cat. no. 27106) according to the manufacturer's protocol; the correct inserts of plasmids were confirmed by sequencing.

### Lentiviral production and transduction

HEK293T cells were used to produce lentiviral particles. The ecotropic envelope protein of murine leukaemia virus (MLV) was used for mouse-derived cells, while the amphitropic envelope protein VSV-G was used for human-derived cells. A third-generation lentiviral packaging system consisting of transfer plasmids, envelope plasmids (pEcoEnv-IRES-puro or pHCMV-EcoEnv (ecotropic particles) or pMD2.G (pantropic particles)) and packaging plasmids (pMDLg_pRRE and pRSV_Rev or psPAX2) was co-lipofected into HEK293T cells using transfection reagent (PEI MAX (Polysciences, cat. no. 24765) or X-tremeGENE HP reagent (Roche, cat. no. 06366236001)). Viral particle-containing cell culture supernatant was collected 48–72 h after transfection, filtered through a 0.45-μm PVDF filter (Millipore, cat. no. SLHV033RS) and then used for lentiviral transduction.

Cells were seeded on 12- or 6-well plates in medium supplemented with 10 μg ml$^{-1}$ protamine sulfate and lentivirus was incubated with cells overnight. The next day, the cell culture medium was replaced with fresh medium containing appropriate antibiotics, such as puromycin (Gibco, cat. no. A11138-03; 1 μg ml$^{-1}$), blasticidin (Invitrogen, cat. no. A1113903; 10 μg ml$^{-1}$) or G418 (Invitrogen, cat. no. 10131-035; 1 mg ml$^{-1}$)) and cells were cultured until non-transduced cells were dead.

### CRISPR–Cas9-mediated gene knockout

sgRNAs were designed to target critical exons of the genes of interest, and gene knockout was confirmed by western blotting. sgRNAs were

cloned into BsmBI-digested lentiCRISPRv2-blast, lentiCRISPRv2-puro and lentiGuide-neo vectors (Addgene, cat. nos. 98293, 98290 and 139449). All sequences for sgRNAs are listed in Supplementary Table 1.

To generate knockout cells, MDA-MB-436, 786-O, A375, H460, B16F10 and 4T1 cells were transiently co-transfected with the desired sgRNAs expressed from lentiCRISPRv2-blast and lentiCRISPRv2-puro using X-tremeGENE HP reagent as described previously[6,42]. One day after transfection, selection was started with puromycin (1 µg ml[−1]) and blasticidin (10 µg ml[−1]). After selection for 2–3 days, single-cell clones were isolated, and knockout clones were validated by immunoblotting and sequencing of genomic DNA.

To generate Dox-inducible knockout cells, lentiviruses from pCW-Cas9-blast (Addgene, cat. no. 83481) were transduced into HT-1080 cells followed by selection and establishment of single-cell clones as described previously[6]. Lentiviruses generated from lentiGuide-neo vectors harbouring sgRNAs targeting FSP1 were transduced into HT-1080 pCW-Cas9-blast cells, followed by selection with G418 as above. After Dox induction, loss of FSP1 expression was confirmed by immunoblotting.

To generate Dox-inducible FSP1–EGFP-expressing cells, H460 *FSP1*-knockout cells were transduced with lentivirus (pCW-FSP1[WT]-EGFP-blast or pCW-FSP1[Q319K]-EGFP-blast). After Dox treatment of cells, scalable FSP1 expression was confirmed by immunoblotting.

## Stable expression by transfection

*Gpx4*-knockout 4T1 cells and *Gpx4* and *Fsp1* double-knockout B16F10 cells were transfected with 141-IRES-puro, 141-hFSP1[WT] or hFSP1[Q319K]-IRES-puro and 141-mFsp1-IRES-puro vectors using X-tremeGENE HP reagent. One day after transfection, selection was started with puromycin (1 µg ml[−1]) and Lip-1 was removed from the medium to select cells with stable FSP1 expression. To obtain cells with stable expression, cells were maintained under the selection condition.

## Production and purification of FSP1 enzyme

Recombinant hFSP1 protein (non-myr-FSP1) was produced in BL21 *E. coli* and purified by affinity chromatography with a Ni-NTA system as described previously[5].

For protein isolation by pulldown, HEK293T cells were seeded on 10- or 15-cm dishes and transfected with constructs encoding EGFP–Strep-tagged protein. After washing with PBS, cells were lysed in LCW lysis buffer supplemented with protease and phosphatase inhibitor cocktail and 1 mM dithiothreitol (DTT). Cell extracts were collected with a cell scraper and centrifuged at 20,000*g* for 1 h at 4 °C. Supernatants were incubated with MagStrep 'type3' XT beads (Biozol, cat. no. 2-4090-002) at 4 °C for 1–2 h. Beads were washed twice with washing buffer (100 mM Tris-HCl, 150 mM NaCl and 1 mM EDTA). EGFP–Strep-tagged proteins were eluted with elution buffer (100 mM Tris-HCl, 150 mM NaCl, 1 mM EDTA and 50 mM biotin), followed by dilution to 3 µM with TBS (50 mM Tris-HCl and 150 mM NaCl). Protein concentration was estimated from the absorbance at 280 nm measured with a UV5Nano spectrophotometer (Mettler Toledo). The coefficient was calculated by using ExPASy ProtParam (https://web.expasy.org/protparam/).

Myristoylated protein was obtained by coexpressing *N*-myristoyltransferase 1 (hsNMT1) with FSP1. *E. coli* BL21 cells were transformed with constructs encoding hsNMT1 (petCDF vector, spectinomycin resistance) and FSP1 (FSP1–EGFP with a C-terminal His₆ tag, petM13, kanamycin resistance). Purification was performed as for wild-type FSP1 with a final step of gel filtration chromatography. Purified protein was obtained and confirmed by denaturing SDS gel. Confirmation of the presence of myristoylated protein was obtained with MS.

## Mass spectrometry

Myristoylation was confirmed by LC–ESI–MS (Waters Synapt XS). Proteins were separated on an Acquity UPLC Protein BEH C4 column

(0.4 ml min[−1]; buffer A, 0.1% formic acid in water; buffer B, 0.1% formic acid in acetonitrile), and data were analysed using Masslynx v.4.2 (Waters).

## FSP1 enzyme activity and inhibition assays

For resazurin assays, enzyme reactions were prepared in TBS (50 mM Tris-HCl and 150 mM NaCl) containing 50 nM non-myr-FSP1, 200 µM NADH and inhibitor (iFSP1 or icFSP1). After the addition of 100 µM resazurin sodium salt (Sigma, cat. no. R7017), fluorescence intensity (*F*; excitation/emission wavelength of 540/590 nm) was measured every 1 min using a SpectraMax iD5 microplate reader with SoftMax Pro v.7 (Molecular Devices) at 37 °C. Reactions with an equivalent volume of DMSO and without resazurin were used to calculate IC₅₀ values. Curve fitting and calculation of IC₅₀ values were conducted using GraphPad Prism 9.

For NADH consumption assays, enzyme reactions were prepared in PBS (Gibco, cat. no. 14190094) containing 25 nM non-myr-FSP1, 200 µM menadione (Sigma, cat. no. M5625) or 200 µM CoQ₀ (Sigma, cat. no. D9150), and inhibitor (iFSP1 or icFSP1). After the addition of 200 µM NADH, absorbance at 340 nm at 37 °C was measured every 30 s using a SpectraMax M5 microplate reader (Molecular Devices). Reactions without NADH and without enzyme were used to normalize the results. Curve fitting was conducted using GraphPad Prism 9.

## In vitro FSP1 condensation assays

Purified EGFP–Strep or hFSP1–EGFP–Strep tagged protein was diluted in TBS supplemented with 1 mM DTT. Purified Strep-tagged proteins were then mixed with PEG 3350 (Sigma, cat. no. P3640) and/or icFSP1; final concentrations of the proteins, PEG and icFSP1 are indicated in each figure legend. For confocal microscopy analysis, samples were immediately transferred onto objective slides and EGFP signal was quickly captured using an LSM880 microscope with Zen Black software (v.2.3, ZWISS) with a ×63 water-immersion objective. For confocal microscopy analysis, recombinant C-terminally GFP-tagged FSP1 and myr-FSP1 were measured at a 15 µM protein concentration in PBS (pH 7.4; 300 mM NaCl) or at a 10 µM protein concentration in PBS (pH 7.4; 150 mM NaCl). Confocal fluorescence microscopy was performed at 255 °C on a Leica TCS SP8 confocal microscope using a ×63 water-immersion objective. Samples were excited with a 488-nm laser (GFP) and imaged at 498–545 nm.

To measure turbidity, different concentrations of non-myr-FSP1 and PEG were reconstituted in 10 µl in a 384-well plate and the absorbance at 600 nm was measured using a SpectraMax iD5 microplate reader (Molecular Devices). To show non-myr-FSP1 condensation in PCR tubes, images were acquired using a smartphone. Representative bright-field images of non-myr-FSP1 condensates on an objective slide were captured using an Eclipse Ts2 microscope (Nikon) with a ×40 objective.

For sedimentation assays, recombinant non-myr-FSP1 was mixed with the same amount of TBS with 0% or 20% PEG and 1 mM DTT and samples were centrifuged at 2,500*g* for 5 min. The supernatant was collected in a new tube and the pellet was resuspended in TBS supplemented with 1 mM DTT. Supernatant and resuspended non-myr-FSP1 were collected by adding 6× SDS sample buffer and subsequently resolved by SDS–PAGE. One gel was subjected to western blotting and probed with anti-FSP1 (1:1,000; Santa Cruz, cat. no. sc-377120, AMID). The other gel was immediately stained with Coomassie staining solution (1 mg ml[−1] Coomassie Brilliant Blue G-250 (Sigma, cat. no. 1154440025), 50% methanol and 10% acetic acid) for 15 min and then soaked in washing buffer (70% methanol and 7% acetic acid). The washing buffer was heated using a microwave, and the buffer was changed until protein bands gave clear signals.

## In vitro saturation transfer difference experiments

Saturation transfer difference experiments were performed on a Bruker Avance III HD spectrometer at 600-MHz [1]H frequency using an H/N/C

triple-resonance cryogenic probe. Spectra were recorded at 10 °C with 5 μM recombinant hFSP1 (mutant) and 100-fold molar excess of icFSP1 in PBS containing 150 mM NaCl, 1% (v/v) DMSO-d6 and 10% (v/v) $D_2O$ for deuterium-lock. The saturation time was 2.5 s, and the on and off frequencies were 0.68 and −17 ppm, respectively. NMR spectra were processed using Topspin 4.0.6 (Bruker).

## Immunocytochemistry

All confocal microscopy images were acquired using an LSM880 microscope (Zeiss) with a ×63 objective and the corresponding appropriate filter sets for fluorophores and analysed with Zen Blue software (v.3.2, ZWISS) or ImageJ/Fiji unless noted otherwise. Cells were seeded on μ-Slide 8-well slides (Ibidi, cat. no. 80826) 1 day before the experiments. The next day, the medium was changed to fresh cell culture medium supplemented with 2.5 μM icFSP1. After incubation for the indicated times, cells were fixed and stained according to the following procedure: fixation with 4% paraformaldehyde for 5–10 min; permeabilization and blocking for 15 min with 0.3% Triton X-100 and 10 mg ml⁻¹ BSA in PBS; and incubation at 4 °C overnight with primary antibodies or undiluted supernatant for anti-AIFM2 (FSP1; clone 14D7, homemade). Antibody dilutions were as follows: 1:10 for anti-YPYDVPDYA-tag (HA; clone 3F10) and 1:100 for anti-calnexin (Abcam, cat. no. ab22595), anti-GM130 (clone EP892Y, Abcam, cat. no. ab52649), anti-EEA1 (clone C45B10, Cell Signaling Technology, cat. no. 3288) and anti-LAMP1 (clone H4A3, Santa Cruz, cat. no. sc-20011)) in primary antibody dilution buffer. Cells were further stained with appropriate fluorophore-conjugated secondary antibodies (1:500 dilution) and DAPI (1:10,000 dilution) in TBS-T or PBS for 1–2 h at room temperature, avoiding light. Finally, all samples were mounted in Aqua-/PolyMount (Polysciences, cat. no. 18606-20) and dried at 4 °C overnight. Staining of mitochondria, lipid droplets and aggresomes was performed using MitoTracker Red CMXRos (20 nM; Invitrogen, cat. no. M7512), LipidSpot 610 (1:1,000; Biotium, cat. no. 70069-T) and the Proteostat detection kit (1:10,000; Enzo, cat. no. ENZ-51035-0025), respectively, according to the manufacturer's protocol.

## FRAP

Pfa1 cells (20,000 cells) were seeded on μ-Slide VI 0.4 slides (Ibidi, cat. no. 80606) 1 day before the experiments. The next day, the medium was changed to high-glucose DMEM supplemented with 10% FBS, 2 mM L-glutamine, 1% penicillin-streptomycin, 2.5 μM icFSP1 and 10 mM HEPES. After incubation with icFSP1 for 2–4 h, 2–5 rectangular areas that each contained more than three FSP1 condensates were selected as bleaching areas. One image acquired before bleaching was considered to correspond to time '0'. Subsequently, the selected areas were photobleached using the maximum intensity of the laser and FRAP was monitored at minimum intervals (~5 s) using an LSM880 microscope (Zeiss).

To quantify the FRAP rate, a region of interest (ROI) for each condensate ($i$) in the photobleached area and one condensate ($c$) in a non-photobleached area was determined using ImageJ/Fiji and the mean fluorescence intensity of condensate $i$ at time $t$, $F_i(t)$ was obtained. After determining each time of fluorescence value, $F_i(t)$ was normalized by the value of $F_i(0)$ to obtain the relative fluorescence ($RF_i(t)$) of each bleached condensate area. To reflect quenching effects during observation and photobleaching, each $RF_i(t)$ value was normalized by relative fluorescence value at time $t$ of condensate ($c$) in non-bleached condensate areas ($RF_c(t)$) as follows: $F_i(t) = RF_i(t)/RF_c(t) = (F_i(t)/F_i(0))/(F_c(t)/F_c(0))$. Finally, the FRAP rate (%) at time $t$ in the particles was calculated as the mean of $F_i(t) × 100$ as described previously[21].

## Live-cell imaging

For co-staining or washout analyses, Pfa1 cells (15,000–30,000 cells) were seeded on μ-Dish 35-mm low dishes (Ibidi, cat. no. 80136) and incubated overnight. The next day, the cell culture medium was changed to FluoroBrite DMEM (Gibco, cat. no. A1896701) supplemented with 10% FBS, 2 mM L-glutamine and 1% penicillin-streptomycin. Live-cell microscopy was performed using the 3D Cell Explorer (Nanolive) with Eve v.1.8.2 software and the corresponding appropriate filter sets. During imaging, cells were maintained at 37 °C and 5% $CO_2$ using a temperature-controlled incubation chamber. For co-staining analysis, cells were pretreated for 1 h with 5 μM Liperfluo (Dojindo, cat. no. L248-10) and then changed to FluoroBrite DMEM containing 0.2 μg ml⁻¹ propidium iodide (Sigma, cat. no. P4170) and acquisition was started using Nanolive. After recording one image, 1 mM of icFSP1 in FluoroBrite DMEM was added to the dishes (final concentration of 10 μM) while continuing to record images. Images were acquired every 10 min for more than 4 h, and GFP, BFP and RFP filter sets were used to acquire signal. For washout experiments, high-glucose DMEM was changed to FluoroBrite DMEM before the experiments followed by data acquisition using Nanolive. After recording a few images, 0.25 mM of icFSP1 in FluoroBrite DMEM was added to the dishes (final concentration of 2.5 μM) and recording of images continued for 4 h. Thereafter, the dishes were carefully washed once with fresh FluoroBrite DMEM without icFSP1 and refilled with medium. Image acquisition was then immediately restarted. Images were recorded every 5 min for one more hour; that is, the total duration of data acquisition was around 5 h.

To determine the number of condensates in cells, Pfa1 cells (15,000–20,000 cells) were seeded on μ-Slide 8-well slides (Ibidi, cat. no. 80826) and incubated overnight. The next day, the medium was changed to high-glucose DMEM supplemented with 10% FBS, 2 mM L-glutamine, 1% penicillin-streptomycin, 2.5 μM icFSP1 and Hoechst. Immediately thereafter, the focus was adjusted and Hoechst and EGFP images were recorded using an Axio Observer Z1 imaging system with VisView v.4.0 (Visitron Systems, ZWISS) with a ×20 air objective and a CCD camera (CoolSnap ES2, Photometrics) with the corresponding filter sets. During imaging, cells were maintained at 37 °C and 5% $CO_2$ using a temperature-controlled incubation chamber. The imaging software ImageJ/Fiji was used for visualization, and CellProfiler (v.4.1.3, Broad Institute) was used to count the condensates in each cell.

## Subcutaneous tumour models

All mice were obtained from Charles River. For syngeneic subcutaneous tumour experiments, *Gpx4* and *Fsp1* double-knockout B16F10 cells stably overexpressing hFSP1–HA (1 × 10⁶ cells in 100 μl PBS) were injected subcutaneously into the right flank of 7-week-old female C57BL/6J mice. After tumours reached approximately 25–50 mm³ in size, mice were randomized and treated with vehicle or icFSP1 (50 mg kg⁻¹, Intonation) by intraperitoneal injection twice daily for 4–5 days. To generate tumour samples for staining, *Gpx4* and *Fsp1* double-knockout B16F10 cells stably expressing hFSP1^WT–HA or hFSP1^Q319K–HA (1 × 10⁶ cells in 100 μl PBS) were injected subcutaneously into the right flank of 7-week-old female C57BL/6J mice. After tumours reached approximately 25 mm³ in size, mice were randomized and treated with vehicle or icFSP1 (50 mg kg⁻¹, Intonation) by intraperitoneal injection twice daily .

For xenograft subcutaneous tumour experiments, *GPX4*-knockout A375 cells (5 × 10⁶ cells in 100 μl PBS) were injected subcutaneously into the right flank of 7-week-old female athymic nude mice. After tumours reached approximately 25–100 mm³ in size, mice were randomized and treated with vehicle or icFSP1 (50 mg kg⁻¹, Intonation) by intraperitoneal injection twice daily for the first 4 days and once daily afterwards.

For xenograft subcutaneous tumour experiments, *GPX4*-knockout H460 cells (5 × 10⁶ cells in 100 μl PBS) were injected subcutaneously into the right flank of 6-week-old female athymic nude mice. After tumours reached approximately 100 mm³ in size, mice were randomized and treated with vehicle or icFSP1 (50 mg kg⁻¹, Intonation) by intraperitoneal injection twice daily.

icFSP1 was dissolved in 45% PEG E 300 (Sigma, cat. no. 91462-1KG) and 55% PBS (Gibco, cat. no. 14190094). Tumours were measured by calliper every day, and tumour volume was calculated using the following

formula: tumour volume = length × width$^2$ × 0.52. When the tumour was greater than 1,000 mm$^3$ in size at measurement or the tumour became necrotic, tumours were considered to have reached the humane endpoint. When tumours reached the humane endpoint, the experiment was stopped and no further study was conducted.

## Tumour tissue staining

Dissected tissues were fixed in 4% paraformaldehyde in PBS overnight at 4 °C. For immunofluorescence staining, fixed tissues were incubated in 20% sucrose in PBS overnight at 4 °C, followed by embedding in OCT mounting compound (TissueTek, Sakura) on dry ice and storage at −80 °C. Frozen tissues were cut into 5-μm-thick sections using a Cryostat Microm HM 560 (Thermo Fisher Scientific) at −30 °C. Tissue sections were postfixed with 1% paraformaldehyde in PBS for 10 min and subsequently fixed with 67% ethanol and 33% acetic acid for 10 min. Sections were incubated with blocking solution (5% goat serum and 0.3% Triton X-100 in PBS) for 30 min and then incubated with primary antibodies (anti-HA (clone, 3F10; 1:10; developed in house), anti-4-HNE (JaICA, cat. no. HNEJ-2; 1:50) and anti-AIFM2 (FSP1, clone 14D7; undiluted; developed in house)) diluted in blocking solution overnight at 4 °C. The next day, sections were incubated with appropriate fluorophore-conjugated secondary antibodies (goat anti-rat Alexa Fluor 488 IgG (H+L) (1:500; A-11006, Invitrogen), goat anti-mouse IgG H&L Alexa Fluor 647 (1:500; ab150115, Abcam) and donkey anti-rat IgG Alexa Fluor 555 (1:500; ab150154, Abcam)) in secondary dilution buffer (1% BSA and 0.3% Triton X-100 in PBS) for 2 h at room temperature. DNA was visualized with DAPI staining for 5 min, and slides were mounted in Aqua-/PolyMount. Images were obtained using an LSM880 microscope (Zeiss) and analysed with Zen Blue or ImageJ/Fiji software.

## Pharmacokinetics and metabolic stability analyses

All studies were performed by Bienta/Enamine Ltd.

## Statistical analysis

All data shown are the mean ± s.e.m. or mean ± s.d., and the number (*n*) in each figure legend represents biological or technical replicates as specified. All experiments (except those described otherwise in the legend) were performed independently at least twice. For mouse experiments, at least three animals were included per group once or twice. Two-tailed Student's *t* tests and one-way or two-way ANOVA followed by Bonferroni's, Dunnett's, Tukey's or Sidak's multiple-comparison tests were performed using GraphPad Prism 9 (GraphPad Software) (also see figure legends for more detail). The results of the statistical analyses are presented in each figure. *P* < 0.05 was considered to be statistically significant.

## Reporting summary

Further information on research design is available in the Nature Portfolio Reporting Summary linked to this article.

## Data availability

All data are available in the article and its Supplementary Information as well as from the corresponding author on reasonable request. Gel source images are shown in Supplementary Fig. 2. Cancer cell line data were mined from https://depmap.org/portal/. Prediction of the phase separation of FSP1 was conducted with https://iupred2a.elte.hu and https://mobidb.bio.unipd.it. Source data are provided with this paper.

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

**Acknowledgements** We are grateful to all current and former members of the Conrad laboratory for providing valuable materials and fruitful discussions. This work was supported by funding from the Deutsche Forschungsgemeinschaft (DFG) (CO 291/9-1, 461385412; and the Priority Program SPP 2306 (CO 291/9-1, 461385412; CO 291/10-1, 461507177)) to M.C. and PR 1752/3-1 to B.P., a Kekulé fellowship from the Fonds der Chemischen Industrie to C.H., the Priority Program SPP2191 to M.S., a Helmholtz Munich internal development grant and the European Research Council under the European Union's Horizon 2020 research and innovation programme (grant agreement no. GA 884754) to M.C. We thank M. Rehberg for his initial help with live-cell imaging. SUDHL5, SUDHL6, DOHH2 and OCI-Ly19 cells were kindly provided by S. Hailfinger.

**Author contributions** T.N., M.S., B.P. and M.C. conceived the study and wrote the manuscript. T.N. and C.H. performed in vitro phase separation experiments. T.N., D.E., J.W., E.L. and E.M. performed experiments in cells. T.N. and J.W. established and performed in vivo experiments. M.A. and B.H. performed oxilipidomic analysis. J.B. and A.S.D.M. expressed and purified recombinant FSP1 and conducted NMR experiments. All authors read and agreed on the content of the paper.

**Funding** Open access funding provided by Helmholtz Zentrum München - Deutsches Forschungszentrum für Gesundheit und Umwelt (GmbH).

**Competing interests** M.C., B.P. and T.N. hold patents for some of the compounds described herein, and M.C. and B.P. are co-founders and shareholders of ROSCUE Therapeutics GmbH. The other authors declare no competing interests.

**Additional information**
**Correspondence and requests for materials** should be addressed to Marcus Conrad.

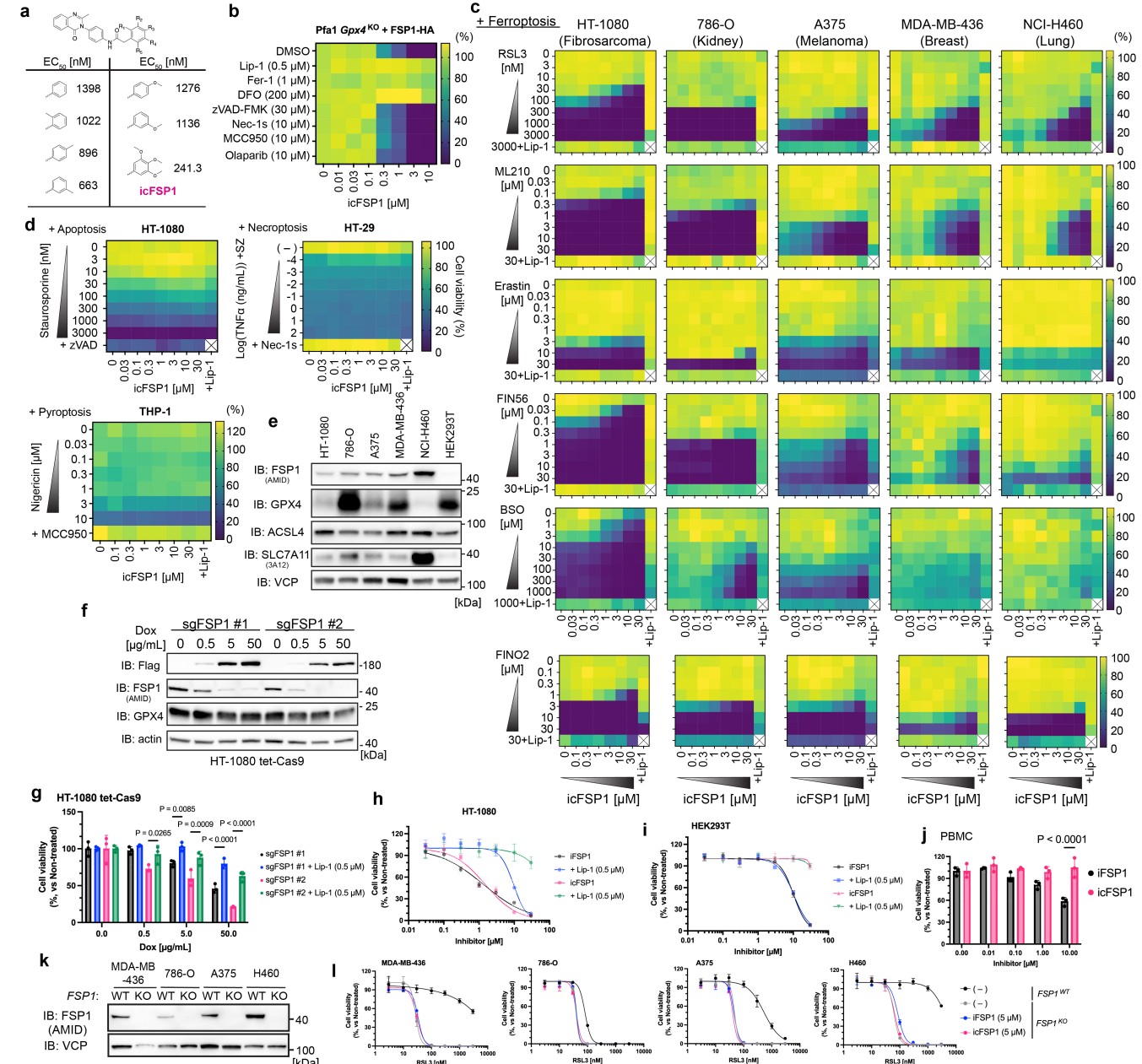

**Extended Data Fig. 1 | Development of icFSP1 and synergistic effects of icFSP1 with ferroptosis inducers in a variety of human cancer cells.**

**a**. Structure and EC50 values of icFSP1 derivatives. **b**. Cell viability was measured after treating Pfa1 *Gpx4* KO cells stably overexpressing FSP1-HA with icFSP1 along with different cell death inhibitors against ferroptosis (Lip-1 and ferrostatin-1 [Fer-1] or an iron chelator deferoxamine [DFO]), apoptosis (z-VAD-FMK), necroptosis (Nec-1s), pyroptosis (MCC950), and parthanatos (Olaparib). Heatmaps represent the mean of 3 wells of a 96 well plate from one out of 2 independent experiments. **c**. Cell viability was measured after treating HT-1080, A375, 786-O, MDA-MB-436 and H460 cells with different ferroptosis inducers (RSL3, ML210, erastin, FIN56 and FINO2 for 48 h, BSO for 72 h). Heatmaps represent one out of 2 independent experiments. 0.5 μM Lip-1 was used as control. **d**. Cell viability was measured after treating HT-1080, HT-29 and THP-1 cells with icFSP1 and respective cell death inducers (staurosporine for apoptosis, TNFα + smac mimetic (S) + z-VAD-FMK (Z) for necroptosis, and nigericin for pyroptosis) for 4 h (in case of pyroptosis) or 24 h (others). Heatmaps represent one out of 2 independent experiments. 0.5 μM Lip-1, 30 μM z-VAD-FMK, 10 μM Nec-1s and 10 μM MCC950 were used as positive controls for each mode of cell death. **e**. Representative immunoblot analysis of GPX4, FSP1 (AMID), SLC7A11(3A12), ACSL4 and VCP expression in the different cell lines from one out of 2 independent experiments. **f**. Representative immunoblot analysis of GPX4, FSP1, Flag (Cas9) and actin expression in HT-1080 cells with doxycycline (Dox)-inducible expression of Flag-Cas9 and stably expressing different sgRNAs targeting FSP1 (sgFSP1) from one out of 2 independent experiments. **g**. Cell viability was measured after treating HT-1080 cells expressing dox-inducible Flag-Cas9 and stably expressing sgFSP1 with doxycycline for 72 h. 0.5 μM Lip-1 was used as a positive control for the prevention of ferroptosis. **h**. Cell viability in HT-1080 cells treated with iFSP1 or icFSP1 and 0.5 μM Lip-1 for 72 h. **i**. Cell viability in HEK293T cells treated with iFSP1 or icFSP1 and 0.5 μM Lip-1 for 72 h. **j**. Cell viability in human PBMC cells treated with iFSP1 or icFSP1 for 24 h. **k**. Representative immunoblot analysis of FSP1 and VCP expression in WT and *FSP1* KO different cancer cell lines from one out of 2 independent experiments. **l**. Cell viability in *FSP1* WT or *FSP1* KO MDA-MB-436, 786-O, A375, H460 cells treated with 5 μM iFSP1 or icFSP1 for 48 h. Data represents the mean ± SD of 3 wells of a 96 well or 384 well plates from one out of 2 independent experiments (g-i, l) or a single experiment (j). Two-way ANOVA followed by Tukey's multiple comparison tests (g, j).

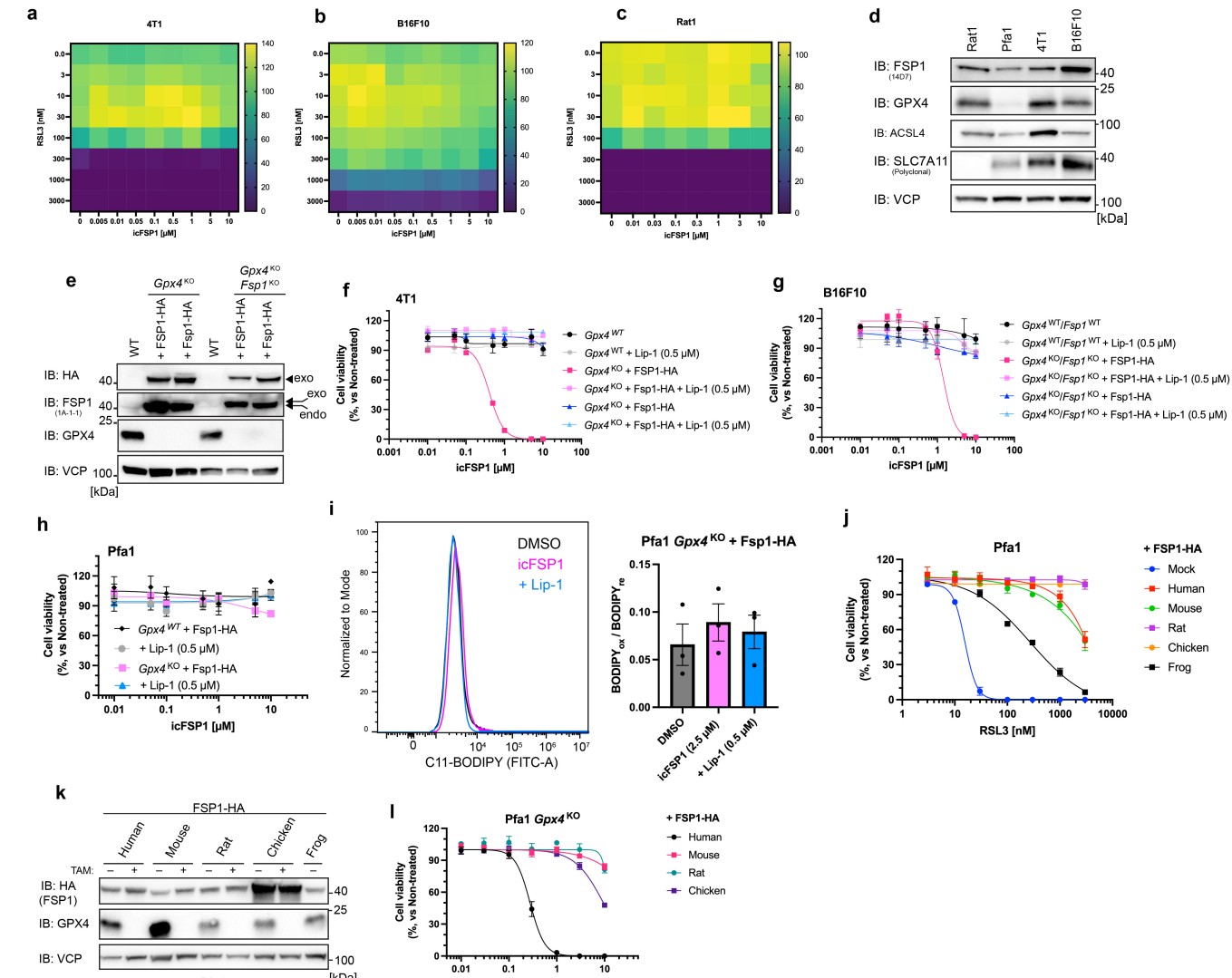

**Extended Data Fig. 2 | icFSP1 fails to inhibit mouse FSP1. a.** Cell viability was measured after treating 4T1 cells with icFSP1 and RSL3 for 48 h. **b.** Cell viability was measured after treating B16F10 cells with icFSP1 and RSL3 for 48 h. **c.** Cell viability was measured after treating Rat1 cells with icFSP1 and RSL3 for 48 h. **d.** Representative immunoblot analysis of GPX4, FSP1 (14D7), SLC7A11, ACSL4 and VCP expression in different cell lines from one out of 2 independent experiments. **e.** Representative immunoblot analysis of GPX4, FSP1, and VCP expression of 4T1 *Gpx4*^WT cells, 4T1 *Gpx4*^KO cells stably overexpressing hFSP1-HA or mFsp1-HA, B16F10 *Gpx4*^WT/*Fsp1*^WT cells, B16F10 *Gpx4*^KO/*Fsp1*^KO cells stably overexpressing hFSP1-HA or mFsp1-HA from one out of 2 independent experiments. **f.** Cell viability was measured after treating 4T1 *Gpx4*^WT cells, 4T1 *Gpx4*^KO cells stably overexpressing hFSP1-HA or mFsp1-HA with icFSP1 and 0.5 μM Lip-1 for 48 h. **g.** Cell viability was measured after treating B16F10 *Gpx4*^WT/*Fsp1*^WT cells, B16F10 *Gpx4*^KO/*Fsp1*^KO cells stably overexpressing hFSP1-HA or mFsp1-HA with icFSP1 and 0.5 μM Lip-1 for 48 h. **h.** Cell viability was measured after treating Pfa1 *Gpx4*^WT and *Gpx4*^KO cells stably overexpressing hFSP1-HA or mFsp1-HA with icFSP1 and 0.5 μM Lip-1 for 48 h. **i.** Lipid peroxidation was measured by C11-BODIPY 581/591 staining after treating Pfa1 *Gpx4*^KO cells stably overexpressing Fsp1-HA with DMSO or 2.5 μM icFSP1 and 0.5 μM Lip-1 for 3 h. Representative plots of one out of 3 independent experiments (left) and quantified median values (right, mean ± SEM) of 3 independent experiments are shown. **j.** Cell viability was measured after treating Pfa1 *Gpx4*^WT cells stably overexpressing FSP1 from different species (i.e., *Homo sapiens* (human), *Mus musculus* (mouse), *Rattus norvegicus* (rat), *Gallus gallus* (chicken), and *Xenopus laevis* (frog)) with RSL3 for 24 h. **k.** Representative immunoblot analysis of GPX4, HA (FSP1) and VCP expression in Pfa1 *Gpx4*^WT and *Gpx4*^KO cells stably overexpressing orthogonal FSP1 from one out of 2 independent experiments. **l.** Cell viability was measured after treating Pfa1 *Gpx4*^KO cells stably expressing FSP1 with icFSP1 for 24 h. Data represents the mean ± SD of 3 wells of a 96 well plate from one out of 2 independent experiments (f,g,h,j,l). Heatmaps represent the mean of 3 independent experiments (a-c).

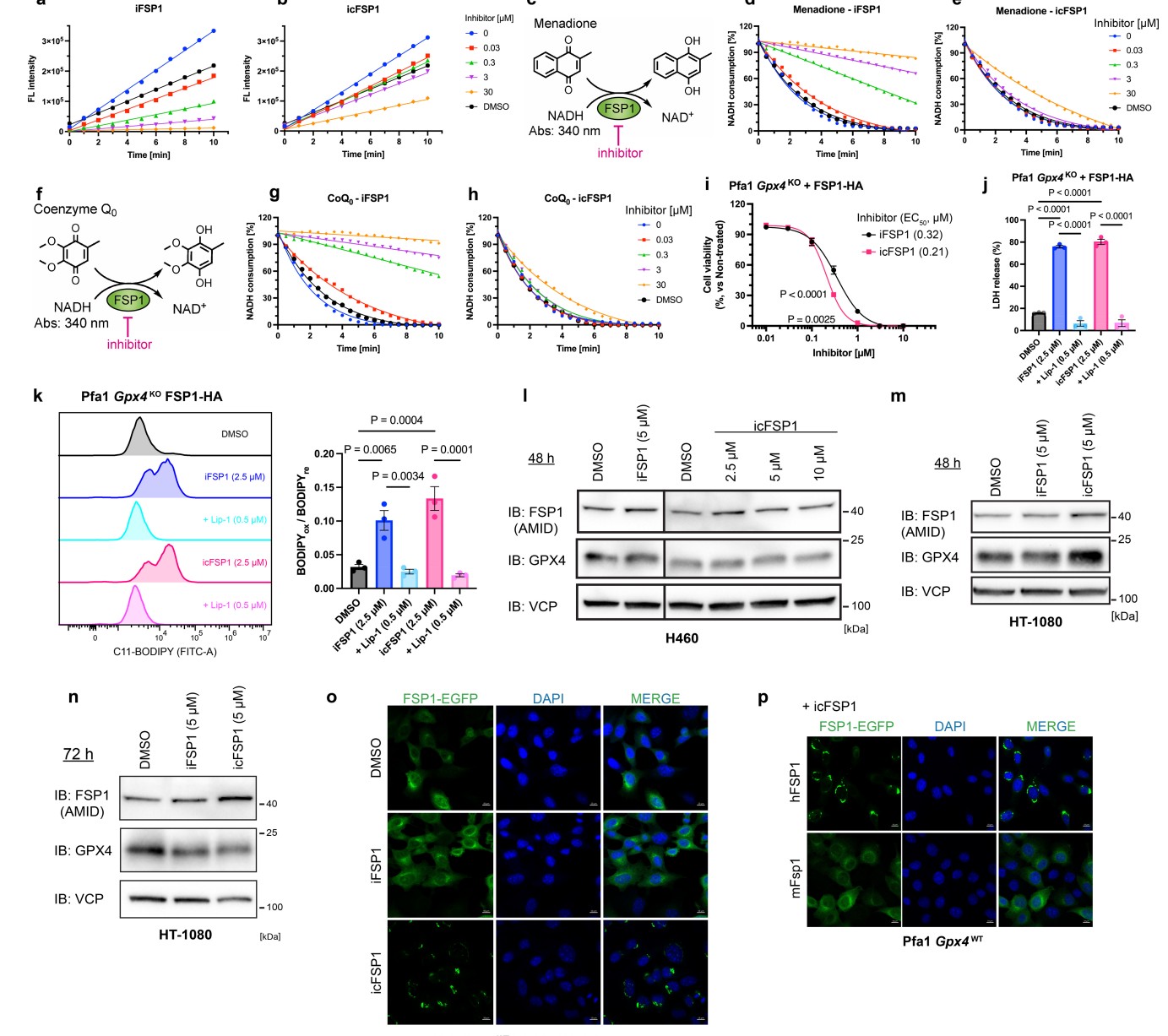

**Extended Data Fig. 3 |** See next page for caption.

**Extended Data Fig. 3 | icFSP1 has no impact on FSP1 enzyme activity and expression. a**. Representative reaction curves of *in vitro* assay. iFSP1 toward recombinant hFSP1 activity was assessed by fluorescent (FL) intensity of reduced form of resazurin. **b**. Representative reaction curves of *in vitro* assay. icFSP1 toward recombinant hFSP1 activity was assessed by FL intensity of reduced form of resazurin. Data represents the mean of 3 wells of a 96 well plate from one out of 3 independent experiments (**a**,**b**). **c**. Schematic representation of the FSP1 enzyme activity assay using menadione as a substrate. **d**. NADH consumption assay *in vitro*. Representative reaction curves of the FSP1 enzyme activity assay for measuring NADH consumption. iFSP1 toward recombinant hFSP1 activity was assessed by determining the absorbance of NADH. **e**. NADH consumption assay *in vitro*. Representative reaction curves of the FSP1 enzyme activity assay for measuring NADH consumption. icFSP1 toward recombinant hFSP1 activity were assessed by determining the absorbance of NADH. Data represents the curve fitting line and an original single value of one well of a 96-well plate from one out of 3 independent experiments. 0 μM and DMSO samples represent the same samples (**d**,**e**). **f**. Schematic representation of the FSP1 enzyme activity assay using $CoQ_0$ as the substrate. **g**. NADH consumption assay *in vitro*. Representative reaction curves of the FSP1 enzyme activity assay for measuring NADH consumption. iFSP1 toward recombinant hFSP1 activity was assessed by determining the absorbance of NADH. **h**. NADH consumption assay *in vitro*. Representative reaction curves of the FSP1 enzyme activity assay for measuring NADH consumption. icFSP1 toward recombinant hFSP1 activity were assessed by determining the absorbance of NADH. Data represents the curve fitting line and an original single value of one well of a 96-well plate from one out of 3 independent experiments. 0 μM and DMSO samples represent the same samples (**g**, **h**). **i**. Cell viability was measured after treating Pfa1 *Gpx4*[KO] cells stably overexpressing hFSP1-HA with iFSP1 or icFSP1 for 24 h. **j**. Lactate dehydrogenase (LDH) release was determined after treating Pfa1 *Gpx4*[KO] hFSP1-HA overexpressing cells with DMSO, 2.5 μM iFSP1 or icFSP1 or 0.5 μM Lip-1 for 24 h. Data represents the mean ± SD of 3 wells of a 96 well plate from one out of 3 independent experiments. P values were calculated by two-way ANOVA followed by Bonferroni's multiple comparison test (i, j). **k**. Lipid peroxidation was evaluated by C11-BODIPY 581/591 staining after treating *Gpx4*[KO] cells stably overexpressing hFSP1-HA with DMSO, 2.5 μM iFSP1 or icFSP1 and 0.5 μM Lip-1 for 3 h. Representative plots of one out of 3 independent experiments (left) and quantified median values of 3 independent experiments (right) are shown. Data represents the mean ± SEM of 3 independent experiments (k). one-way ANOVA followed by Tukey's multiple comparison test. **l**. Representative immunoblot analysis of GPX4, FSP1 and VCP expression after treating H460 cells with iFSP1 or icFSP1 for 48 h. **m**. Representative immunoblot analysis of GPX4, FSP1 and VCP expression after treating HT-1080 cells with iFSP1 or icFSP1 for 48 h. **n**. Immunoblot analysis of GPX4, FSP1 and VCP expression after treating HT-1080 cells with iFSP1 or icFSP1 for 72 h. **o**. Confocal microscopy fluorescence images after treating Pfa1 *Gpx4*[WT] cells stably overexpressing hFSP1-EGFP-Strep with 2.5 μM icFSP1 or iFSP1 for 4 h. Scale bars, 10 μm. **p**. Confocal microscopy fluorescence images after treating Pfa1 *Gpx4*[WT] cells stably overexpressing hFSP1-EGFP-Strep or mFsp1-EGFP-Strep with 2.5 μM icFSP1 or iFSP1 treatment for 4 h. Scale bars, 10 μm. Representative results are from one out of 2 independent experiments (l-p).

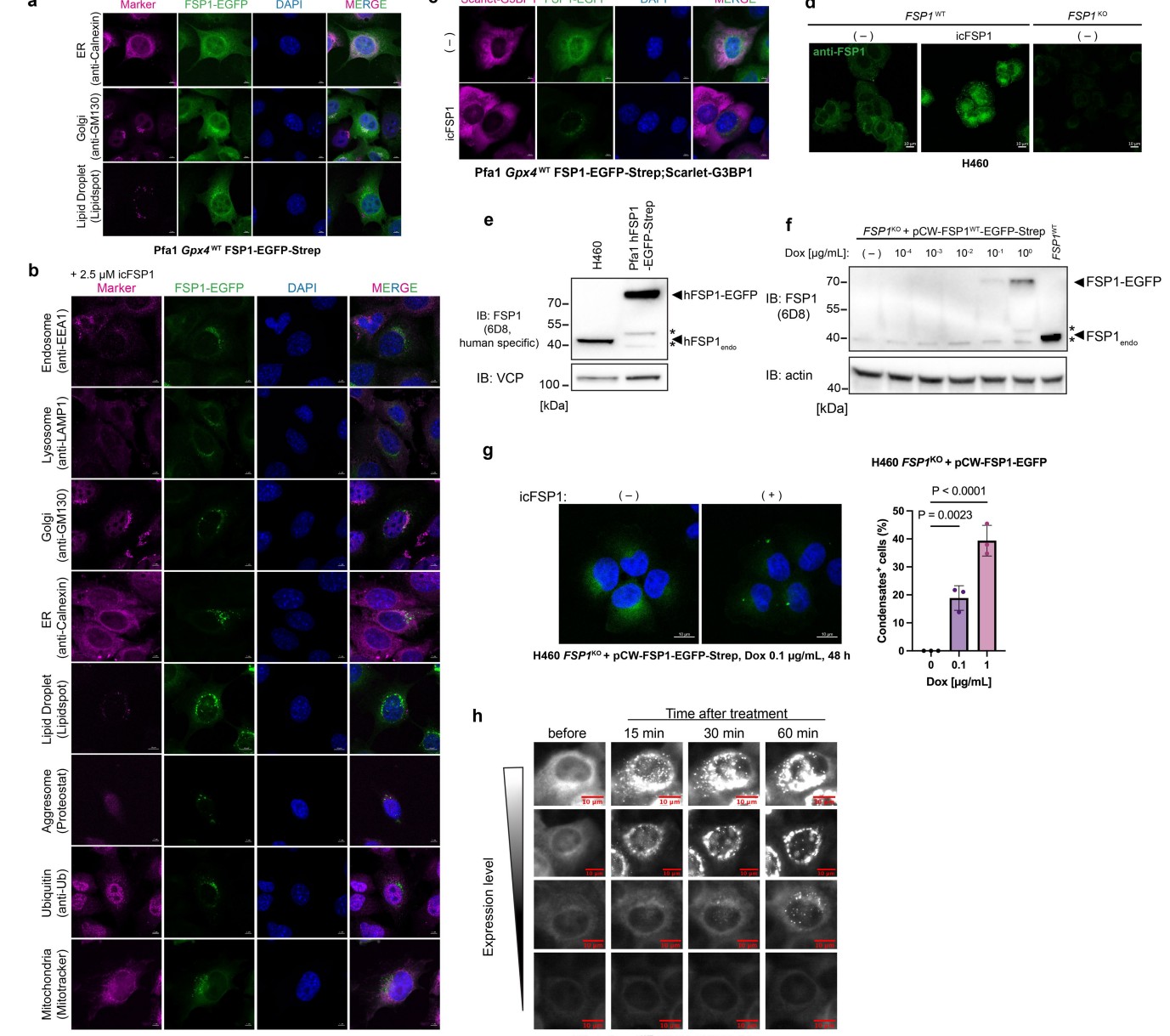

**Extended Data Fig. 4 | FSP1 condensates do not localize to specific organelles. a**. Confocal microscopy images of FSP1-EGFP-Strep overexpressing Pfa1 *Gpx4* WT cells stained with ER, Golgi and lipid droplet markers under normal cell culture conditions. Scale bars, 5 μm. **b**. Confocal microscopy images of FSP1-EGFP-Strep overexpressing Pfa1 *Gpx4* WT cells after treating with 2.5 μM icFSP1 for 120 min, and subsequently stained with markers for endosome, lysosome, Golgi, ER, lipid droplet, aggresome, ubiquitin and mitochondria. Scale bars, 5 μm. **c**. Confocal microscopy images of FSP1-EGFP-Strep and Scarlet-G3BP1 overexpressing Pfa1 *Gpx4* WT cells after treating with 0 or 2.5 μM icFSP1 for 120 min. Scale bars, 5 μm. **d**. Confocal microscopy images of H460 or *FSP1* KO H460 cells after treating with 0 or 10 μM icFSP1 for 4 h, following stained with FSP1 antibody (14D7). Scale bars, 10 μm. Representative results are from one out of 2 independent experiments (a-d). **e**. Representative immunoblot analysis of FSP1 and VCP expression in H460 cells and hFSP1-EGFP-Strep overexpressing Pfa1 *Gpx4* WT cells from one out of 2 independent experiments. **f**. Representative immunoblot analysis of FSP1 and actin expression in H460 WT cells and Dox-inducible hFSP1-EGFP-Strep overexpressing H460 *FSP1* KO cells after treating with Dox for 48 h from one out of 2 independent experiments. **g**. Confocal microscopy images of Dox-inducible hFSP1-EGFP-Strep overexpressing H460 *FSP1* KO cells after treating with 0.1 μg/mL for 48 h and followed by treating with 0 or 10 μM icFSP1 for 4 h. Scale bars, 10 μm. Representative images from one out of 2 independent experiments (left) and quantified values (right, mean ± SD of 3 independent fields) are shown. One-way ANOVA followed by Dunnett's multiple comparison test. **h**. Time-lapse fluorescent images of Pfa1 *Gpx4* WT cells stably overexpressing hFSP1-EGFP-Strep before and immediately after treatment with 2.5 μM icFSP1 for the indicated time. Scale bars, 10 μm. Representative results are from one out of 2 independent experiments.

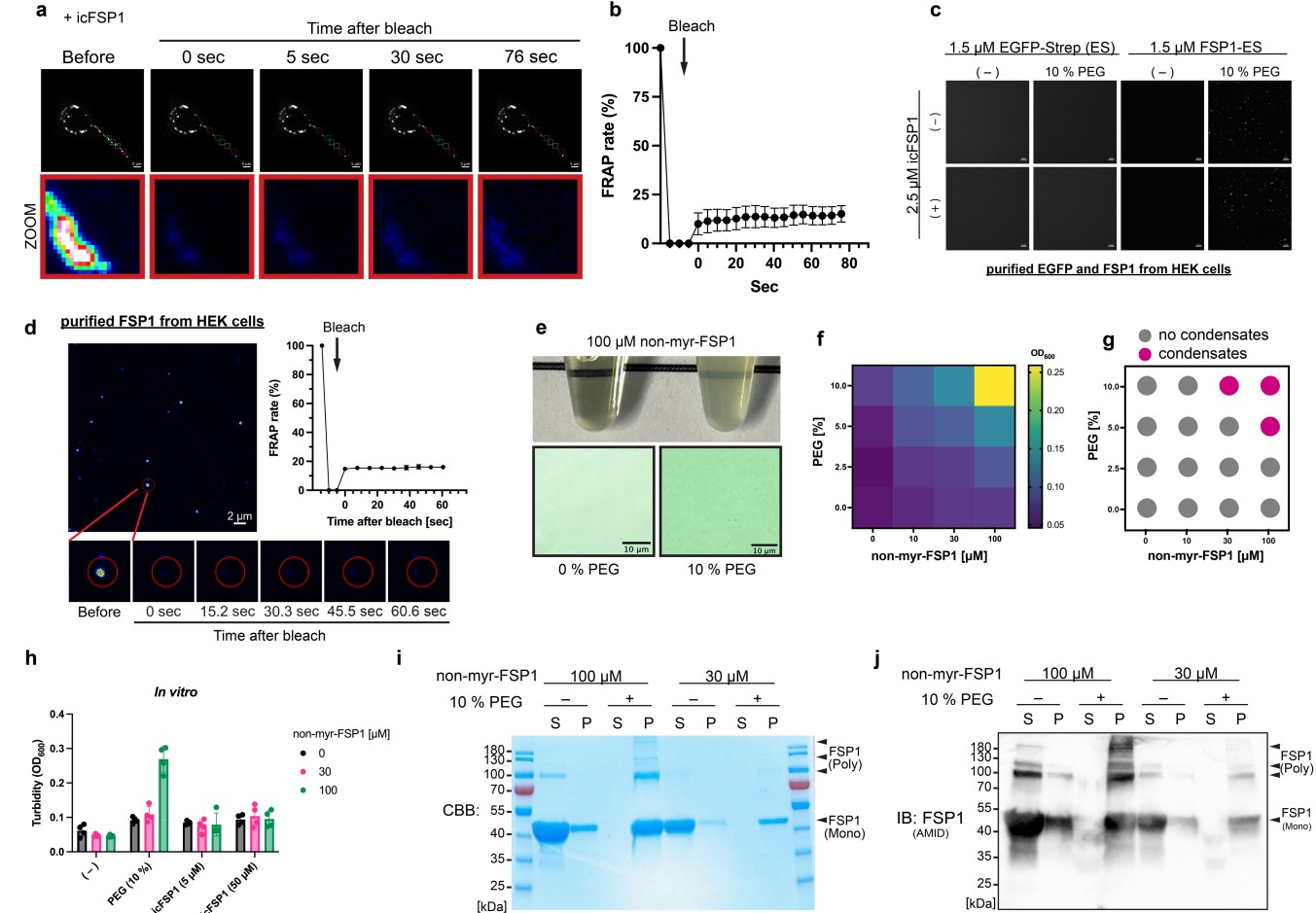

**Extended Data Fig. 5 | FSP1 forms viscoelastic material. a**. Fluorescence recovery after photobleaching (FRAP) assay after treating Pfa1 *Gpx4*[WT] cells stably overexpressing hFSP1-EGFP-Strep with 2.5 µM icFSP1 for 240 min. Greyscale images show representative FRAP images right before and at indicated time points after photo-bleaching. Lookup Table (LUT) images show enlarged red rectangle areas of upper FRAP images. Scale bars, 5 µm. Representative results from one out of 2 independent experiments. See also Supplementary Video 6. **b**. Quantified FRAP rate of each condensate. Data represents the mean ± SD of 4 condensates from Extended Data Fig. 5a. Representative results from one out of 2 independent experiments are shown. **c**. FSP1 condensation *in vitro*. Representative confocal microscopy images of 1.5 µM EGFP-Strep and hFSP1-EGFP-Strep purified from transfected HEK29T cells were obtained immediately after mixing with 2.5 µM icFSP1 with or without 10% PEG. Scale bars, 10 µm. Representative results are shown from one out of 2 independent experiments. **d**. FRAP assay after incubating 0.5 µM hFSP1-EGFP-Strep with 10 % PEG. LUT images show enlarged red rectangle areas of FRAP images (left and below). Scale bars, 2 µm. Representative results from one out of 2 independent experiments. Quantified FRAP rate of each condensate (right). Data represents the mean ± SD of 4 independent condensates. **e**. Recombinant non-myristoylated FSP1 (non-myr-FSP1) condensation *in vitro*. Representative bright field images of

100 µM non-myr-FSP1 expressed and purified from *E.coli* right after mixing with or without 10% PEG in PCR tubes (top). Representative blight field images (bottom) were obtained using the microscope. Scale bars, 10 µm. Representative results are from one out of 2 independent experiments. **f**. Absorbance of 600 nm was measured for different concentrations of PEG and non-myr-FSP1. Heatmaps represent the mean of 2 independent experiments. **g**. Phase diagram of the presence of condensates was determined by microscopy after absorbance measurement (f). **h**. Absorbance of 600 nm was measured for different concentrations of PEG and icFSP1, non-myr-FSP1. Data represents the mean ± SD from 4 wells of 384 well plates from one out of 2 independent experiments. **i**. Sedimentation assay of non-myr-FSP1. Representative Coomassie brilliant blue (CBB) staining of non-myr-FSP1 fraction after centrifugation in the presence or absence of 10% PEG. S: supernatant, P: re-suspended pellet. Monomer (Mono, ~ 42 kDa) and estimated oligomerized FSP1 (Poly, ~ 90 kDa) can be observed. **j**. Immunoblot analysis of non-myr-FSP1 fraction after centrifugation in the presence or absence of 10% PEG. S: supernatant, P: re-suspended pellet. Monomer (Mono, ~ 42 kDa) and estimated oligomerized FSP1 (Poly, ~ 90 kDa) can be observed. The same samples but different gels were used for CBB staining (i) and immunoblot analysis (j). Representative results from one out of 2 independent experiments (i. j).

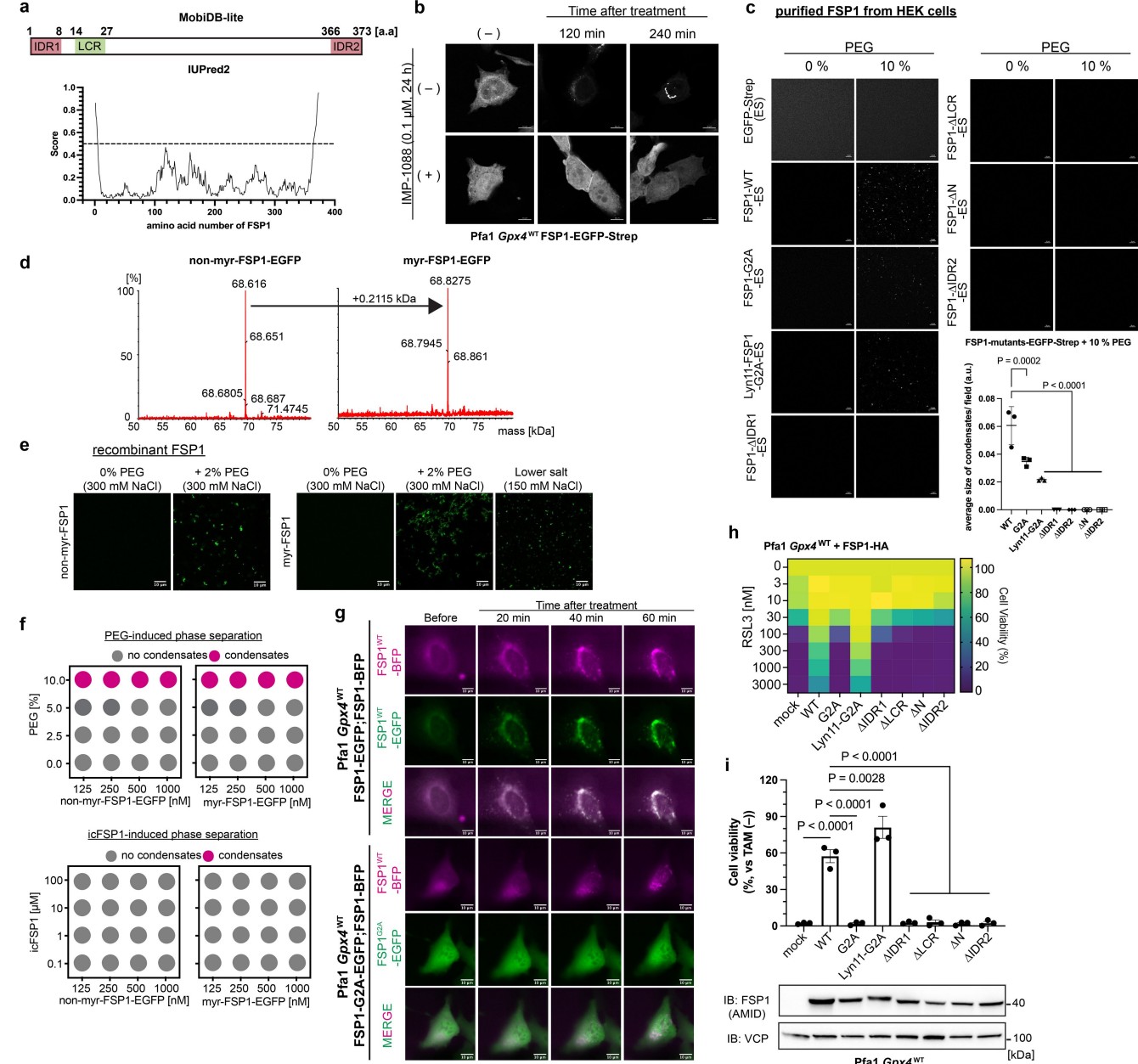

**Extended Data Fig. 6 | Myristoylation, IDRs, and LCR are required for the anti-ferroptotic role of FSP1 as well as condensations. a**. IDR and LCR prediction from FSP1 sequence. From MobiDB-lite prediction (top), there are 2 IDRs (1–8, 366-373 amino acid [a.a.]) and one LCR (14−27 a.a.) in FSP1. IUPred2 scores are visualized, and the dashed line shows the threshold; 0.5 (bottom). **b**. Confocal microscopy images of hFSP1-EGFP-Strep overexpressing Pfa1 *Gpx4* ^WT cells after pre-treating with or without 0.1 μM IMP-1088 for 24 h and subsequently treating with or without 2.5 μM icFSP1 for the indicated time. Scale bars, 10 μm. Representative results from one out of 2 independent experiments. **c**. FSP1 condensation *in vitro*. Fluorescent images of 1.5 μM EGFP-Strep and hFSP1-EGFP-Strep mutants purified from transfected HEK29T cells were obtained right after mixing with or without 10% PEG. Scale bars, 10 μm. Representative results from one out of 2 independent experiments of 3 different fields. Quantification results are shown as mean ± SD of n = 3 from one out of 2 independent experiments. Average sizes of condensates were calculated by Fiji/ImageJ. **d**. Mass spectrometry analysis of recombinant non-myristoylated FSP1-EGFP (non-myr-FSP1-EGFP) and myristoylated FSP-EGFP1 (myr-FSP1-EGFP) purified from *E. coli*. **e**. FSP1 condensation *in vitro*. Representative Fluorescent images represent 15 μM recombinant non-myr or myr-FSP1-EGFP with 0% or

2% PEG in 300 mM NaCl PBS or 10 μM recombinant myr-FSP1-EGFP with 150 mM NaCl in PBS as a lower salt from one out of 2 independent experiments. **f**. Phase diagram of the presence of condensates was determined by microscopy from one out of 2 independent experiments. **g**. Time-lapse fluorescent images of Pfa1 *Gpx4* ^WT cells stably overexpressing hFSP1-EGFP-Strep;hFSP1-mTagBFP or hFSP1-G2A-EGFP-Strep;hFSP1-mTagBFP before and immediately after treatment with 2.5 μM icFSP1 for the indicated times. Scale bars, 10 μm. Representative results are from one out of 2 independent experiments. **h**. Cell viability was measured after treating Pfa1 *Gpx4* ^WT cells stably overexpressing hFSP1 mutants with RSL3 for 24 h. The heatmap represents the mean of 3 wells of 96 well plates from one out of 2 independent experiments. **i**. Cell viability (top) was measured after treating Pfa1 *Gpx4* ^WT cells stably overexpressing hFSP1 mutants with or without 1 μM TAM for 72 h TAM. Data were normalized by each group of non-treatment with TAM. Immunoblot analysis (bottom) of FSP1 and VCP expression in Pfa1 *Gpx4* ^WT cells stably overexpressing hFSP1 mutants in cells from one out of 2 independent experiments. Data represents mean ± SEM of 3 independent experiments (i). P values were calculated by one-way ANOVA followed by Dunnett's multiple comparison test (c,i).

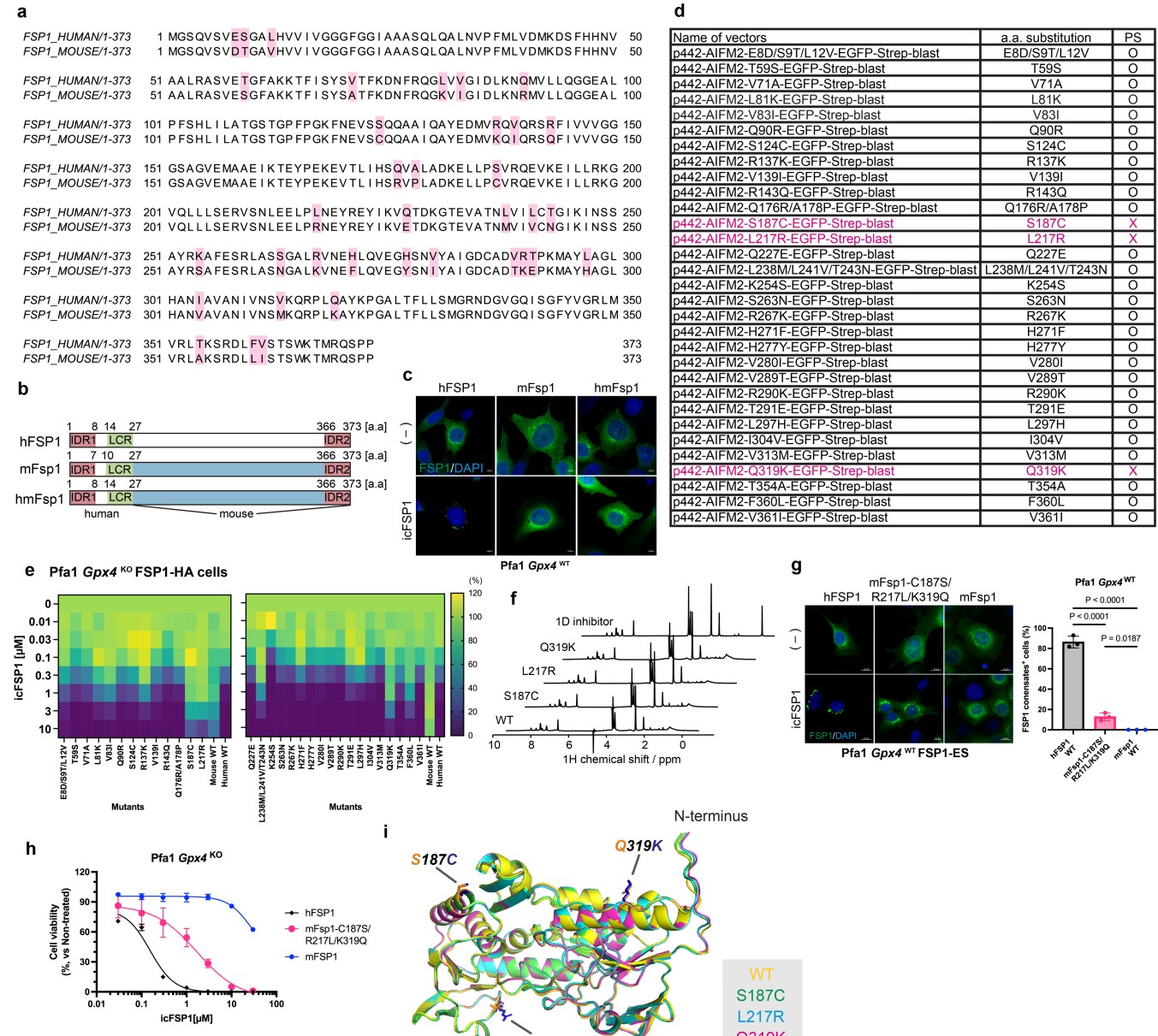

| Name of vectors | a.a. substitution | PS |
|---|---|---|
| p442-AIFM2-E8D/S9T/L12V-EGFP-Strep-blast | E8D/S9T/L12V | O |
| p442-AIFM2-T59S-EGFP-Strep-blast | T59S | O |
| p442-AIFM2-V71A-EGFP-Strep-blast | V71A | O |
| p442-AIFM2-L81K-EGFP-Strep-blast | L81K | O |
| p442-AIFM2-V83I-EGFP-Strep-blast | V83I | O |
| p442-AIFM2-Q90R-EGFP-Strep-blast | Q90R | O |
| p442-AIFM2-S124C-EGFP-Strep-blast | S124C | O |
| p442-AIFM2-R137K-EGFP-Strep-blast | R137K | O |
| p442-AIFM2-V139I-EGFP-Strep-blast | V139I | O |
| p442-AIFM2-R143Q-EGFP-Strep-blast | R143Q | O |
| p442-AIFM2-Q176R/A178P-EGFP-Strep-blast | Q176R/A178P | O |
| p442-AIFM2-S187C-EGFP-Strep-blast | S187C | X |
| p442-AIFM2-L217R-EGFP-Strep-blast | L217R | X |
| p442-AIFM2-Q227E-EGFP-Strep-blast | Q227E | O |
| p442-AIFM2-L238M/L241V/T243N-EGFP-Strep-blast | L238M/L241V/T243N | O |
| p442-AIFM2-K254S-EGFP-Strep-blast | K254S | O |
| p442-AIFM2-S263N-EGFP-Strep-blast | S263N | O |
| p442-AIFM2-R267K-EGFP-Strep-blast | R267K | O |
| p442-AIFM2-H271F-EGFP-Strep-blast | H271F | O |
| p442-AIFM2-H277Y-EGFP-Strep-blast | H277Y | O |
| p442-AIFM2-V280I-EGFP-Strep-blast | V280I | O |
| p442-AIFM2-V289T-EGFP-Strep-blast | V289T | O |
| p442-AIFM2-R290K-EGFP-Strep-blast | R290K | O |
| p442-AIFM2-T291E-EGFP-Strep-blast | T291E | O |
| p442-AIFM2-L297H-EGFP-Strep-blast | L297H | O |
| p442-AIFM2-I304V-EGFP-Strep-blast | I304V | O |
| p442-AIFM2-V313M-EGFP-Strep-blast | V313M | O |
| p442-AIFM2-Q319K-EGFP-Strep-blast | Q319K | X |
| p442-AIFM2-T354A-EGFP-Strep-blast | T354A | O |
| p442-AIFM2-F360L-EGFP-Strep-blast | F360L | O |
| p442-AIFM2-V361I-EGFP-Strep-blast | V361I | O |

**Extended Data Fig. 7 | Mutational analysis of human FSP1 resistant to icFSP1. a.** Protein sequence alignment of human and mouse FSP1 using Jalview (v2.11.2.6). **b.** Schematic model of chimeric FSP1 mutants. From MobiDB-lite prediction, there are 2 IDRs (1–7, 366–373 a.a.) and one LCR (10–27 a.a.) in FSP1. Chimeric hmFsp1 was generated from human FSP1 (1–27 a.a.) and mouse Fsp1 (28–373 a.a., marked as blue). **c.** Confocal microscopy images after treating Pfa1 *Gpx4* WT cells stably overexpressing FSP1-EGFP-Strep mutants with and without 2.5 μM icFSP1 for 240 min. Scale bars, 5 μm. Representative results from one out of 2 independent experiments. **d.** Summary of the viability assay using single point mutations (as indicated) from human to mouse in FSP1-EGFP-Strep expressing constructs. After treating cells with 2.5 μM icFSP1 for overnight, the presence of condensates was determined using a microscope. Human FSP1 is sensitive to icFSP1 (O), while mouse Fsp1 is resistant to icFSP1 (X). 3 mutants show complete resistance (X). This screening was performed once using a 96 well plate. **e.** Cell viability was measured after treating Pfa1 *Gpx4* KO cells stably overexpressing hFSP1 mutants with icFSP1 for 24 h. Heatmaps represent the mean of 3 wells of 96 well plate from one out of 2 independent experiments. **f.** Saturation transfer difference (STD) spectra of WT hFSP1 or its mutants S187C, L217R and Q319K show binding of icFSP1 (bottom to top). Top spectrum shows a 1D ¹H reference spectrum of icFSP1. **g.** Confocal microscopy images after treating Pfa1 *Gpx4* WT cells stably overexpressing human and mouse FSP1-EGFP-Strep or the combination mutant (mFSP1-C187S/R217L/K319Q) with and without 2.5 μM icFSP1 for 240 min. Scale bars, 10 μm. Representative images (left) and quantitative results (right, mean ± SD of 3 different fields) from one out of 2 independent experiments. One-way ANOVA followed by Bonferroni's multiple comparison test. **h.** Cell viability was measured after treating Pfa1 *Gpx4* KO cells stably overexpressing human and mouse FSP1-HA or the combination mutant with icFSP1 for 24 h. Data represents the mean ± SD of 3 wells of a 96 well plate from one out of 2 independent experiments. **i.** Predicted cartoon structure of hFSP1 WT (yellow) S187C (green), L217R (cyan), and Q319K (magenta) by AlphaFold2 or ColabFold. Each mutant amino acid residue represents human FSP1 (orange) and mouse FSP1 (dark blue) as sticks.

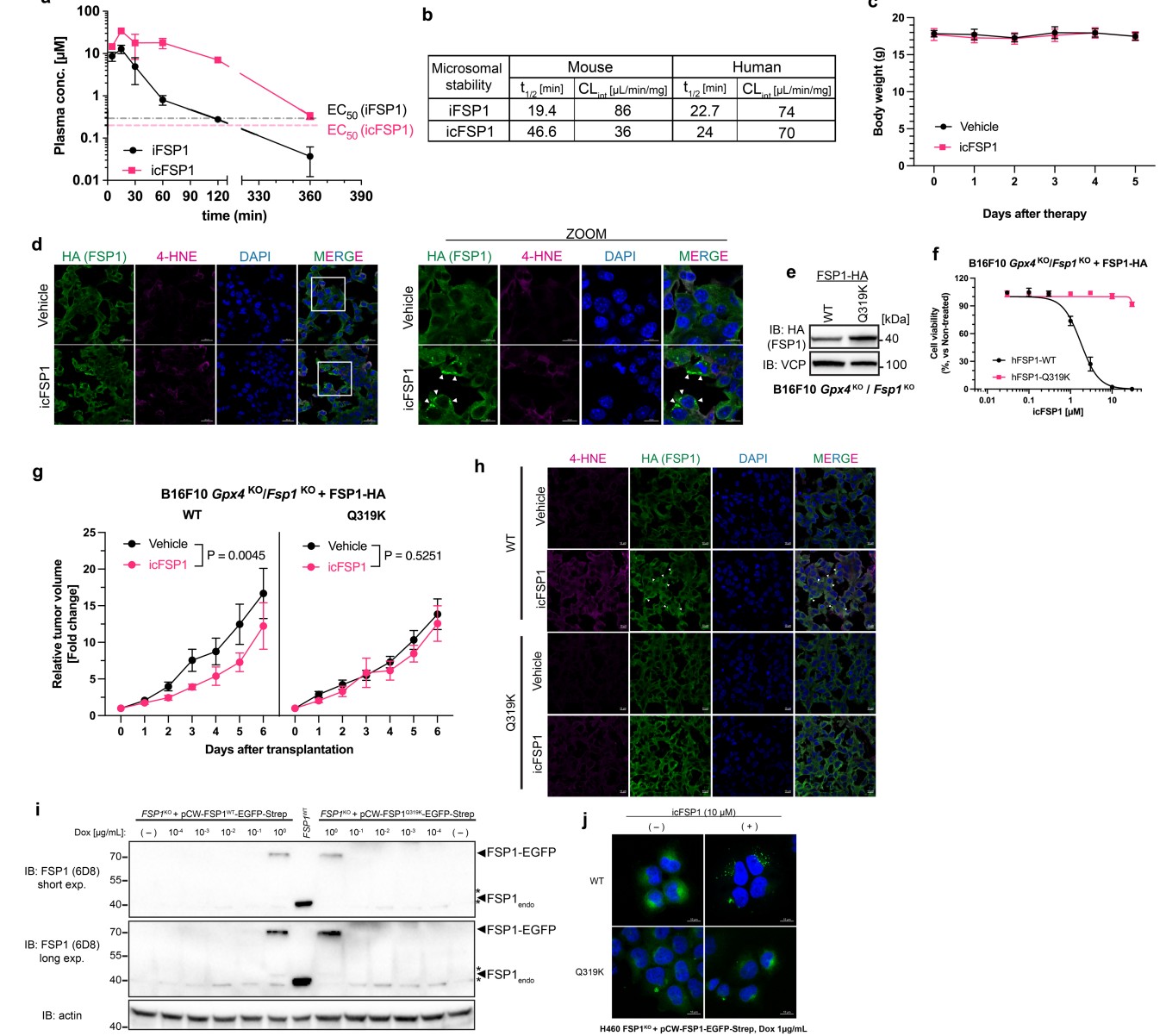

**Extended Data Fig. 8 | Targeting of FSP1 by icFSP1 as potential anti-cancer therapy using mouse cells. a**. Pharmacokinetic (PK) parameters of icFSP1 and iFSP1. Plasma concentration was measured after single i.p. administration (10 mg/kg). Data represents mean ± SD from 4 mice of one experiment. **b**. Summary of microsomal stability analysis of icFSP1 and iFSP1. **c**. Body weight of tumor-baring mice during the treatment of mice with icFSP1 (50 mg/kg i.p. twice a day, n = 7) and vehicle (n = 6) as a control. These mice are the same as in Fig. 4f. Data represents mean ± SD from one out of 2 independent experiments. **d**. At the end of the *in vivo* pharmacological studies, tumors were dissected, cryosectioned and stained with anti-HA to visualize hFSP1 and with anti-4-HNE to visualize the lipid peroxidation breakdown product. Representative confocal microscopy images are shown from one out of 2 independent experiments. Arrowheads indicate FSP1 condensates. Scale bars, 20 µm or 10 µm. **e**. Immunoblot analysis of HA (FSP1) and VCP expression of B16F10 *Gpx4* KO/*Fsp1* KO cells stably overexpressing FSP1-WT or the Q319K mutant. Representative results from one out of 2 independent experiments. **f**. Cell viability was measured after treating B16F10 *Gpx4* KO/*Fsp1* KO cells stably overexpressing FSP1-WT or Q319K mutants with icFSP1 for 48 h. Data represent mean ± SD of 3 wells from one out of 2 independent experiments. **g**. icFSP1 inhibits tumor growth of hFSP1 WT but not of FSP1-Q319K expressing cells *in vivo*. B16F10

*Gpx4* KO/*Fsp1* KO cells stably expressing hFSP1 WT or Q319K were subcutaneously implanted into C57BL/6J mice (n = 33, in total). Treatment with vehicle (n = 10 for WT and 8 for Q319K) or icFSP1 (50 mg/kg i.p. twice a day, n = 8 for WT and 7 for Q319K) was started from day 6 after randomization. Data represents the mean ± SEM from one out of 2 experiments. P values were calculated by two-way ANOVA followed by Tukey's multiple comparison test. **h**. At the end of the *in vivo* pharmacological studies, tumors were dissected, cryosectioned and stained with anti-HA to visualize hFSP1 and with anti-4-HNE to visualize the lipid peroxidation breakdown product. Representative confocal microscopy images are shown from one out of 2 independent experiments. Arrowheads indicate FSP1 condensates. **i**. Immunoblot analysis of FSP1 and actin expression of H460 WT and *FSP1* KO cells with doxycycline (Dox)-inducible FSP1-WT or the Q319K mutant after Dox induction with indicated concentrations for 48 h. The left part of the blot is already shown in Extended Data Fig. 4f. Representative results from one out of 2 independent experiments. **j**. Confocal microscopy images after treating H460 *FSP1* KO cells with doxycycline-inducible FSP1-WT or the Q319K mutant with 1 µg/mL of Dox for 48 h, followed by treatment with and without 10 µM icFSP1 for 240 min. Scale bars, 10 µm. Representative images from one out of 2 independent experiments.

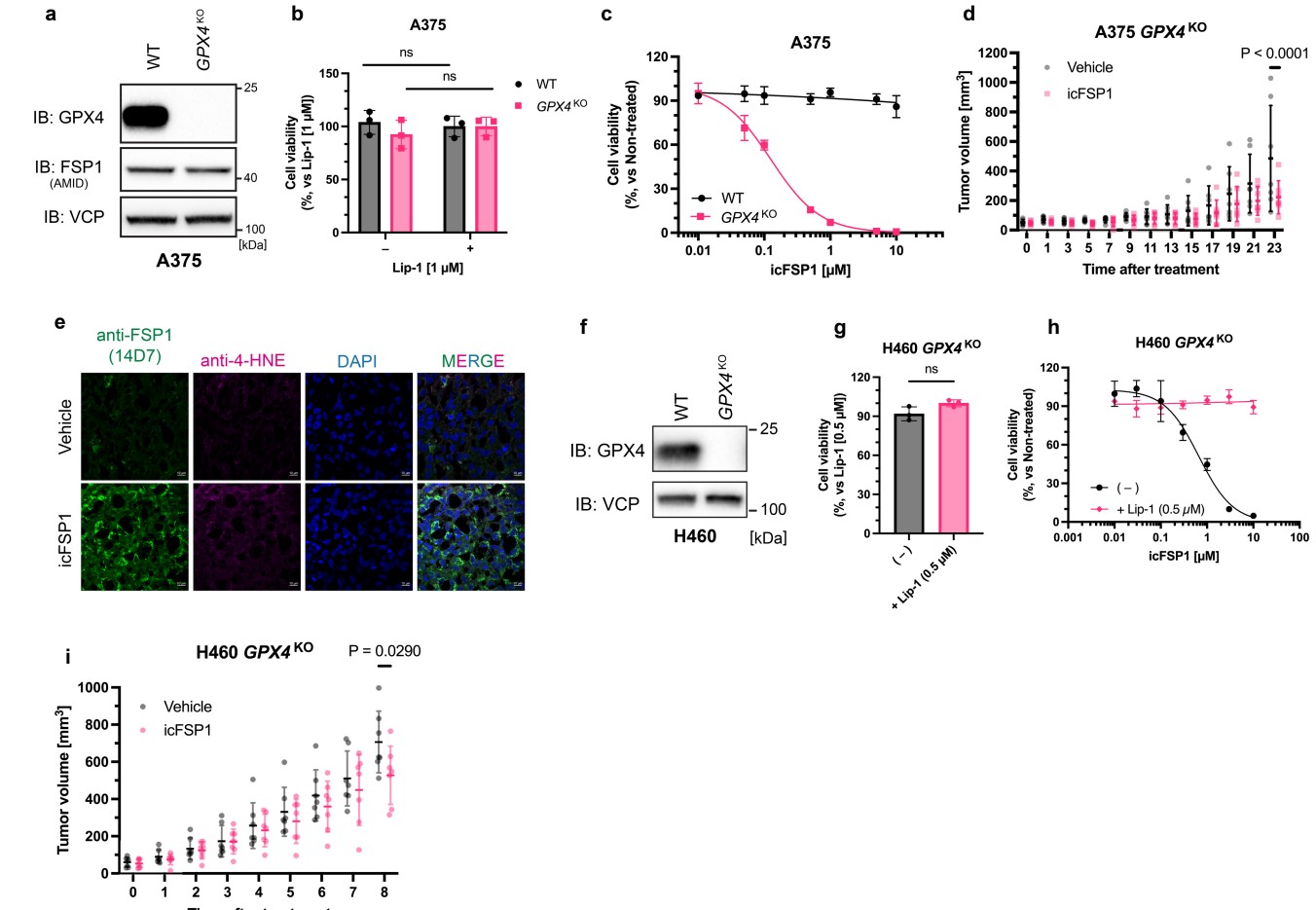

**Extended Data Fig. 9 | Targeting of FSP1 by icFSP1 as potential anti-cancer therapy using human cells. a**. Immunoblot analysis of GPX4, FSP1 and VCP expression of A375 WT and *GPX4*[KO] cells. Representative images from one out of 2 independent experiments. **b**. Cell viability was measured after Lip-1 withdrawal for 72 h in A375 WT and *GPX4*[KO] cells. Data represent the mean ± SD of 3 wells from one out of 2 independent experiments. **c**. Cell viability was measured after treating A375 WT and *GPX4*[KO] cells with icFSP1 for 48 h. Data represents the mean ± SD of 3 wells from one out of 2 independent experiments. **d**. icFSP1 inhibits tumor growth *in vivo*. A375 *GPX4*[KO] cells (a human melanoma cell line) were subcutaneously implanted into the flanks of Athymic Nude mice. After tumors became palpable (day 3), mice were randomized and treatment was started with icFSP1 (50 mg/kg i.p. twice a day for beginning 4 days and once a day afterward, n = 7) or vehicle (n = 7). Data represents the mean ± SD from one experiment. P value was calculated by two-way ANOVA followed by Bonferroni's multiple comparison. **e**. At the end of the *in vivo* pharmacological studies, tumors were dissected, cryosectioned and stained with anti-FSP1 (14D7) to visualize hFSP1 and with anti-4-HNE to visualize the lipid peroxidation breakdown product. Representative confocal microscopy images of 3 different samples from a single experiment are shown. **f**. Immunoblot analysis of GPX4, and VCP expression of H460 WT and *GPX4*[KO] cells from one out of 2 independent experiments. **g**. Cell viability was measured after Lip-1 withdrawal for 72 h in H460 WT and *GPX4*[KO] cells. Data represents the mean ± SD of 3 wells from one out of 2 independent experiments. Two-tailed unpaired t-test. **h**. Cell viability was measured after treating H460 WT and *GPX4*[KO] cells with icFSP1 for 48 h. Data represents the mean ± SD of 3 wells from one out of 2 independent experiments. **i**. icFSP1 inhibits tumor growth *in vivo*. H460 *GPX4*[KO] cells (a human lung cancer cell line) were subcutaneously implanted into the flanks of Athymic Nude mice. After tumors became palpable (day 8), mice were randomized and treatment was started with icFSP1 (50 mg/kg i.p. twice a day, n = 7) or vehicle (n = 7). Data represents the mean ± SD from one experiment. P value was calculated by two-way ANOVA followed by Sidak's multiple comparison tests.

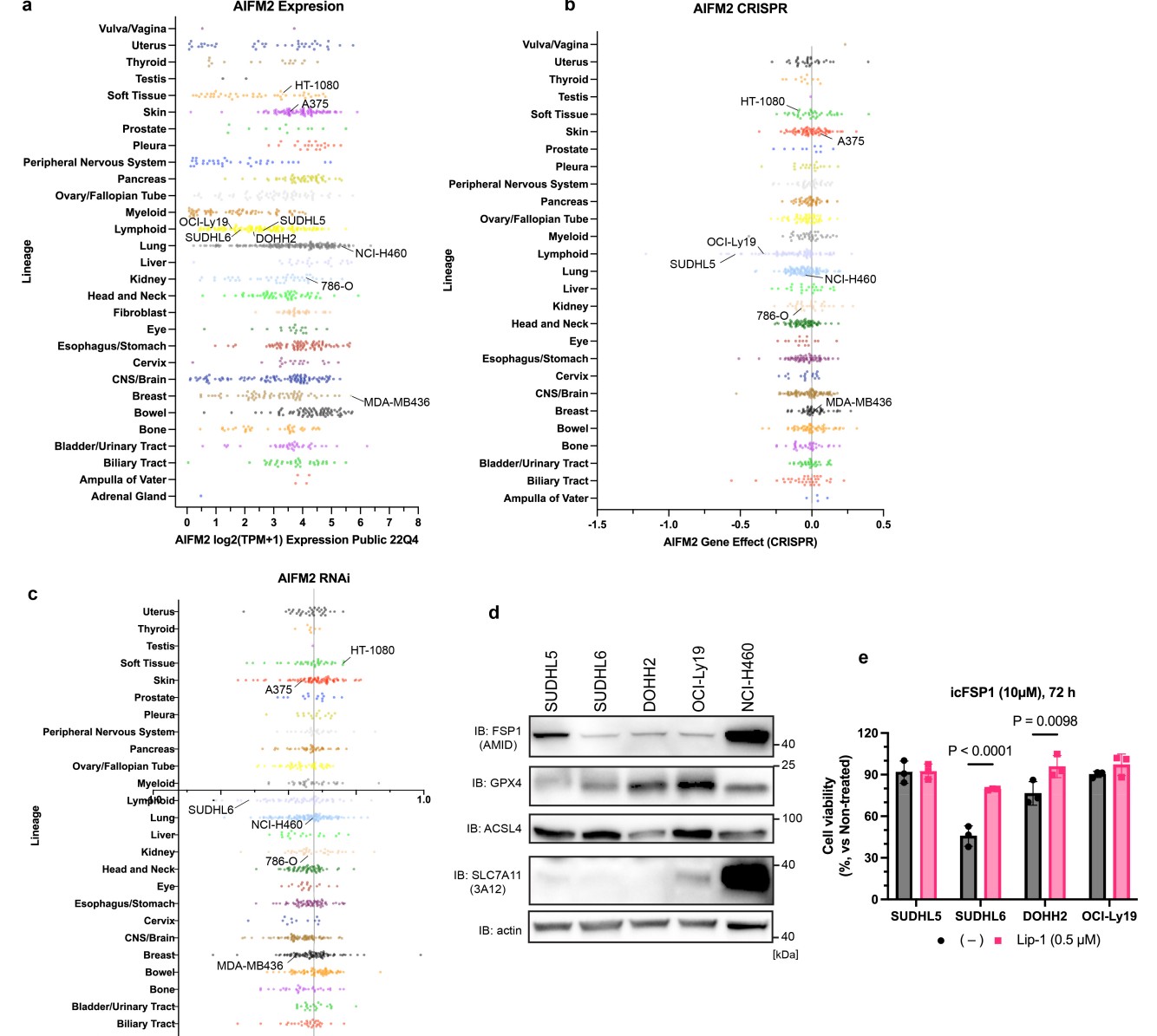

**Extended Data Fig. 10 | FSP1 is a potential target in multiple cancer cells.**
**a**. AIFM2 (FSP1) expression in different cancer cells from different origins.
**b**. AIFM2 gene effect (CRISPR) in different cancer cells from different origins.
**c**. AIFM2 gene effect (RNAi) in different cancer cells from different origins.
Data were mined from https://depmap.org/portal/ and cell lines used in this
study were highlighted with the names of cancer cell lines. **d**. Representative

immunoblot analysis of GPX4, FSP1, ACSL4, XCT (SLC7A11) and actin expression
of indicated cell lines from one out of 2 independent experiments. **e**. Cell viability
in lymphoma (SUDHL5, SUDHL6, DOHH2, OCI-Ly19) cells treated with icFSP1 in
the presence or absence of 0.5 μM Lip-1 for 72 h. Data represents the mean ± SD
of 3 wells from one out of 2 independent experiments. P values were calculated
by two-way ANOVA followed by Tukey's multiple comparison tests.

# Reporting Summary

## Statistics

For all statistical analyses, confirm that the following items are present in the figure legend, table legend, main text, or Methods section.

| n/a | Confirmed | |
|---|---|---|
| ☐ | ☒ | The exact sample size (*n*) for each experimental group/condition, given as a discrete number and unit of measurement |
| ☐ | ☒ | A statement on whether measurements were taken from distinct samples or whether the same sample was measured repeatedly |
| ☐ | ☒ | The statistical test(s) used AND whether they are one- or two-sided *Only common tests should be described solely by name; describe more complex techniques in the Methods section.* |
| ☒ | ☐ | A description of all covariates tested |
| ☐ | ☒ | A description of any assumptions or corrections, such as tests of normality and adjustment for multiple comparisons |
| ☐ | ☒ | A full description of the statistical parameters including central tendency (e.g. means) or other basic estimates (e.g. regression coefficient) AND variation (e.g. standard deviation) or associated estimates of uncertainty (e.g. confidence intervals) |
| ☐ | ☒ | For null hypothesis testing, the test statistic (e.g. *F*, *t*, *r*) with confidence intervals, effect sizes, degrees of freedom and *P* value noted *Give P values as exact values whenever suitable.* |
| ☒ | ☐ | For Bayesian analysis, information on the choice of priors and Markov chain Monte Carlo settings |
| ☒ | ☐ | For hierarchical and complex designs, identification of the appropriate level for tests and full reporting of outcomes |
| ☒ | ☐ | Estimates of effect sizes (e.g. Cohen's *d*, Pearson's *r*), indicating how they were calculated |

*Our web collection on statistics for biologists contains articles on many of the points above.*

## Software and code

Policy information about availability of computer code

| Data collection | CytExpert v2.4 (Beckman Coulter), Eve v1.8.2 (Nanolive), Image Lab v6.0 (Biorad), SoftMax Pro v7 (Molecular Devices), VisiView v4.0 (Visitron System), ZEN Black v2.3 (ZWISS). |
|---|---|
| Data analysis | CellProfiler v4.1.3 (Broad Institute), Flow Jo v10 software (Treestar, Inc), GraphPad Prism v9 (GraphPad Software), Image J v1.52 &1.53 (NIH), Masslynx V4.2 (Waters), MetaboAnalyst online platform v5.0 (PMID 31756036), ZEN Blue v3.2 (ZWISS), Topspin 4.0.6 (Bruker). |

For manuscripts utilizing custom algorithms or software that are central to the research but not yet described in published literature, software must be made available to editors and reviewers. We strongly encourage code deposition in a community repository (e.g. GitHub). See the Nature Portfolio guidelines for submitting code & software for further information.

## Data

Policy information about availability of data

All manuscripts must include a data availability statement. This statement should provide the following information, where applicable:

- Accession codes, unique identifiers, or web links for publicly available datasets
- A description of any restrictions on data availability
- For clinical datasets or third party data, please ensure that the statement adheres to our policy

All data are available in the Article and the Supplementary Information, and from the corresponding author on reasonable request. Gel source images are shown in

Supplementary Fig. 2. All source data are provided in this paper. Cancer cell line data were mined from https://depmap.org/portal/. The prediction for phase separation of FSP1 was conducted by https://iupred2a.elte.hu and https://mobidb.bio.unipd.it.

# Research involving human participants, their data, or biological material

Policy information about studies with human participants or human data. See also policy information about sex, gender (identity/presentation), and sexual orientation and race, ethnicity and racism.

| | |
|---|---|
| Reporting on sex and gender | N/A |
| Reporting on race, ethnicity, or other socially relevant groupings | N/A |
| Population characteristics | N/A |
| Recruitment | N/A |
| Ethics oversight | N/A |

Note that full information on the approval of the study protocol must also be provided in the manuscript.

# Field-specific reporting

Please select the one below that is the best fit for your research. If you are not sure, read the appropriate sections before making your selection.

☒ Life sciences   ☐ Behavioural & social sciences   ☐ Ecological, evolutionary & environmental sciences

For a reference copy of the document with all sections, see nature.com/documents/nr-reporting-summary-flat.pdf

# Life sciences study design

All studies must disclose on these points even when the disclosure is negative.

| | |
|---|---|
| Sample size | For in vitro experiments, sample sizes were determined based on previous similar studies that have given statistically significant results (PMID: 31634899, 35922516). The number of animals studied per treatment group was determined based on our preliminary data and previous similar studies that have given statistically significant results (PMID: 35922516 and 25402683), and respects the limited use of animal models in line with the 3R recommendations: Replacement, Reduction, Refinement. |
| Data exclusions | No data exclusions. |
| Replication | The experimental findings were reproduced as validated by at least three independent experiment in Fig 1c-f, Fig 2b-e, Fig 3a-e, Fig 4b-c and e, Extended Fig 2a-c and i, Extended Fig 3a-k; and at least two independent experiments in Fig 1b and g, Fig 4f-h, Extended Fig 1b-i,k and l, Extended Fig 2d-h and j-l, Extended Fig 3l-p, Extended Fig 4a-h, Extended Fig 5, Extended Fig 6, Extended Fig 7c and e-h, Extended Fig 8c-j, Extended Fig 9a-c and f-h, and Extended Fig 10d-e. |
| Randomization | For animal studies, mice were randomized into separate cages. For in vitro studies, randomization is not relevant since cells come in millions of populations and are automatically randomized and seeded to different wells for treatment. |
| Blinding | For animal study, mice were given a number prior to data collection and analysis. Data was collected and analyzed blindly. For in vitro experiments, investigators were not blinded, as standard in this manner of study, which contained multiple steps requiring distinct operations for accuracy and precision precluding blinding to experimental variables. |

# Reporting for specific materials, systems and methods

We require information from authors about some types of materials, experimental systems and methods used in many studies. Here, indicate whether each material, system or method listed is relevant to your study. If you are not sure if a list item applies to your research, read the appropriate section before selecting a response.

## Materials & experimental systems

| n/a | Involved in the study |
|-----|-----------------------|
| ☐ | ☒ Antibodies |
| ☐ | ☒ Eukaryotic cell lines |
| ☒ | ☐ Palaeontology and archaeology |
| ☐ | ☒ Animals and other organisms |
| ☒ | ☐ Clinical data |
| ☒ | ☐ Dual use research of concern |
| ☒ | ☐ Plants |

## Methods

| n/a | Involved in the study |
|-----|-----------------------|
| ☒ | ☐ ChIP-seq |
| ☐ | ☒ Flow cytometry |
| ☒ | ☐ MRI-based neuroimaging |

# Antibodies

**Antibodies used**

GPX4 (1:1000 for WB, ab125066, Abcam), 4-HNE (0.5 g/mL for IF, MHN-20P, JaICA), human FSP1 (1:1000, sc-377120, Santa Cruz Biotechnology), human FSP1(1:5 for WB, clone AIFM2 6D8, rat IgG2a, developed in-house: available from Sigma, Cat#MABC1638-25UL), mouse FSP1 (1:100 for WB, clone AIFM2 1A1 rat IgG2a, developed in-house), mouse FSP1(1:5 for WB and undilute or 1:1 for ICC and IF, clone AIFM2 14D7 IgG2b, developed in-house), human SLC7A11 (1:10; Rat IgG2a monoclonal antibody against an N-terminal peptide of hXCT, clone 3A12-1-1, developed in-house) mouse SLC7A11 (1:1000, Cell Signaling Technology, Cat#98051S), ACSL4 (1:1000, clone A-5, Santa Cruz, Cat#sc-271800), β-actin-HRP (1:50000, A3854, Sigma-Aldrich), valosin containing protein (VCP, 1:10000, ab11433 or ab109240, Abcam), Calnexin (1:100 for ICC, ab22595, Abcam), GM130 (1:100 for ICC, Clone: EP892Y, ab52649, Abcam), EEA1 (1:100 for ICC, Clone: C45B10, 3288S, Cell SignalingTechnology), and LAMP1 (1:100 for ICC, Clone: H4A3, sc20011, Santa Cruz Biotechnology), HA tag (YPYDVPDYA, 1:1000 for WB and 1:10 for IF, clone 3F10 rat IgG1, developed in-house), FLAG tag (DYKDDDDK, 1:5000, 2368S, Cell Signaling Technology), Goat anti-Rat Alexa Fluor 488 IgG (H+L) (1:500, A-11006, Invitrogen), Goat Anti-Mouse IgG H&L Alexa Fluor 647 (1:500, ab150115, Abcam), Donkey anti-rat IgG alexa fluor 555 (1:500, ab150154, Abcam) were used in this study.

**Validation**

GPX4 (ab125066), VCP (ab11433 or ab109240), Human FSP1 (sc-377120), HA tag (clone 3F10), FLAG tag (2368S), β-actin-HRP (A3854), ACSL4 (sc-271800), human SLC7A11 (clone 3A12) antibodies were validated for WB using mouse and human cells samples in previous publications (PMID: 35922516, 31634899, and 27842070).
4-HNE (MHN-20P) antibody was validated for IHC using mouse samples in a previous publication (PMID: 35922516) and for IF in this study in Extended Fig 8d,h.
FSP1 antibody (clone AIFM2 1A1/6D8 rat IgG2a, and clone AIFM2 14D7 IgG2b, developed in-house) has been validated for WB in previous study (PMID: 35922516), and for ICC/IF in this study in Extended Fig 4d.
Calnexin antibody (ab22595) was validated for ICC using human and mouse cell samples on Cell Signaling Technology website.
GM130 antibody (ab52649) was validated for WB using mouse embryonic fibroblasts on Abcam website and for ICC in this study in Extended Fig 4b.
mouse xCT/SLC7A11 antibody (98051S) was validated for WB using mouse cell samples on Cell Signaling Technology website.
EEA1 antibody (3288S) was validated for ICC using human and mouse cell samples on Cell Signaling Technology website.
LAMP1 antibody (sc20011) was validated for ICC using human and mouse cell samples on Santa Cruz Biotechnology website.
Secondary antibodies were validated for ICC/IF on each provider on website (Cell Signaling Technology, Invitrogen, Abcam).

# Eukaryotic cell lines

Policy information about cell lines and Sex and Gender in Research

**Cell line source(s)**

4-OH-TAM-inducible Gpx4-/- murine immortalized fibroblasts (Pfa1) were reported previously (PMID: 18762024). HT-1080 (CCL-121), 786-O (CRL-1932), A375 (CRL-1619), B16F10 (CRL-6475), HEK293T (CRL-3216), HT-29 (HTB-38), NCI-H460 (HTB-177), MDA-MB-436 (HTB-130), and 4T1 (CRL-2539) cells were obtained from ATCC. THP-1 cells were obtained from DSMZ (Germany). Human PBMC cells were purchased from Tebu-bio (Cat#088SER-PBMC-F). SUDHL5, SUDHL6, DOHH2, and OCI-Ly19 cells (available from DSMZ) were kindly gifted from Dr. Stephan Hailfinger. Rat1 cells (available from Thermo Fisher) were kindly gifted from Medizinische Hochschule Hannover.

**Authentication**

None of the cell lines used were authenticated.

**Mycoplasma contamination**

All cell lines were tested negative for mycoplasma contamination.

**Commonly misidentified lines**
(See ICLAC register)

No commonly misidentified cell lines were used.

# Animals and other research organisms

Policy information about studies involving animals; ARRIVE guidelines recommended for reporting animal research, and Sex and Gender in Research

**Laboratory animals**

Five to six-weeks-old female C57BL6/J and athymic nude mice were obtained from Charles River and six to seven weeks-old mice were used for experiments. All mice were kept under standard conditions with water and food ad libitum and in a controlled environment (22 ± 2°C, 55 ± 5% humidity, 12 h light/dark cycle) in Helmholtz Munich animal facility under SPF-IVC standard conditions.

| | |
|---|---|
| Wild animals | The study did not involve wild animals. |
| Reporting on sex | Experiments were performed on only female mice. |
| Field-collected samples | The study did not involve field-collected samples. |
| Ethics oversight | All experiments were performed in compliance with the German Animal Welfare Law and have been approved by the institutional committee on animal experimentation and the government of Upper Bavaria (ROB-55.2-2532.Vet_02-17-167). |

Note that full information on the approval of the study protocol must also be provided in the manuscript.

# Flow Cytometry

## Plots

Confirm that:

☒ The axis labels state the marker and fluorochrome used (e.g. CD4-FITC).

☒ The axis scales are clearly visible. Include numbers along axes only for bottom left plot of group (a 'group' is an analysis of identical markers).

☒ All plots are contour plots with outliers or pseudocolor plots.

☒ A numerical value for number of cells or percentage (with statistics) is provided.

## Methodology

| | |
|---|---|
| Sample preparation | 100,000 cells per well were seeded on a 12-well plate one day prior to the experiments. On the next day, cells were treated with 2.5 µM icFSP1 for 3 h, and then incubated with 1.5 µM C11-BODIPY 581/591 (Invitrogen, Cat#D3861) for 30 min in a 5% $CO_2$ atmosphere at 37°C. Subsequently, cells were washed by PBS once and trypsinized, and then resuspended in 500 µL PBS. Cells were passed through a 40 m cell strainer and analyzed by a flow cytometer (CytoFLEX, Beckman Coulter) with a 488-nm laser for excitation. Data was collected from the FITC detector (for the oxidized form of BODIPY) with a 525/40nm bandpass filter and from the PE detector (for the reduced form of BODIPY) with a 585/42 nm bandpass filter. |
| Instrument | CytoFLEX (Beckman Coulter) |
| Software | CytExpert v2.4 was used for data collection. FlowJo v10 was used for data analysis. |
| Cell population abundance | At least 10,000 cells were analyzed for each sample. |
| Gating strategy | Cell populations were separated from cellular debris using FSC and SSC. |

☒ Tick this box to confirm that a figure exemplifying the gating strategy is provided in the Supplementary Information.

