## [Peer Review File · Nature]

Manuscript Title: Phase separation of FSP1 promotes ferroptosis

Reviewer Comments & Author Rebuttals

Reviewer Reports on the Initial Version:

Referees' comments:

Referee #1 (Remarks to the Author):

A. Summary of the key results

In their manuscript "Liquid-liquid phase separation of FSP1 triggers ferroptosis", Nakamura et al. present their discovery of the first ferroptosis suppressor protein-1 (icFSP1) inhibitor with potential for in vivo efficacy in a comprehensive addition to their prior Nature 2019 work where they characterized the first FSP1 inhibitor (iFSP1). icFSP1 does not inhibit the FSP1 competitively (as did iFSP1), but works in synergy with GPX4 inhibition to induce subcellular re-localization from the membrane and condensation of FSP1. This finding is extremely exciting for the field of ferroptosis research, especially considering recent studies which show that FSP1, along with extramitochondrial ubiquinone or vitamin K and NAD(P)H/H⁺ as an electron donor, is the second ferroptosis checkpoint. Thus, the discovery of an on-target FSP1 inhibitor opens the field to exploring the therapeutic potential of FSP1 as a ferroptosis-inducer. Furthermore, the authors convincingly show proof-of-concept that icFSP1 is capable of potentiating ferroptosis, impairing tumor growth, and inducing FSP1 condensates in tumor in vivo.

B. Originality and significance: if not novel, please include reference

This novel work describes the first ferroptosis suppressor protein-1 (FSP1) inhibitor that works in synergy with GPX4 inhibition to induce subcellular re-localization from the membrane and condensation of FSP1. This work fills an important gap in the field as the first FSP1 inhibitor cannot be used as an anticancer drug because of its unfavorable structure and substitution pattern, and icFSP has clinical potential as it addresses those medicinal chemistry structural issues. Furthermore, their work demonstrates how liquid-liquid phase separation is capable of modulating biological activity in vivo. These findings are highly novel, and have not been demonstrated before. This work has relevance to many cancers and neurodegenerative disorders, which may see therapeutic response to FSP1 inhibition.

The authors take care to distinguish the originality and significance in the context of their Nature 2019 work identifying the first FSP1 inhibitor. This distinction is important as it is only the author's prior work of the first FSP1 inhibitor that is capable of diminishing their finding of the first FSP1 inhibitor with therapeutic potential. The authors expertly add additional novelty to this current work by demonstrating that the mechanism of the new FSP1 inhibitor acts in a novel way by liquid-liquid phase separation. A further suggestion to emphasize the originality of this work as compared to the Nature 2019 work would be to include a few sentences (on p.3) how the hit validation studies of the

same 10,000 drug-like small molecule screen from your Nature 2019 differ from the top hit validation studies performed in that work. Were different metrics used to screen for chemicals with favorable structure and substitution patterns for potential medicinal chemistry development? Adding further emphasize on the novelty that this work is the first evidence for a LLPS-targetable process in ferroptosis should also be considered.

I recommend publication following the revisions listed below.

C. Data & methodology: validity of approach, quality of data, quality of presentation

The quality of the data and presentation is strong, and the approaches the authors use to support their conclusions throughout the work are appropriate and valid.

Suggestions for improvement:

- Fig. 1e – it is difficult to see the replicate dots in the black coloration – please adjust color
- Extended Fig. 3 – to facilitate ease of reading, please break up current panel a and b into multiple panels

D. Appropriate use of statistics and treatment of uncertainties

The statistical tests performed in this work and treatment of uncertainties are appropriate.

Suggestions for improvement:

- Fig. 1e – please show lower limit of error bars
- Fig. 2b – please include error bars on iFSP1 experiment

E. Conclusions: robustness, validity, reliability

The conclusions the authors make based on their data presented in this manuscript are robust, valid, and reliable. The authors appropriately interpret their data and do not make overly ambitious statements based on the results.

F. Suggested improvements: experiments, data for possible revision

Major comments:

1.) Does icFSP1 increase LDH release and inhibit lipid ROS to the same extent as iFSP? Based on the finding icFSP alone (without co-treatment with canonical ferroptosis inducers) was capable of inducing ferroptosis it is an important observation for the field to know the extent to which icFSP1 decreases lipid ROS compared to other FSP1 inhibitors. A suggestion to address this would be to include Fig. 1e, 1f as a comparison with both iFSP and icFSP1.

2.) From your Nature 2019 work, the cell lines you studied in Extended Data Figure 7 demonstrate varying levels of FSP1 expression - from high to low expression: NCI-H460, 786-O, A375, MDA-MB-436. Except for NCI-H460, it seems like baseline FSP1 expression may have correlation with the extent to which you are observing decreased cell viability with icFSP1. It would be helpful to include

a figure for the cell lines shown in Extended Fig 1c showing immunoblot analysis of the expression of key ferroptosis players (ACSL4, FSP1, GPX4, XCT) to help interpret the differential responses of the cell lines with icFSP1. Is there a way to help predict which cancers might see response with icFSP1?

3.) Since the in vivo models were in a mouse background (where icFSP1 would not be expected have an effect as it appears to be specific to human), it is difficult to discern the specificity of icFSP1 to cancer cells especially in the in vivo context. It would be helpful to include viability experiments from few non-cancerous cell lines to determine if the icFSP1 would be specific to inhibiting cancer cells, or alternatively if there a role for icFSP1 in non-cancerous cells?

4.) Further to the point in #3, FSP1 appears to be highly expressed in bone marrow and other tissues, it will be important to speculate on the off-target effects of this drug which would need to be considered in a clinical setting. Inclusion of data or discussion to contribute to this point would greatly increase the impact of this work.

5.) In light of the author's recent publications, to what extent is icFSP v iFSP acting via ubiquinone versus vitamin E/K? As shown in Doll et al., Nature 2019, CoQ10 is a prime substrate for FSP1. Can icFSP1, as an in vivo inhibitor of FSP1, help to shed light on this preference?

Minor comments

1.) Please clarify why the human cell line experiments in Extended Data Fig. 1c were carried out for 72 hours, whereas the mouse/rat line experiments in Extended Data Fig. 2a-c were carried out for 48 hours. Do you see any changes in viability at 72 hours?

2.) Additionally in Extended Data Fig. 3d, the immunoblots were performed at 48 hours, would you expect differences in expression at 72 hours where you observed viability differences?

3.) It might be helpful to include a panel for the cell lines shown in Extended Fig 2a-c showing immunoblot analysis of the expression of key ferroptosis players (ACSL4, FSP1, GPX4, XCT) to help interpret the differential responses of the cell lines with icFSP1 as compared to the human cell lines.

4.) Pg. 7 – consider adding a few sentences explaining the relevance of LLPS in cancer and neurodegeneration

5.) Cell lysis and immunoblotting – please double check that the cells were centrifuged for 1 hour as stated in methods

G. References: appropriate credit to previous work?

Authors appropriately provide credit and citations to previous work.

H. Clarity and context: lucidity of abstract/summary, appropriateness of abstract, introduction and conclusions

The abstract is clear and concise and is an appropriate representation of the introductions and conclusions provided in this manuscript. Adding even further emphasize in the abstract on the novelty that this work is the first evidence for a LLPS-targetable process in ferroptosis should also be considered.

Referee #2 (Remarks to the Author):

Ferroptosis is emerging as an attractive pathway for achieving targeted cell killing in various pathologies, including treatment-resistant cancers. Meanwhile, LLPS is gaining attention as an important molecular activity whose precise regulation is critical for cellular function. Here, Nakamura and colleagues highlight the intersection of these two processes by identifying a small-molecule inhibitor of the ferroptosis checkpoint enzyme FSP1 (icFSP1) that acts not by altering FSP1 catalytic activity but by inducing FSP1 phase separation. The authors demonstrate that icFSP1 promotes ferroptotic cell death in various cancer cell lines and that cellular FSP1 relocalizes into discrete cytosolic puncta within minutes of icFSP1 application. Through both in vitro and cellular characterization, they also demonstrate that FSP1 LLPS is dependent on N- and C-terminal IDRs, an N-terminal LC region, and myristoylation. Finally, Nakamura et al. show that icFSP1 treatment induces ferroptotic death in tumor cells in vivo and can suppress tumor growth in mouse xenograft models. This is an exciting study that should be of great interest to the scientific community. Nevertheless, the results raise several questions that should be addressed to improve the overall rigor of the study and better support some of the authors' conclusions.

- 1.) Early in the text (top of page 6), the authors claim that the icFSP1-triggered relocalization of FSP1 via phase separation "causes lipid peroxidation and ferroptosis". While they go on to persuasively show that icFSP1 causes both ferroptosis and FSP1 phase separation, including the formation of FSP1 condensates in implanted tumor cells, the evidence of causal link between the two phenomena rather thin. Such a claim would be better supported by repeating some of the functional assays in cells and in vivo using the LLPS-defective FSP1 mutants described in Fig. 4a.
- 2.) The authors note that unlike the other cell types tested, icFSP1 could trigger cell death in HT-1080 cells even without co-application of a ferroptosis inducer. Why do these cells behave differently? More generally, what is the significance of the cancer cell-type specific variations in icFSP1-induced responses observed by the authors? It is unclear if icFSP1 will produce consistent effects across different cancer cell types, particularly given that the in vivo studies used only one cancer cell line.
- 3.) The apparent species-selectivity (i.e., human vs mouse) of icFSP1 is curious. Have the authors verified that icFSP1 binds to mouse Fsp1? For the three mutations (S187C, L217R, Q319K) shown to disrupt icFSP1-induced phase separation of human FSP1, have the authors tested whether the opposite mutations (C187S, R217L, K319Q) sensitize mouse Fsp1 to icFSP1? Furthermore, given the authors' speculation that the specific residues in FSP1 may be involved in protein-membrane interactions, how does this impact the function of mouse Fsp1? Does the mouse enzyme use a different mechanism?
- 4.) How were the in vitro condensates characterized as being solid-like? Was this based entirely on sedimentation assays? The authors should perform in vitro FRAP analyses to more rigorously

investigate this point.

5.) The authors claim that icFSP1 did not promote FSP1 condensate formation in vitro. However, the authors only performed this assay at a single concentration of purified FSP1. To better support their conclusion, the authors should test different FSP1 concentrations to determine whether icFSP1 can shift the phase diagram of FSP1 LLPS.

6.) Similarly, why is FSP1 myristoylation required for phase separation in cells but not in vitro? The authors should at least compare the phase diagrams of FSP1 with and without myristoylation. Based on the literature cited and the schematic presented in Fig. 4i, myristoylation should be expected to lower the concentration threshold for FSP1 phase separation, if not also produce more liquid-like condensates.

Referee #3 (Remarks to the Author):

In their study “liquid-liquid phase separation of FSP1 triggers ferroptosis”, Nakamura et al. identified a novel FSP1 inhibitor, termed icFSP1, and characterized its mechanism of action. In contrast to FSP1, they find that icFSP1 does not inhibit the catalytic activity of FSP1 but rather mislocalizes FSP1 into lipid condensates. The identification of a novel FSP1 inhibitor will be an important tool for the ferroptosis field. However, there are notable limitations of this study. First, almost all experiments rely on FSP1 overexpressed to very high levels and the similar behavior of endogenous protein is not well demonstrated. Second, the in vivo evidence for superiority of icFSP1 is not well established, nor are the on-target effects sufficiently shown in vivo. With revisions, this manuscript will likely be of interest to the chemical biology community and a more specialized journal should be considered.

Major points

1. Almost all experiments within this study are performed with overexpressed FSP1, including fluorescence tagged versions. The degree of overexpression is particularly noticeable in Extended Data figure 2D, where the endogenous FSP1 levels are several orders of magnitude lower than the ectopic. Aggregation and/or mislocalization is a known problem with overexpressed proteins.
2. While H460 were used to show condensation of endogenous FSP1, it is unclear whether the few punctae observed are actual condensates. The authors should consider knocking in a tag and the Q319K mutant into the endogenous FSP1 locus to definitively demonstrate condensation of endogenous FSP1.
3. The authors argue that icFSP1 is superior for in vivo use compared to iFSP1, but the data do not support improved stability of icFSP1 in vivo. It appears that higher plasma concentrations are achieved with icFSP1 compared to iFSP1, but the decay kinetics look similar. Additional evidence is needed to support this claim. Moreover, the microsomal stability experiments (Extended data figure 8) show a similar stability in human microsomes when comparing icFSP1 and iFSP1.
4. The authors elegantly identify a Q319K mutant of FSP1 that cannot be localized to condensates and show in vitro that this mutant is resistant to the effects of icFSP1. It is unclear why this mutant was used to show the ability of icFSP1 to induce condensates in vivo (Figure 4h) but was not used in experiments testing the ability of icFSP1 to impair tumor growth and promote lipid peroxidation in vivo. These experiments should also be performed.

Other points

1. The second sentence in the abstract is misleading, as the recent paper by the Conrad group did not demonstrate that endogenous vitamin K contributes to ferroptosis suppression by FSP1.

Referee #4 (Remarks to the Author):

The central result in this work is that inhibition of the ferroptosis suppressor protein by a small molecule involves the induction of condensate formation. This is a novel and exciting mechanism of action, suggesting that instead of directly suppressing the enzymatic activity of FSP-1, the small molecule icFSP1 acts as a ligand that drives FSP1 phase separation. If this result stands up to continued scrutiny, and the data suggest that it might, this could provide an entirely new route for identifying condensate modulators through ligands that drive condensate formation, assuming the condensate inactivates the function that one seeks to inactivate. For me, this was one of the most exciting reads of the year. I see one important issue that needs work and scrutiny, and this pertains to the issue of how icFSP1 drives the condensate formation of FSP1. The in vitro investigations are really important, but they are incomplete / inconclusive in terms of provide the full picture.

Ligands modulate the driving forces for phase separation through a mechanism known as polyphasic linkage. This was articulated by Wyman and Gill in 1980. Please see: <https://doi.org/10.1073/pnas.77.9.5239>. The concept has been applied in recent work to study ligand modulation of phase behavior, and rules have emerged regarding the types of ligands that drive vs. destabilize phase separation. Please see: <https://doi.org/10.1073/pnas.2017184118> and <https://doi.org/10.1063/5.0050059>. Why is this important? The drivers of phase transitions are all exemplars of associative macromolecules featuring stickers and spacers. Ligands modulate the phase transitions of these systems through site-specific interactions with the driver macromolecules, and these interactions are invariably different across the phase boundary. This gives rise to the concept of preferential binding, and the fact that there is a dose-dependent enhancement in the driving forces for hFSP1 phase separation in the presence of icFSP1 implies that the small molecule, as a ligand, binds preferentially to the protein in its dense phase. This leads to the question of how this preferential binding comes about. The papers cited above lay out rules for how this can come about. Deciphering the relevant rules will require further experiments. I think the myristoylation adds a multivalent sticker and that the icFSP1 molecule enables physical crosslinking of these moieties. However, there likely are conformational changes as well, suggestive of a myristoylation induced partial unfolding that is being aided in the dense phase. A scenario for how this can happen was explained recently by unfolding of model proteins in live cells. Please see: <https://doi.org/10.1016/j.molcel.2022.06.024>. So, there are two possible revisions to suggest here: First, reanalyze the extant data using the polyphasic linkage formalism, and revise the MS accordingly. Or, second, find a way to follow the myristoyl groups and the small molecules in the dense phase, compare this to the dilute phase, and discern the molecular level MoA. I think the second is a very tall order, and given the exciting nature of the observation, and its timeliness, I

would advocate for the first option. In doing so, I would recommend against the assertion that this LLPS followed by solidification. In all likelihood, the authors are not observing an equilibrium transition. What they are observing is the aging of a viscoelastic material, the physics of which is still being sorted, and has very little to do with an equilibrium liquid-to-solid transition. In any case, one cannot argue for this type of a transition based on FRAP. Therefore, as recommended recently, it is best to abandon the adherence to LLPS, stick with phase separation, and revisit the data through the lens of polyphasic linkage. Please see: <https://doi.org/10.1016/j.molcel.2022.05.018>.

On a technical note, the use of PEG is a concern. It would help to see a titration of how the phase boundary shifts as a function of PEG concentration, lower concentrations are of particular interest. This way, one will get a sense of how strong the driving forces for phase separation are with this system.

On a second technical note, while the structure of hFSP1 may not be available, an AlphaFold prediction is available. It appears that, as expected, the IDRs overlap with regions of low confidence in prediction, but the LCR does not. Please see the structure here: <https://www.uniprot.org/uniprotkb/Q9BRQ8/entry>. This structure is suggestive of a couple of possibilities. It might be useful for a molecular level understanding of the MoA. It is also suggestive of possible oligomerization being coupled to myristoylation induced increased stabilization. Studies to date suggest that myristoylation significantly enhances protein stability. Please see: <https://doi.org/10.1073/pnas.1008026107>. This would suggest that the difference across the phase boundary is protein stability, and this in turn enables sticker-sticker interactions between the myristoyl groups and the small molecule.

Author Rebuttals to Initial Comments:

Nature manuscript 2022-08-13275

We thank all reviewers and the editor for the critical assessment of our manuscript and the highly appreciative comments made by the reviewers. Please find below our response to each comment on a point-by-point basis. Please see the changes of the text highlighted in this letter as underlined and in the revised manuscript as marked in red.

Referees' comments:

Referee #1 (Remarks to the Author):

A. Summary of the key results

In their manuscript "Liquid-liquid phase separation of FSP1 triggers ferroptosis", Nakamura et al. present their discovery of the first ferroptosis suppressor protein-1 (icFSP1) inhibitor with potential for in vivo efficacy in a comprehensive addition to their prior Nature 2019 work where they characterized the first FSP1 inhibitor (iFSP1). icFSP1 does not inhibit the FSP1 competitively (as did iFSP1), but works in synergy with GPX4 inhibition to induce subcellular re-localization from the membrane and condensation of FSP1. This finding is extremely exciting for the field of ferroptosis research, especially considering recent studies which show that FSP1, along with extramitochondrial ubiquinone or vitamin K and NAD(P)H/H⁺ as an electron donor, is the second ferroptosis checkpoint. Thus, the discovery of an on-target FSP1 inhibitor opens the field to exploring the therapeutic potential of FSP1 as a ferroptosis-inducer. Furthermore, the authors convincingly show proof-of-concept that icFSP1 is capable of potentiating ferroptosis, impairing tumor growth, and inducing FSP1 condensates in tumor in vivo.

We are most thankful for the highly appreciative comments and are very much pleased that the reviewer shares the enthusiasm for our work.

B. Originality and significance: if not novel, please include reference

This novel work describes the first ferroptosis suppressor protein-1 (FSP1) inhibitor that works in synergy with GPX4 inhibition to induce subcellular re-localization from the membrane and condensation of FSP1. This work fills an important gap in the field as the first FSP1 inhibitor cannot be used as an anticancer drug because of its unfavorable structure and substitution pattern, and icFSP has clinical potential as it addresses those medicinal chemistry structural issues. Furthermore, their work demonstrates how liquid-liquid phase separation is capable of modulating biological activity in vivo. These findings are highly novel, and have not been demonstrated before.

This work has relevance to many cancers and neurodegenerative disorders, which may see therapeutic response to FSP1 inhibition.

The authors take care to distinguish the originality and significance in the context of their Nature 2019 work identifying the first FSP1 inhibitor. This distinction is important as it is only the author's prior work of the first FSP1 inhibitor that is capable of diminishing their finding of the first FSP1 inhibitor with therapeutic potential. The authors expertly add additional novelty to this current work by demonstrating that the mechanism of the new FSP1 inhibitor acts in a novel way by liquid-liquid phase separation. A further suggestion to emphasize the originality of this work as compared to the Nature 2019 work would be to include a few sentences (on p.3) how the hit validation studies of the same 10,000 drug-like small molecule screen from your Nature 2019 differ from the top hit validation studies performed in that work. Were different metrics used to screen for chemicals with favorable structure and substitution patterns for potential medicinal chemistry development? Adding further emphasize on the novelty that this work is the first evidence for a LLPS-targetable process in ferroptosis should also be considered.

Thank you for raising this point: the basis for the present study was a follow-up detailed validation of new FSP1 inhibitor candidates based on the screening performed by Doll et al. Nature 2019. We first conducted cell-based assays to identify new FSP1 inhibitors and thereby found a reasonable set of candidates entailing both iFSP1 and icFSP1 in addition to some that have been recently reported by Olzmann's group (see a recent preprint: <https://doi.org/10.1101/2022.12.14.520445>).

Subsequent in-depth DMPK ("Drug Metabolism and Pharmacokinetics") analysis of these clusters identified icFSP1 as a highly promising *in vivo* applicable compound as most of the other compounds are not suitable for *in vivo* use largely due to insufficient metabolic stability (which by the way still remains a major issue for many of the reported ferroptosis inducing compounds targeting other nodes of the ferroptosis pathway). Moreover, studies into the mechanism-of-action (MoA) of icFSP1 as presented here revealed a striking difference between its cell-based and *in vitro* inhibitory activity towards FSP1, which clearly differed from that of iFSP1.

Previously, we performed a screening of $\approx 10,000$ drug-like small molecule compounds, yielding the FSP1 inhibitor iFSP1, as described in Doll et al. 2019. These initial hit validation studies were focused on the identification of highly efficacious compounds that inhibit FSP1 in cells, as well as on the enzyme level, omitting icFSP1, which clearly failed to meet these criteria. In our quest to identify *in vivo* applicable FSP1-specific inhibitors as potential future drugs to combat difficult-to-treat cancers displaying high vulnerability to ferroptosis (see works by Viswanathan et al. Nature 2017 [PMID: 28678785]; Hangauer et al Nature 2017 [PMID: 29088702]; Tsoi et al Cancer Cell 2018

[PMID: 29657129]), we re-evaluated our earlier screening results with emphasis on *in silico* drug-likeness parameters and potential for medicinal chemistry development (such as aromatic ring count, ease of synthesis, lipophilicity, ligand efficiency etc.). Additionally, we were highly intrigued by the finding that icFSP1 is a potent inhibitor of FSP1 in cells, while it does not block FSP1 protein activity, prompting us to further investigate its MoA.

We accordingly amended the text, which now reads as follows (Page 3):

To identify possible in vivo applicable FSP1-specific inhibitors as potential future drugs to combat difficult-to-treat cancers, we carefully re-validated hit compounds from ≈10,000 drug-like small molecule compounds (as described previously⁵), in terms of their potential for medicinal chemistry development using cheminformatics tools⁷ for the prediction of physicochemical properties and drug-likeness.

I recommend publication following the revisions listed below.

C. Data & methodology: validity of approach, quality of data, quality of presentation

The quality of the data and presentation is strong, and the approaches the authors use to support their conclusions throughout the work are appropriate and valid.

Suggestions for improvement:

- Fig. 1e – it is difficult to see the replicate dots in the black coloration – please adjust color.

We thank for the reviewers' suggestion and modified the figure accordingly (Fig. 1e):

- Extended Fig. 3 – to facilitate ease of reading, please break up current panel a and b into multiple panels

We modified the corresponding figures accordingly:

D. Appropriate use of statistics and treatment of uncertainties

The statistical tests performed in this work and treatment of uncertainties are appropriate.

Suggestions for improvement:

- Fig. 1e –please show lower limit of error bars

We thank the reviewer for the suggestion and modified the figure as shown below (Fig. 1e).

- Fig. 2b – please include error bars on iFSP1 experiment

We already included them in the original manuscript, but the error bars of iFSP1 are very small. We modified the figure and hope that the reviewer can appreciate them as such (Fig. 2b).

E. Conclusions: robustness, validity, reliability

The conclusions the authors make based on their data presented in this manuscript are robust, valid, and reliable. The authors appropriately interpret their data and do not make overly ambitious statements based on the results.

We are most grateful for the reviewers' positive evaluation.

F. Suggested improvements: experiments, data for possible revision

Major comments:

1.) Does icFSP1 increase LDH release and inhibit lipid ROS to the same extent as iFSP? Based on the finding icFSP alone (without co-treatment with canonical ferroptosis inducers) was capable of inducing ferroptosis it is an important observation for the field to know the extent to which icFSP1 decreases lipid ROS compared to other FSP1 inhibitors. A suggestion to address this would be to include Fig. 1e, 1f as a comparison with both iFSP and icFSP1.

Per the reviewer's suggestions, we extended the results of Fig. 1e and 1f in order to directly compare icFSP1 with iFSP1. icFSP1 can induce ferroptosis and lipid peroxidation equally well or even slightly better than iFSP1 at the same concentration (Extended Data Fig. 3j,k).

Regarding these additional results, we amended the main text as follows on page 5:

While hFSP1 enzymatic activity was inhibited by iFSP1 in a cell-free system as reported^{5,6}, icFSP1 failed to inhibit hFSP1 activity at the low micromolar range, albeit it clearly affected cell viability and lipid peroxidation of Pfa1 cells overexpressing hFSP1 (Fig. 2b, Extended Data Fig. 3a-k).

2.) From your Nature 2019 work, the cell lines you studied in Extended Data Figure 7 demonstrate varying levels of FSP1 expression - from high to low expression: NCI-H460, 786-O, A375, MDA-MB-436. Except for NCI-H460, it seems like baseline FSP1 expression may have correlation with the extent to which you are observing decreased cell viability with icFSP1. It would be helpful to include a figure for the cell lines shown in Extended Fig 1c showing immunoblot analysis of the expression of key ferroptosis players (ACSL4, FSP1, GPX4, XCT) to help interpret the differential responses of the cell lines with icFSP1. Is there a way to help predict which cancers might see response with icFSP1?

Thanks for the suggestion. We now include an immunoblot analysis of the expression of key ferroptosis players (ACSL4, FSP1, GPX4, XCT) in the different cell lines in Extended Data Fig. 1e.

In terms of predicting the sensitivity of icFSP1 or FSP1 dependency, we also included an analysis derived from the Depmap portal (<https://depmap.org/portal/>) in the revised manuscript. From this data, one might be able to deduce that tumor cells with robust expression of FSP1 can be a candidate cell line for icFSP1 in synergism with ferroptosis inhibitors targeting the XCT/GPX4/GSH axis. Besides, when the gene effect score (CRISPR or RNAi) of FSP1 is small (i.e., a clear negative value), these cancer cells might be potentially sensitive to icFSP1 alone, as now shown in Extended Data Fig. 10. For instance, HT-1080 cells display a negative CRISPR value meaning that the KO of FSP1 can reduce cell viability (as shown in Extended Data Fig. 10a), and as is the case for HT-1080 cells as shown in Fig. 1d.

Extended Fig. 10

In this respect, we further investigated whether icFSP1 treatment alone might induce ferroptosis in some human B cell lymphoma cell lines (i.e., SUDHL-5, SUDHL-6, DOHH2, and OCI-Ly19), which have negative gene effect scores of CRISPR or RNAi.

While these cell lines were not as sensitive to icFSP1 treatment as compared to HT-1080 cells, cell viability of two of them was partially decreased upon icFSP1 treatment, which was rescued by Lip-1, as shown below (Extended Data Fig. 10e).

Regarding these additional results, we are now discussing in the main text as follows on page 13: *Although inhibition of FSP1 alone is generally not sufficient to drive various cancer cell death via ferroptosis^{5,18}, a subset of cancer cell lines can also be potentially sensitive to FSP1 inhibition alone under certain conditions, such as seen for HT-1080. In this respect, database analysis might be useful to predict the sensitivity of certain cancers toward FSP1 inhibition (Extended Data Fig. 10).*

3.) Since the in vivo models were in a mouse background (where icFSP1 would not be expected have an effect as it appears to be specific to human), it is difficult to discern the specificity of icFSP1 to cancer cells especially in the in vivo context. It would be helpful to include viability experiments from few non-cancerous cell lines to determine if the icFSP1 would be specific to inhibiting cancer cells, or alternatively if there a role for icFSP1 in non-cancerous cells?

We thank the reviewer for raising this valid point, as the species-specificity of icFSP1 toward human FSP1 would not allow to directly assess potential side-effects in mice. Per the reviewers' suggestion, we additionally investigated the cell viability analysis of HEK293T cells (non-cancerous cells) treated with high concentrations of icFSP1 (the highest concentration was limited owing to its solubility properties). While icFSP1 did not show any toxic effects even at high concentrations, iFSP1 showed toxic effects in HEK293T cells already at concentrations higher than 10 μM that could not be rescued by liproxstatin-1 (Extended Data Fig. 1i).

Based on these additional results, we adapted the main text, which now reads as follows on page 4:

To investigate whether icFSP1 has an off-target effect, FSP1 inhibitors (icFSP1 and iFSP1) were treated with HT-1080 and HEK293T cells for 72 h, and primary Peripheral Blood Mononuclear Cells

(PBMC) for 24 h. Thereby, icFSP1 shows no off-target activity at higher concentrations as compared to iFSP1 (Extended Data Fig.1h-j).

4.) Further to the point in #3, FSP1 appears to be highly expressed in bone marrow and other tissues, it will be important to speculate on the off-target effects of this drug which would need to be considered in a clinical setting. Inclusion of data or discussion to contribute to this point would greatly increase the impact of this work.

Related to 3), we additionally investigated the cell viability of Primary Peripheral Blood Mononuclear Cells (PBMC, human normal) exposed to iFSP1 and icFSP1. Similar to the results obtained with HEK293T cells, this data shows that icFSP1 does not show any off-target effect on normal cells, even when used at high concentrations as compared to iFSP1 (Extended Data Fig. 1j).

To further rule out potential off-target effects, we performed additional experiments showing that FSP1 KO cancer cells cannot be further sensitized to icFSP1 in synergy with GPX4 inhibition. This data suggests that icFSP1 is specific to FSP1 and does not target other ferroptosis key players known to enhance ferroptosis sensitivity (Extended Data Fig. 1l-o).

We accordingly modified the main text, which now reads as follows on page 4:

Besides, FSP1 KO cells do not show any additional synergistic effect with icFSP1 treatment after GPX4 inhibition (Extended Data Fig.1k-o). Thus, icFSP1 is considered to be a selective FSP1 inhibitor.

Moreover, FSP1 KO mice (*Aifm2^{tm1Marc}*) are fully viable and display no overt phenotype under normal housing conditions (see Tonnus *et al.* Nat Commun 2021, PMID: 34285231; Mishima *et al.* Nature 2022, PMID: 35922516); thus, we deem FSP1 as a very attractive target for cancer therapy likely allowing for a broad therapeutic window.

Regarding this point, we amended the main text which now reads as follows on page 13:

In light of the fact that *Fsp1* knockout mice are fully viable³⁵, and that icFSP1 treatment does not show any off-target activity and does not affect body weight even under high concentrations (Extended Data Figures 1 and 8), FSP1 should be regarded as an attractive target for tumor treatment. Thus, future studies should be geared towards developing pharmacological approaches that simultaneously target both the cyst(e)ine/GSH/GPX4 node and the FSP1 system allowing for efficient tumor cell eradication by triggering ferroptosis as a new anticancer paradigm.

5.) In light of the author's recent publications, to what extent is icFSP v iFSP acting via ubiquinone versus vitamin E/K? As shown in Doll et al., Nature 2019, CoQ10 is a prime substrate for FSP1. Can icFSP1, as an in vivo inhibitor of FSP1, help to shed light on this preference?

We thank the reviewer for raising this valid point. According to the reviewers' suggestion, we investigated the differences between ubiquinone and vitamin K (exemplified with Vitamin K₃) as a substrate for FSP1 inhibition assay, as shown in the new Extended Data Fig. 3c-h.

The inhibition kinetics of FSP1 inhibitors using CoQ₀ and menadione as substrates, which have the shortest side chains of ubiquinone and vitamin K respectively, are almost the same, with icFSP1 showing again an indirect inhibitory activity FSP1. Thus, it emerges that there is no difference in substrate preference of FSP1 towards CoQ10 vs. vitamin K when comparing iFSP1 and icFSP1 as FSP1 inhibitors in this setting.

Regarding these additional results, we describe in the main text, as follows on page 5:

While hFSP1 enzymatic activity was inhibited by iFSP1 in a cell-free system as reported^{5,6}, icFSP1 failed to inhibit hFSP1 activity at the low micromolar range, albeit it clearly affected cell viability and lipid peroxidation of Pfa1 cells overexpressing hFSP1 (Fig. 2b and Extended Data Fig. 3a-k).

Minor comments

1.) Please clarify why the human cell line experiments in Extended Data Fig. 1c were carried out for 72 hours, whereas the mouse/rat line experiments in Extended Data Fig. 2a-c were carried out for 48 hours. Do you see any changes in viability at 72 hours?

We apologize that the experimental conditions were not clearly delineated. For the combination treatment experiments, we treated cells with BSO for 72 hours as BSO usually requires increased incubation times to fully deplete GSH in order to efficiently induce ferroptosis. The experiments employing human cell lines were carried out for 48 hours when using RSL3, ML210, erastin, FIN56, and FINO2 in the same way like in the mouse/rat cell experiments.

2.) Additionally in Extended Data Fig. 3d, the immunoblots were performed at 48 hours, would you expect differences in expression at 72 hours where you observed viability differences?

Thanks for bringing this up. We additionally performed immunoblot analysis of HT-1080 cells after treatment with icFSP1 for 72h. As illustrated in Extended Data Fig. 3n, it seems there is no major difference by treating cells with either iFSP1 or icFSP1 (Extended Data Fig. 3n).

We accordingly adapted the main text as follows on page 5:

Immunoblotting of hFSP1 expressing H460 or HT-1080 cells after icFSP1 treatment for 48 h and 72h showed that expression levels of hFSP1 were not affected by icFSP1 treatment (Extended Data Fig. 3l-n).

3.) It might be helpful to include a panel for the cell lines shown in Extended Fig 2a-c showing immunoblot analysis of the expression of key ferroptosis players (ACSL4, FSP1, GPX4, XCT) to help interpret the differential responses of the cell lines with icFSP1 as compared to the human cell lines.

Per the reviewers' suggestion, we now include the immunoblot analysis of the expression of key ferroptosis players (ACSL4, FSP1, GPX4, XCT) in murine and rat cell lines, as shown in Extended Data Fig. 2d.

4.) Pg. 7 – consider adding a few sentences explaining the relevance of LLPS in cancer and neurodegeneration

Despite the overall space limitations, we added some explanations in the main text, which now reads as follows on page 7:

These condensates are involved in the regulation of cellular signaling upon stress and have been linked to human diseases including cancer and neurodegeneration^{7,23-26}. In particular, phase separation is involved in the partitioning of target proteins in cancer tissue²⁷, and promotes the formation of aggregates of neurodegenerative disease-related proteins²⁸.

5.) Cell lysis and immunoblotting – please double check that the cells were centrifuged for 1 hour as stated in methods

We apologize for providing insufficient experimental conditions. After centrifugation for a short period of time, cells were sometimes not fully pelleted. Thus, we increased the duration of centrifugation as stated in the methods.

G. References: appropriate credit to previous work?

Authors appropriately provide credit and citations to previous work.

We thank for the reviewer's evaluation.

H. Clarity and context: lucidity of abstract/summary, appropriateness of abstract, introduction and conclusions

The abstract is clear and concise and is an appropriate representation of the introductions and conclusions provided in this manuscript. Adding even further emphasize in the abstract on the novelty that this work is the first evidence for a LLPS-targetable process in ferroptosis should also be considered.

Regarding this comment, we slightly amended the abstract, which now reads as follows:

Hence, our results suggest that icFSP1 displays a unique mechanism-of-action (MoA) and synergizes with ferroptosis-inducing agents to potentiate the ferroptotic cell death response, thus providing the rationale for targeting FSP1-dependent phase separation as an efficient anticancer therapy.

Referee #2 (Remarks to the Author):

Ferroptosis is emerging as an attractive pathway for achieving targeted cell killing in various pathologies, including treatment-resistant cancers. Meanwhile, LLPS is gaining attention as an important molecular activity whose precise regulation is critical for cellular function. Here, Nakamura and colleagues highlight the intersection of these two processes by identifying a small-molecule inhibitor of the ferroptosis checkpoint enzyme FSP1 (icFSP1) that acts not by altering FSP1 catalytic activity but by inducing FSP1 phase separation. The authors demonstrate that icFSP1 promotes ferroptotic cell death in various cancer cell lines and that cellular FSP1 relocalizes into discrete cytosolic puncta within minutes of icFSP1 application. Through both in vitro and cellular characterization, they also demonstrate that FSP1 LLPS is dependent on N- and C-terminal IDRs, an N-terminal LC region, and myristoylation. Finally, Nakamura et al. show that icFSP1 treatment induces ferroptotic death in tumor cells in vivo and can suppress tumor growth in mouse xenograft models. This is an exciting study that should be of great interest to the scientific community. Nevertheless, the results raise several questions that should be addressed to improve the overall rigor of the study and better support some of the authors' conclusions.

We greatly appreciate the very kind words and highly supportive comments made by the reviewer, as well as for recognizing the importance and relevance of our study.

1.) Early in the text (top of page 6), the authors claim that the icFSP1-triggered relocalization of FSP1 via phase separation "causes lipid peroxidation and ferroptosis". While they go on to persuasively show that icFSP1 causes both ferroptosis and FSP1 phase separation, including the formation of FSP1 condensates in implanted tumor cells, the evidence of causal link between the two phenomena rather thin. Such a claim would be better supported by repeating some of the functional assays in cells and in vivo using the LLPS-defective FSP1 mutants described in Fig. 4a.

We apologize for the early statement in the sentence. We modified the text which now reads as following on page 6:

These results indicate that changes of the subcellular localization of hFSP1 precede lipid peroxidation and ferroptosis.

2.) The authors note that unlike the other cell types tested, icFSP1 could trigger cell death in HT-1080 cells even without co-application of a ferroptosis inducer. Why do these cells behave differently? More generally, what is the significance of the cancer cell-type specific variations in

icFSP1-induced responses observed by the authors? It is unclear if icFSP1 will produce consistent effects across different cancer cell types, particularly given that the in vivo studies used only one cancer cell line.

We agree that HT-1080 cells behave differently. HT-1080 cells are one of the most frequently used cancer cell lines in ferroptosis research, likely because of their high sensitivity toward ferroptosis. These reasons for this can be manifold, but likely to the fact that HT-1080 cells are in a highly mesenchymal state and that expression of both GPX4 and FSP1 is rather low, while expression of ACSL4 is prominent (see also comment by reviewer 1, point #2). As such, FSP1 KO or FSP1 inhibitors alone might be sufficient to induce ferroptosis at least to some extent when plated at low cell densities. This phenotype is consistent with other reports showing that FSP1 inhibitor treatment alone (high dose of brequinar) can induce ferroptosis in HT-1080 cells (see also Mao *et al.* Nature 2021, PMID: 33981038 <https://doi.org/10.1038/s41586-021-03539-7>; and a recent preprint from our lab by Mishima *et al.* <https://doi.org/10.21203/rs.3.rs-2190326/v1>). Besides, cell lines other than HT-1080 might be sensitive to FSP1 KO as well, as best illustrated in the Depmap portal (<https://depmap.org/portal/>) (Extended Data Fig. 10), revealing that some cell lines have negative values for the CRISPR score. We therefore investigated some of these cell lines with a small CRISPR score.

Extended Fig. 10

To this end, we additionally addressed whether icFSP1 treatment alone might be sufficient to induce ferroptosis in some human B cell lymphoma cell lines that show negative CRISPR values (i.e., SUDHL-5, SUDHL-6, DOHH2, and OCI-Ly19). While these cell lines were not as sensitive to icFSP1 treatment as compared to HT-1080 cells, cell viability of two of them was partially decreased upon icFSP1 treatment as shown below (Extended Data Fig. 1e).

Yet, most other cancer cell lines generally require inhibition of the XCT/GPX4/GSH axis to efficiently induce ferroptosis as shown in Doll *et al.* Nature 2019 (10.1038/s41586-019-1707-0) and Bersuker *et al.* Nature 2019 (10.1038/s41586-019-1705-2), while some cell lines are resistant to this inhibition of this axis. Thus, we consider that these types of cancer entities might be a good target for icFSP1 to induce ferroptosis synergistically.

Regarding consistent effects across different cancer cell types, we have included two cancer cells, B16F10 (murine cells for the syngeneic model) and A375 (human cells for the xenograft model) cells, for tumor experiments in the initial manuscript. To reinforce the notion that other cancer types can also be applied for icFSP1 treatment *in vivo*, we additionally examined H460 *GPX4* KO cells (human lung cancer cells) in the xenograft model (Extended Data Fig. 9h).

3.) The apparent species-selectivity (i.e., human vs mouse) of icFSP1 is curious. Have the authors verified that icFSP1 binds to mouse Fsp1? For the three mutations (S187C, L217R, Q319K) shown to disrupt icFSP1-induced phase separation of human FSP1, have the authors tested whether the opposite mutations (C187S, R217L, K319Q) sensitize mouse Fsp1 to icFSP1? Furthermore, given the authors' speculation that the specific residues in FSP1 may be involved in protein-membrane interactions, how does this impact the function of mouse Fsp1? Does the mouse enzyme use a different mechanism?

We thank the reviewer for bringing up this very interesting point. Per the reviewers' suggestion, we investigated the opposite combined mutations (i.e., C187S, R217L, K319Q) in mouse FSP1. Interestingly, the compound mutant sensitizes mouse FSP1 to icFSP1, in that it clearly increases

condensate formation and reduces cell viability, as shown in new Extended Data Fig. 7. Hence, these amino acids are indeed required for condensate formation.

Regarding the functional differences between human and mouse FSP1, both FSP1 can prevent ferroptosis (Extended Data Fig. 2j) because both orthologues contain the myristoylation motif (MGxxxS), which enables FSP1 to attach to cellular membranes. Moreover, the G2A mutant, which is a myristoylation-defective mutant of FSP1, invariably fails to rescue from ferroptosis, whereas FSP1 with an N-terminal membrane targeting sequence (i.e., lyn11 tag) protects against ferroptosis (Extended Data Fig. 6f; Doll *et al.* Nature 2019, Bersuker *et al.* Nature 2019, Mishima *et al.* Nature 2022). These results show that myristoylation and proper localization are essential to confer the anti-ferroptotic function of FSP1. Due to the structural similarities between human and murine FSP1 at the N-terminus, we are confident that mouse FSP1 involves the same mechanism. Although

the structural and molecular mechanisms of how myristoylated FSP1 in fact binds to lipid bilayers remains elusive (despite more than seven years of research alone in our and our collaborators' labs), we believe that future structural approaches including cryo-EM will help to shed light into this conundrum.

Regarding these additional results, we amend the main text as follows on page 10:

Given that the binding of icFSP1 toward FSP1 was not affected by these mutants (Extended Data Fig. 7f) and reverse mutations of the three residues to the human variants allow mouse FSP1 to form icFSP1-dependent condensate formation and induce ferroptosis (Extended Data Fig. 7g-i), S187C, L217R, and Q319K may have critical effects on FSP1-FSP1 interactions to trigger phase separation in cells in the presence of icFSP1 (albeit the precise binding site of icFSP1 to WT FSP1 remains to be structurally resolved).

4.) How were the in vitro condensates characterized as being solid-like? Was this based entirely on sedimentation assays? The authors should perform in vitro FRAP analyses to more rigorously investigate this point.

Per the reviewers' suggestion, we additionally characterized *in vitro* condensates using FRAP analysis as shown in new Extended Data Fig. 5d.

Regarding these additional results and suggestions from reviewer#4, we amend the main text as follows on page :

Purified hFSP1 was then reconstituted with 10% PEG, whereupon hFSP1 immediately formed viscoelastic material^{B0}, in contrast to immunoprecipitated EGFP-Strep controls (Fig. 3d-e and Extended Data Fig. 5c-d).

5.) The authors claim that icFSP1 did not promote FSP1 condensate formation in vitro. However, the authors only performed this assay at a single concentration of purified FSP1. To better support their conclusion, the authors should test different FSP1 concentrations to determine whether icFSP1 can shift the phase diagram of FSP1 LLPS.

Related to the reviewers' suggestion, we additionally investigated *in vitro* condensates with higher concentrations of icFSP1 and recombinant non-myristoylated FSP1 (non-myr-hFSP1) in Extended Data Fig. 5h, 6f. While PEG increases turbidity with higher concentrations of FSP1, icFSP1 alone cannot induce phase separation and does not change the phase diagram (Extended Data Fig. 5h, 6f). However, we note that the concentrations of FSP1 and icFSP1 (in aqueous solution) we can reach for *in vitro* experiments are rather low (Extended Data Fig. 6h), our cell-free system using recombinant FSP1 and icFSP1 might not reflect the in the cellular situation, where local concentrations of FSP1/icFSP1 are likely increased by its membrane localization (please also see the discussion about myristoylation regarding #6 of the reviewer comment below).

6.) Similarly, why is FSP1 myristoylation required for phase separation in cells but not in vitro? The authors should at least compare the phase diagrams of FSP1 with and without myristoylation. Based on the literature cited and the schematic presented in Fig. 4i, myristoylation should be expected to lower the concentration threshold for FSP1 phase separation, if not also produce more liquid-like condensates.

We thank the reviewers' comment and suggestions. We fully agree that the environments in cells and *in vitro* are very different, and we are aiming to reconstitute the cellular conditions by considering various factors, especially protein concentration, and the presence of cofactors. Thus, we first used high concentrations of PEG to simulate the crowded cell environment to initiate phase separation of purified protein (natively myristoylated FSP1) from HEK cells. Although we could induce the formation of viscoelastic material at low protein concentration using PEG (Fig. 3d-e and Extended Data Fig. 5c-d), we could not see a notable difference in phase separation of myristoylated (FSP1-WT-ES) and non-myristoylated FSP1 (FSP1-G2A-ES) (Extended Data Fig. 6c), at the concentrations that could be tested.

To obtain higher protein concentration for more conclusive phase separation studies (phase diagrams), we set out to produce recombinant FSP1 with or without myristoylation by co-expressing or enzymatic treatment of recombinant FSP1 with NMT (N-myristoyl-transferase). We could prepare recombinant myristoylated FSP1 as confirmed by mass spectrometry; however, we faced difficulties in obtaining soluble myristoylated FSP1 protein. Although FSP1-EGFP eluted as a monomeric protein at the final steps of purification, myristoylated FSP1-EGFP mostly eluted in the void fraction of the column, indicating that the myristoylated FSP1 is highly oligomeric or aggregated. This observation suggests that contrary to other myristoylated proteins, such as myristoylated methionine sulfoxide reductase A (PDB: 2L90, [10.1074/jbc.M112.368936](https://doi.org/10.1074/jbc.M112.368936)), or the GTPase ARF1 (PDB: 2K5U, [10.1016/j.str.2008.10.020](https://doi.org/10.1016/j.str.2008.10.020)), the myristoyl moiety in myristoylated FSP1 may not be shielded by interacting with the globular domain of FSP1, thus contributing to limited solubility and aggregation propensity of the myristoylated protein in the absence of a membrane.

Despite inherent difficulties in obtaining soluble myristoylated FSP1, we nonetheless investigated the difference in the phase diagram of non-myristoylated FSP1 and myristoylated FSP1 with different concentrations of PEG or icFSP1 (Extended Data Fig. 6e,f). Although there were no differences in the phase boundary of condensates, we observed that myristoylated FSP1 may have a higher probability of forming phase-separated condensates at the same concentrations.

e recombinant FSP1

f PEG induced phase separation

icFSP1 induced phase separation

These data indicate that further optimization of sample and experimental conditions will be required to reproduce the conditions that promoted phase separation in the cellular environment. To this end, we explored the role of the membrane environment, as our *in cells* data showed the importance of membrane localization for the phase separation. Membrane localization is expected to increase the local concentration of FSP1 and may thereby enhance or even enable phase separation upon icFSP1 treatment. We therefore performed experiments to test phase separation in context of the membrane *in vitro*. To this end, we reconstituted the membrane environment by using supported lipid bilayers of different lipid compositions. Our preliminary data shows that the very limited incorporation of recombinantly produced myristoylated FSP1-EGFP (Figure for the reviewer, see below) and not non-myristoylated FSP1-EGFP into the membrane.

As the reconstitution of myr-FSP1 to monitor *in vitro* droplet formation is very challenging, a potential further optimization may involve stabilizing the myristoylated FSP1 by a GDI-like solubilizing factor (GSF) during purification. This may facilitate the preparation of soluble myristoylated FSP1 for further experiments in the future to comprehensively explore the contributions of membrane composition, icFSP1, and FSP1 structure and its cofactors and/or potential additional binding proteins for phase separation.

In light of these intrinsic difficulties in studying the recombinant myristoylated FSP1 *in vitro*, we decided to investigate side-by-side the behavior of myristoylation-proficient and -deficient FSP1 in the cellular context.

Related to your comment and to the role of myristoylated FSP1 in phase separation upon icFSP1 treatment, we additionally analyzed phase separation of FSP1 with and without myristoylation in cells by generating cells that stably co-express FSP1-G2A-EGFP and FSP1-BFP. While phase separation of myristoylated FSP1 can be induced by icFSP1, non-myristoylated FSP1 (G2A) failed to form condensates even when FSP1 condensates of the wildtype enzyme were present. Hence, these data suggest that myristoylation (i.e., membrane localization) itself is an essential requirement for condensation in a cellular context and that the interaction between myristoylation and its potential function as a sticker in concert with icFSP1 (in the sticker-spacer model for phase separation) is necessary for phase separation in cells (Extended Data Fig. 6g).

Given that the conformation of myristoylated proteins can be modulated after some stimulus, such as calcium binding and post-translational modifications and that membrane-bound myristoylated FSP1 (not freely distributed FSP1) is required for triggering phase separation in cells, membrane binding and/or additional factors might be required for phase separation *in vitro*.

Regarding these additional results, we amended the text and discussed more the hypothesis of myristoylation and icFSP1 interactions as also suggested by Referee#4 as follows:

(Page 9)

Next, we produced recombinant hFSP1 with myristoylation (myr-FSP1) purified from *E. coli* (Extended Data Fig. 6d) to test whether myristoylation may afford *in vitro* phase separation induced by lower concentrations of PEG, lower salt and icFSP1 (Extended Data Fig. 6e,f). While PEG or lower salt seem to facilitate phase separation of myr-FSP1 *in vitro*, icFSP1 alone did not induce phase separation, suggesting that the cellular context (i.e. other binding partners, the

membrane environment, post-translational modifications etc.) are important for phase separation in vitro. Furthermore, phase separation of myristoylated FSP1 can be induced by icFSP1, while non-myristoylated FSP1 (G2A) failed to form condensates even when FSP1 condensates of the wildtype enzyme were present in Pfa1 cells (**Extended Data Fig. 6g**). These data imply that the presence of a myristoylation tag facilitates condensate formation as seen for other myristoylated proteins like enhancer of zeste 2 polycomb repressive complex 2 subunit (EZH2)³². Moreover, myristoylation may function as a sticker enhancing polyphasic linkage^{33,34} for phase separation that is modulated by icFSP1, as a ligand.

(Page 13) The concept that ligands modulate the driving forces for phase separation is known as polyphasic linkage^{33,34}. Conceptually, icFSP1 would modulate the phase transitions of FSP1 likely through interactions that directly or indirectly involve residues such as S187, L217, and Q319 and the disruption of interactions by mutagenesis changes the phase boundary. In particular, myristoylation emerges to be indispensable for this process whereby icFSP1, as a ligand, preferentially binds to the myristoylated form of FSP1 (**Extended Data Fig. 6g**).

Referee #3 (Remarks to the Author):

In their study "liquid-liquid phase separation of FSP1 triggers ferroptosis", Nakamura et al. identified a novel FSP1 inhibitor, termed icFSP1, and characterized its mechanism of action. In contrast to FSP1, they find that icFSP1 does not inhibit the catalytic activity of FSP1 but rather mislocalizes FSP1 into lipid condensates. The identification of a novel FSP1 inhibitor will be an important tool for the ferroptosis field. However, there are notable limitations of this study. First, almost all experiments rely on FSP1 overexpressed to very high levels and the similar behavior of endogenous protein is not well demonstrated. Second, the *in vivo* evidence for superiority of icFSP1 is not well established, nor are the on-target effects sufficiently shown *in vivo*. With revisions, this manuscript will likely be of interest to the chemical biology community and a more specialized journal should be considered.

We thank the reviewer for the assessment and the critical comments made and for recognizing the importance of the first *in vivo* efficacious FSP1 inhibitors, icFSP1, and its role in phase separation.

Major points

1. Almost all experiments within this study are performed with overexpressed FSP1, including fluorescence tagged versions. The degree of overexpression is particularly noticeable in Extended Data figure 2D, where the endogenous FSP1 levels are several orders of magnitude lower than the ectopic. Aggregation and/or mislocalization is a known problem with overexpressed proteins.

In fact, all experiments used overexpressed FSP1. Yet, the high expression of FSP1 is required to overcome cell death triggered by the KO of GPX4 in B16F10 or 4T1 cells, as shown in Extended Data Fig. 2D. To compare the expression level of FSP1-EGFP in Pfa1 cells and endogenous FSP1 expression in human cells, we now include an immunoblots analysis of FSP1 as given below (Extended Data Fig. 4e):

Besides, to exclude the possibility of the artefactual mislocalization/aggregation of overexpressed FSP1 enzyme, we established FSP1 KO cells with Doxycycline (Dox)-inducible FSP1-EGFP expression, allowing for the scalable expression of FSP1-EGFP. When we used a Dox concentration that gives comparable or even much lower expression levels of FSP1-EGFP than endogenous wildtype enzyme, we detected condensate-positive cells in icFSP1-treated cells in a Dox-dependent manner (Extended Data Fig. 4f,g).

From this data, we infer that condensate formation also occurs with endogenous FSP1 even when expressed at very low concentrations.

According to these additional results, we amend the main text as follows on page 6-7:

Upon treatment with icFSP1, these condensates were also detectable in H460 cells (expressing only endogenous FSP1), and in cells with even lower expression levels than endogenous FSP1 using doxycycline-dependent, scalable expression of FSP1 (Extended Data Fig. 4d-g).

2. While H460 were used to show condensation of endogenous FSP1, it is unclear whether the few punctae observed are actual condensates. The authors should consider knocking in a tag and the Q319K mutant into the endogenous FSP1 locus to definitively demonstrate condensation of endogenous FSP1.

Per the reviewer's suggestion, we tried our best to establish FSP1-EGFP knock-in cells in different (cancer) cell lines. Yet, after repeated rounds of transduction, single cell cloning and genotyping, we invariably failed to obtain FSP1-EGFP knock-in cells. We analyzed at least 40 single cell clones for each cell line (A375, HT-1080, and 786-O, cells) without success. Besides, we tried using HEK293T cells, which are known as the most commonly used cell lines for generating knock-in cell

lines, but, unfortunately, we could not get the proper genotypes even though they showed EGFP signals (please see below as an example – a statistics of the results is also given in the table below).

Example of PCR and microscope image

	tested clones	Donor (HR) insertion	GFP+ cells	correct genotype
A375	43	9	0	0
HT-1080	46	6	0	0
786-O	39	2	0	0
HEK293T	56	18	18	0

Since our repeated attempts were unsuccessful, we alternatively addressed this point by using FSP1 KO cells, engineered to express doxycycline (Dox)-inducible FSP1-WT and FSP1-Q319K-EGFP (see below, Extended Data Fig. 8i,j). This would afford to study condensate formation at varying amounts of FSP1 enzyme.

These studies showed that icFSP1-dependent condensate formation of FSP1 can be triggered at expression levels far below the amount of endogenous FSP1 expression levels. Moreover, concerning the Q319K mutant, if the reviewer is skeptical that these phenomena are due to

mislocalization of overexpressed proteins, we are convinced that it might not be necessary to make a knock-in mutant since overexpression of the Q319K mutant is already resistant to condensations.

Moreover, we further tried to stain for endogenous condensates in tumor tissues using our previously established FSP1 antibody. As shown below, icFSP1 treatment induced FSP1 condensates also in tumors (Extended Data Fig. 9e).

Therefore, although we did not succeed in generating EGFP knockin cell lines in the *FSP1* locus, all these data support that icFSP1 can induce condensations even at the endogenous expression level.

3. The authors argue that icFSP1 is superior for in vivo use compared to iFSP1, but the data do not support improved stability of icFSP1 in vivo. It appears that higher plasma concentrations are achieved with icFSP1 compared to iFSP1, but the decay kinetics look similar. Additional evidence is needed to support this claim. Moreover, the microsomal stability experiments (Extended data figure 8) show a similar stability in human microsomes when comparing icFSP1 and iFSP1.

Microsomal stability:

We agree with the reviewer that the microsomal stability in human microsomes is similar for the two compounds, and certainly warrants further optimization by medicinal chemistry to qualify as a future drug. However, we did find a 2-fold improved microsomal stability in mouse microsomes as reflected by the half-life as well as a 2-fold decrease in the calculated intrinsic clearance. Given

the improvement of stability in mouse liver microsomes, we were able to not only show for the first time *in vivo* efficacy in tumor models, but also validate the mechanism-of-action (MoA) of this drug *in vivo*.

	human		mouse	
	t _{1/2} (min)	Cl _{int} (μL/min/mg)	t _{1/2} (min)	Cl _{int} (μL/min/mg)
icFSP1	24	70	46.6	36
iFSP1	22.7	74	19.4	86

PK study:

Table 4. Selected pharmacokinetic parameters for RS-7509 in male Balb/cAnN mice following IP (10 mg/kg) administration

Animal	Administration	Dose, mg/kg	Pharmacokinetic Parameters							
			T _{max} , min	C _{max} , ng/ml	AUC _{0→t=240min} (AUClast) ng*min/ml	AUC _{0→∞} (AUCINF_obs), ng*min/ml	T _{1/2} (HL_Lambda_z), min	K _{el} (Lambda_z), min ⁻¹	MRT (MRTlast), min	MRT (MRTinf), min
Mice	IP	10	15.0	4150	124000	126000	80.3	0.00864	41.3	15.0

Table 4. Selected pharmacokinetic parameters for RS-13274 in male Balb/cAnN mice following IP (10 mg/kg) administration

Animal	Administration	Dose, mg/kg	Pharmacokinetic Parameters							
			T _{max} , min	C _{max} , ng/ml	AUC _{0→t=240min} (AUClast) ng*min/ml	AUC _{0→∞} (AUCINF_obs), ng*min/ml	T _{1/2} (HL_Lambda_z), min	K _{el} (Lambda_z), min ⁻¹	MRT (MRTlast), min	MRT (MRTinf), min
Mice	IP	10	15.0	15700	1120000	1130000	53.2	0.013	80.2	84.0

Yet, we agree with the reviewer that the calculated terminal elimination half-life T_{1/2} as determined by noncompartmental pharmacokinetic analysis using the WinNonlin Software is similar between the compounds. However, we did detect a 9-fold improvement in the area under the curve (AUC) for plasma drug concentration-time, which is a measure of the actual body exposure to the drug after administration.

As illustrated below, improvement in drug exposure and C_{max} levels allows for icFSP1 plasma levels to maintain concentrations above the EC_{50} value for 6 hours (as compared to 2 h for iFSP1) at a dose of 10 mg/kg.

Extended Fig 8

Besides, icFSP1 already has many other advantages over iFSP1, as summarized in the table below:

		iFSP1	icFSP1
in cells	EC50 [μM]	0.3	0.2
	Off target activity [μM]	more than 10	Not observed
	FSP1 species	Human only	Human only
in vivo	Plasma Cmax. [$\mu\text{g/mL}$]	4	15
	Plasma Cmax. [μM]	12.4	32.6
	Time above EC50	2 h	6 h
Solubility	PEG:PBS = 45% : 55%	Insoluble	Soluble ($\geq 10\text{mg/mL}$)
	40% Captisol aqueous solution	Insoluble	-
	Kolliphor HS - saline (20%:80%)	Insoluble	-
	DMSO:Kolliphor EL: 5% Mannitol = 5%:20%:75%	Insoluble	-
	DMSO:PEG:PBS = 20% : 50% : 30%	Soluble ($\leq 10\text{ mg/mL}$) [but easily precipitate]	Soluble ($\geq 10\text{mg/mL}$) [stable]

For instance, icFSP1 has a more potent EC₅₀ value than iFSP1. This would be an important advantage as one would need less compound to reduce potential side-effects (in case there are any). iFSP1 shows off-target activity at concentrations higher than 10 μM, while icFSP1 does not show any such effects even at the highest concentrations, which are limited owing to its solubility.

Finally, the solubility of iFSP1 is pretty poor as one requires 20% DMSO to dissolve iFSP1 for *in vivo* use. Also, this solution (10 mg/mL) easily precipitates as shown further below. In contrast to iFSP1, icFSP1 does not require DMSO to bring it into solution as it readily dissolves in PEG (10 mg/mL, without precipitation), which is much preferred for *in vivo* studies.

Nevertheless, to reinforce this notion, we additionally investigated the efficacy of FSP1 inhibitors *in vivo*. We tested 50 mg/kg (using 10 mg/mL) for 2 days and noticed that the treatment group transiently showed reduced body weight in the iFSP1 treatment group.

Next, we lowered the amount used for the treatment and monitored tumor growth over time, showing that iFSP1 failed to reduce tumor volume. Upon sacrifice of tumor bearing mice, we noticed many crystalline precipitates (yellow particles) in the abdominal cavity of mice treated with iFSP1.

In summary, icFSP1 is superior to iFSP1 in terms of EC_{50} , microsomal stability in mouse, solubility, drug exposure and efficacy *in vivo*, toxicity and off-target effects.

4. The authors elegantly identify a Q319K mutant of FSP1 that cannot be localized to condensates and show *in vitro* that this mutant is resistant to the effects of icFSP1. It is unclear why this mutant was used to show the ability of icFSP1 to induce condensates *in vivo* (Figure 4h) but was not used in experiments testing the ability of icFSP1 to impair tumor growth and promote lipid peroxidation *in vivo*. These experiments should also be performed.

We thank the reviewer for bringing up this interesting point. We additionally investigated whether the Q319K mutant is resistant to icFSP1 treatment *in vivo*. Consistent with the results in cells, icFSP1 had no effect on the growth of FSP1^{Q319K} expressing cells *in vivo* as compared to vehicle-treated mice. Besides, icFSP1-treated tumors with condensates showed a slightly increased 4-HNE signal, whereas the Q319K mutant expressing cells showed decreased 4-HNE signals as compared to WT upon icFSP1 treatment.

In light of these results, we slightly modified the main text, which now reads as follows on page 11:

Like the results obtained with Pfa1 cells, melanoma cell lines dependent on hFSP1 Q319K were resistant to icFSP1 in cultured cells as well as in vivo (Extended Data Fig. 8e-j), and consistently

FSP1 condensates were not observed after icFSP1 treatment in vivo (Fig. 4h and Extended Data Fig. 8h).

Other points

1. The second sentence in the abstract is misleading, as the recent paper by the Conrad group did not demonstrate that endogenous vitamin K contributes to ferroptosis suppression by FSP1.

We apologize for this misleading sentence. We modified the text on page 2 as follows:

Recently, ferroptosis suppressor protein-1 (FSP1), along with extramitochondrial ubiquinone or exogenous vitamin K and NAD(P)H/H⁺ as an electron donor, has been identified as the second ferroptosis checkpoint, which efficiently prevents lipid peroxidation independent of the cyst(e)ine/glutathione (GSH)/glutathione peroxidase 4 (GPX4) axis⁴⁻⁶.

Referee#4 (Remarks to the Author):

The central result in this work is that inhibition of the ferroptosis suppressor protein by a small molecule involves the induction of condensate formation. This is a novel and exciting mechanism of action, suggesting that instead of directly suppressing the enzymatic activity of FSP-1, the small molecule icFSP1 acts as a ligand that drives FSP1 phase separation. If this result stands up to continued scrutiny, and the data suggest that it might, this could provide an entirely new route for identifying condensate modulators through ligands that drive condensate formation, assuming the condensate inactivates the function that one seeks to inactivate. For me, this was one of the most exciting reads of the year. I see one important issue that needs work and scrutiny, and this pertains to the issue of how icFSP1 drives the condensate formation of FSP1. The in vitro investigations are really important, but they are incomplete / inconclusive in terms of provide the full picture.

We are most grateful for the highly appreciative comments and are delighted to hear that the reviewer recognizes the novelty and huge potential of our work.

Ligands modulate the driving forces for phase separation through a mechanism known as polyphasic linkage. This was articulated by Wyman and Gill in 1980. Please see: <https://doi.org/10.1073/pnas.77.9.5239>. The concept has been applied in recent work to study ligand modulation of phase behavior, and rules have emerged regarding the types of ligands that drive vs. destabilize phase separation.

Please see: <https://doi.org/10.1073/pnas.2017184118> and <https://doi.org/10.1063/5.0050059>.

Why is this important? The drivers of phase transitions are all exemplars of associative macromolecules featuring stickers and spacers. Ligands modulate the phase transitions of these systems through site-specific interactions with the driver macromolecules, and these interactions are invariably different across the phase boundary. This gives rise to the concept of preferential binding, and the fact that there is a dose-dependent enhancement in the driving forces for hFSP1 phase separation in the presence of icFSP1 implies that the small molecule, as a ligand, binds preferentially to the protein in its dense phase. This leads to the question of how this preferential binding comes about. The papers cited above lay out rules for how this can come about. Deciphering the relevant rules will require further experiments. I think the myristoylation adds a multivalent sticker and that the icFSP1 molecule enables physical crosslinking of these moieties. However, there likely are conformational changes as well, suggestive of a myristoylation induced

partial unfolding that is being aided in the dense phase. A scenario for how this can happen was explained recently by unfolding of model proteins in live cells.

Please see: <https://doi.org/10.1016/j.molcel.2022.06.024>.

So, there are two possible revisions to suggest here: First, reanalyze the extant data using the polyphasic linkage formalism, and revise the MS accordingly. Or, second, find a way to follow the myristoyl groups and the small molecules in the dense phase, compare this to the dilute phase, and discern the molecular level MoA. I think the second is a very tall order, and given the exciting nature of the observation, and its timeliness, I would advocate for the first option.

We are extremely grateful for these valuable and insightful comments, which shall guide us in the interpretation of our results.

Related to these comments (see also point #6 by reviewer 2), we additionally investigated phase separation of FSP1 with and without myristoylation in cells by generating cells that stably co-express FSP1-G2A-EGFP and FSP1-BFP. While phase separation of myristoylated FSP1 can be induced by icFSP1, non-myristoylated FSP1 (G2A) failed to form condensates even when FSP1 condensates of the wildtype enzyme were present. Hence, this data suggests that myristoylation (i.e., membrane localization) itself is an essential requirement for condensation in a cellular context and that the interaction between myristoylation and its potential function as a sticker in concert with icFSP1 is necessary for phase separation in cells (Extended Data Fig. 6g).

According to the reviewer's comments and these additional results, we also discuss polyphasic linkage in the main text as follows (see also comments further below):

(Page 9) *Next, we produced recombinant hFSP1 with myristoylation (myr-FSP1) purified from E. coli (Extended Data Fig. 6d) to test whether myristoylation may afford in vitro phase separation induced by lower concentrations of PEG, lower salt and icFSP1 (Extended Data Fig. 6e,f). While PEG or lower salt seem to facilitate phase separation of myr-FSP1 in vitro, icFSP1 alone did not induce phase separation, suggesting that the cellular context (i.e. other binding partners, the membrane environment, post-translational modifications etc.) are important for phase separation in vitro. Furthermore, phase separation of myristoylated FSP1 can be induced by icFSP1, while non-myristoylated FSP1 (G2A) failed to form condensates even when FSP1 condensates of the wildtype enzyme were present in Pfa1 cells (Extended Data Fig. 6g). These data imply that the presence of*

a myristoylation tag facilitates condensate formation as seen for other myristoylated proteins like enhancer of zeste 2 polycomb repressive complex 2 subunit (EZH2)³². Moreover, myristoylation may function as a sticker enhancing polyphasic linkage^{33,34} for phase separation that is modulated by icFSP1, as a ligand.

(Page 13) *The concept that ligands modulate the driving forces for phase separation is known as polyphasic linkage^{33,34}. Conceptually, icFSP1 would modulate the phase transitions of FSP1 likely through interactions that directly or indirectly involve residues such as S187, L217, and Q319, and the disruption of these interactions by mutagenesis changes the phase boundary. In particular, myristoylation emerges to be indispensable for this process whereby icFSP1, as a ligand, preferentially binds to the myristoylated form of FSP1 (Extended Data Fig. 6g).*

In doing so, I would recommend against the assertion that this LLPS followed by solidification. In all likelihood, the authors are not observing an equilibrium transition. What they are observing is the aging of a viscoelastic material, the physics of which is still being sorted, and has very little to do with an equilibrium liquid-to-solid transition. In any case, one cannot argue for this type of a transition based on FRAP. Therefore, as recommended recently, it is best to abandon the adherence to LLPS, stick with phase separation, and revisit the data through the lens of polyphasic linkage. Please see: <https://doi.org/10.1016/j.molcel.2022.05.018>.

We thank you for the comment. We agree that molecular details and the mode of action of the condensate formation will require additional experimental investigations in the future to carefully assess the contributions of membrane, icFSP1, FSP1 residues and its cofactors and/or other binding proteins etc. for phase separation. To this end, we removed the statement and figures about liquid-to-solid transition and refer to the formation of condensates as phase separation. We also briefly suggest that mechanisms of condensate formation by icFSP1 may involve polyphasic linkage and refer to relevant literature suggested by the reviewer.

Regarding this, we describe in the main text as follows:

(Page 7) *Purified hFSP1 was then reconstituted with 10% PEG, whereupon hFSP1 immediately formed viscoelastic materia^{β0}, in contrast to immunoprecipitated EGFP-Strep controls (Fig. 3e and Extended Data Fig. 5c,d).*

(Page 8) *Thus, hFSP1 has the propensity to form condensates induced by phase separation.*

On a technical note, the use of PEG is a concern. It would help to see a titration of how the phase boundary shifts as a function of PEG concentration, lower concentrations are of particular interest. This way, one will get a sense of how strong the driving forces for phase separation are with this system.

We thank you for the suggestion. As shown in the original submission (now in Extended Data Fig. 5f,g), we showed that 5% PEG also supported condensate formation. To provide further proof of our model, we set out to produce recombinant FSP1 with or without myristoylation by co-expressing or enzymatic treatment of recombinant FSP1 with NMT (N-myristoyl-transferase) (see comment #6 from reviewer 2). However, as shown further below we only obtained a minor fraction of soluble myristoylated FSP1 allowing us to do some very limited experiments.

On a second technical note, while the structure of hFSP1 may not be available, an AlphaFold prediction is available. It appears that, as expected, the IDRs overlap with regions of low confidence in prediction, but the LCR does not. Please see the structure here:

<https://www.uniprot.org/uniprotkb/Q9BRQ8/entry>. This structure is suggestive of a couple of possibilities. It might be useful for a molecular level understanding of the MoA.

We thank you for this comment, and now additionally discuss the predicted model, concerning the globular regions of FSP1, and the flanking intrinsically disordered regions (IDR and LCR) in the main text as follows on page 12: In fact, the flanking IDRs are important for the phase separation, but are not well predictable by AlphaFold2.

(Page 12) *In the absence of an experimentally determined three-dimensional structure of FSP1, the amino acid residues, which contributes to affecting icFSP1-induced phase separation of FSP1 based on our mutational analysis, are mapped onto the structure predicted by AlphaFold2^{35,36}. While the globular fold prediction has high confidence, IDR1 is indeed annotated as an uncertain region, in contrast to LCR, which is predicted to exhibit an α -helical conformation with an intermediate score. The role of the LCR needs to be experimentally studied in the future, to see how it may contribute to phase separation and potentially interact with the globular domain FSP1. These interactions may well be modulated by icFSP1 and should help to understand the underlying structural mechanisms.*

It is also suggestive of possible oligomerization being coupled to myristoylation induced increased stabilization. Studies to date suggest that myristoylation significantly enhances protein stability. Please see: <https://doi.org/10.1073/pnas.1008026107>.

This would suggest that the difference across the phase boundary is protein stability, and this in turn enables sticker-sticker interactions between the myristoyl groups and the small molecule.

Related to the reviewers' suggestion, we tried to produce recombinant FSP1 with or without myristoylation by co-expressing or enzymatic treatment of recombinant FSP1 with NMT (N-myristoyl-transferase). However, we faced difficulties in obtaining soluble myristoylated FSP1 protein. Although FSP1-EGFP eluted as a monomeric protein at the final steps of purification, myristoylated FSP1-EGFP mostly eluted in the void fraction of the column, indicating that the myristoylated FSP1 is highly oligomeric or aggregated. This observation suggests that contrary to other myristoylated proteins, such as myristoylated methionine sulfoxide reductase A (PDB: 2L90 [10.1074/jbc.M112.368936](https://doi.org/10.1074/jbc.M112.368936)), or the GTPase ARF1 (PDB: 2K5U, [10.1016/j.str.2008.10.020](https://doi.org/10.1016/j.str.2008.10.020)), myristoylated FSP1 seems not to be folded back into its own structure.

We are exploring conditions to stabilize the myristoylated FSP1 (e.g by a GDI-like solubilizing factor (GSF)) or other binding partners (if they exist) to obtain myristoylated FSP1 for additional experiments in the future.

Despite inherent difficulties in obtaining soluble myristoylated FSP1, we nonetheless investigated the difference in the phase diagram of non-myristoylated FSP1 and myristoylated FSP1 with different concentrations of PEG or icFSP1 (Extended Data Fig. 6e,f). Although there were no differences in the phase boundary of condensates, we observed that myristoylated FSP1 may have a higher probability of forming phase-separated condensates at the same concentrations.

e recombinant FSP1

f PEG induced phase separation

icFSP1 induced phase separation

According to the reviewer's comments and these additional results, we modified the main text as follows on page 9:

Next, we produced recombinant hFSP1 with myristoylation (myr-FSP1) purified from E. coli (Extended Data Fig. 6d) to test whether myristoylation may afford in vitro phase separation induced by lower concentrations of PEG, lower salt and icFSP1 (Extended Data Fig. 6e,f). While PEG or lower salt seem to facilitate phase separation of myr-FSP1 in vitro, icFSP1 alone did not induce phase separation, suggesting that the cellular context (i.e. other binding partners, the membrane environment, post-translational modifications etc.) are important for phase separation in vitro. Furthermore, phase separation of myristoylated FSP1 can be induced by icFSP1, while non-myristoylated FSP1 (G2A) failed to form condensates even when FSP1 condensates of the wildtype enzyme were present in Pfa1 cells (Extended Data Fig. 6g). These data imply that the presence of a myristoylation tag facilitates condensate formation as seen for other myristoylated proteins like enhancer of zeste 2 polycomb repressive complex 2 subunit (EZH2)³². Moreover, myristoylation may function as a sticker enhancing polyphasic linkage^{33,34} for phase separation that is modulated by icFSP1, as a ligand.

As also suggested, we additionally investigated the stability of non-myristoylated and myristoylated FSP1 with and without inhibitor using thermal shift assay as shown below (Figures for the reviewer). This data suggests that icFSP1 may contribute to stabilizing myristoylated FSP1. A potential interaction of icFSP1 with myr-FSP1 at the membrane may therefore contribute to the mechanism of phase separation. We further plan to optimize the *in vitro* analysis and study the mechanisms in the future.

Although this data suggests that icFSP1 may stabilize myr-FSP1, we are not quite sure whether or not we should include this set of data in the revised version. Therefore, we leave the decision up to the reviewer's recommendation.

Reviewer Reports on the First Revision:

Referees' comments:

Referee #1 (Remarks to the Author):

The authors have rigorously addressed my comments. This novel work now presents a statistically robust and complete discovery that will have a great impact in the field of ferroptosis.

Referee #2 (Remarks to the Author):

The authors have thoroughly addressed many of the reviewers' comments. With regards to the causal relationship between FSP1 phase separation and ferroptosis, instead of repeating some of the functional assays in cells and in vivo using the LLPS-defective FSP1 mutants described in Fig. 4a, the authors modified the text to read "... results indicate that changes of the subcellular localization of hFSP1 precede lipid peroxidation and ferroptosis". However, the title of the paper is "Phase separation of FSP1 triggers ferroptosis", which is a conclusion not well supported by the data. The authors should either perform the suggested experiments or modify the title as well as other related conclusions.

Referee #3 (Remarks to the Author):

The authors have done an outstanding job addressing my concerns and the concerns of the other reviewers. The introduction of the C187S R217L and K319Q mutations into mouse FSP1 is a particularly powerful experiment to support both condensate formation and the role of these residues in the mechanism of action. The revised manuscript is now suitable for publication and the authors should be commended for this rigorous work.

Referee #4 (Remarks to the Author):

The authors have made extensive revisions in response to the comments of all reviewers. The revisions address essentially all the comments raised. Undoubtedly there are some shortcomings, but these cannot be addressed at this juncture because one will need more elaborate biochemical reconstitutions that will get us to the details, but will not change any of the findings or the excitement and relevance of the results. Therefore, I do not have any revisions to suggest. Assuming all reviewers are satisfied (I hope they are), I would recommend moving this onto being accepted, reformatted, and published post haste. As for the thermal shift assay, I think they are preliminary and the assay itself tends to have too many issues - any shift assay does. So, I would recommend against not including these data as ED figure or SI figures. The issue of stability across the phase boundary is what needs detailed investigation and this will require an entirely new arsenal of tools.

Author Rebuttals to First Revision:

Referees' comments:

Referee #1 (Remarks to the Author):

The authors have rigorously addressed my comments. This novel work now presents a statistically robust and complete discovery that will have a great impact in the field of ferroptosis.

We greatly appreciate the reviewer's positive comment.

Referee #2 (Remarks to the Author):

The authors have thoroughly addressed many of the reviewers' comments. With regards to the causal relationship between FSP1 phase separation and ferroptosis, instead of repeating some of the functional assays in cells and in vivo using the LLPS-defective FSP1 mutants described in Fig. 4a, the authors modified the text to read "... results indicate that changes of the subcellular localization of hFSP1 precede lipid peroxidation and ferroptosis". However, the title of the paper is "Phase separation of FSP1 triggers ferroptosis", which is a conclusion not well supported by the data. The authors should either perform the suggested experiments or modify the title as well as other related conclusions.

Per the reviewer's suggestion, we changed the title below, which now reads as follows:

"Phase separation of FSP1 promotes ferroptosis"

Referee #3 (Remarks to the Author):

The authors have done an outstanding job addressing my concerns and the concerns of the other reviewers. The introduction of the C187S R217L and K319Q mutations into mouse FSP1 is a particularly powerful experiment to support both condensate formation and the role of these residues in the mechanism of action. The revised manuscript is now suitable for publication and the authors should be commended for this rigorous work.

We are delighted to see that the reviewer appreciates our additional findings and thank her/him again for the great suggestions made.

Referee #4 (Remarks to the Author):

The authors have made extensive revisions in response to the comments of all reviewers. The revisions address essentially all the comments raised. Undoubtedly there are some shortcomings, but these cannot be addressed at this juncture because one will need more elaborate biochemical reconstitutions that will get us to the details, but will not change any of the findings or the excitement and relevance of the results. Therefore, I do not have any revisions to suggest. Assuming all reviewers are satisfied (I hope they are), I would recommend moving this onto being accepted, reformatted, and published post haste. As for the thermal shift assay, I think they are preliminary and the assay itself tends to have too many issues - any shift assay does. So, I would recommend against not including these data as ED figure or SI figures. The issue of stability across the phase boundary is what needs detailed investigation and this will require an entirely new arsenal of tools.

We would like to thank the reviewer once again for her/his insightful and constructive suggestions throughout the review process. Per the referee's suggestion, we will not include the thermal shift assays.